# Minibatch Optimal Transport and Perplexity Bound Estimation in Discrete Flow Matching

**Etrit Haxholli** [1]   **Yeti Z. Gurbuz** [1]   **Ogul Can** [1]   **Eli Waxman** [1]

## Abstract

Discrete flow matching, a recent framework for modeling categorical data, has shown competitive performance with autoregressive models. However, unlike continuous flow matching, the rectification strategy cannot be applied due to the stochasticity of discrete paths, necessitating alternative methods to minimize state transitions. We propose a dynamic-optimal-transport-like minimization objective and derive its Kantorovich formulation for discrete flows with convex interpolants, where transport cost depends solely on inter-state dissimilarity and can be optimized via minibatch strategies. We show that such methods can reduce the number of transitions up to 32 times (1024 to 32) to reach the same generative perplexity without compromising diversity. Additionally, path nondeterminism in discrete flows precludes an instantaneous change-of-variables analogue, preventing precise probability estimation available to continuous flows. We therefore propose two upper bounds on perplexity, enabling principled training, evaluation and model comparison. Finally, we introduce Multimask Flows which outperform masked flows in generative perplexity without compromising diversity, particularly when utilizing minibatch Optimal Transport.

## 1. Introduction

Modeling data distributions is central to machine learning. For continuous data, diffusion and flow models (Sohl-Dickstein et al., 2015; Ho et al., 2020; Song et al., 2020; Lipman et al., 2023) have shown impressive results in generation and density estimation (Song et al., 2021b;c; Chen et al., 2018). Rectified flows particularly excel by enabling high-quality generation with few integration steps.

However, these continuous models lag behind autoregressive models on categorical data (Chen et al., 2023; Gulrajani & Hashimoto, 2024; Li et al., 2022; Dieleman et al., 2022; Strudel et al., 2022).

To address this, recent work has developed discrete diffusion (Austin et al., 2021; Campbell et al., 2022; Meng et al., 2022; Lou et al., 2024; Sahoo et al., 2024; Ou et al., 2025; Shi et al., 2024) and discrete flow models (Campbell et al., 2024; Gat et al., 2024), better suited for categorical data. Such models can accelerate generation and, unlike autoregressive models, naturally enable infilling. We focus on discrete flow matching (DFM), which expands the design space beyond discrete diffusion by allowing arbitrary couplings and inner dynamics. While discrete and continuous flows share similarities that facilitate adapting continuous flow matching results, fundamental differences remain. These arise primarily from the nonexistence of a DFM formulation with deterministic sample paths.

A major implication of non-deterministic sample paths is that we cannot use the rectification strategy from Liu et al. (2023). Since paths in discrete flows are sequences of states, we explore minimizing the number of jumps between states, which can be interpreted as the discrete analogue of path length minimization. Using dissimilarity functions between states, we minimize jumps weighted by dissimilarity, yielding a weighted path-length-oriented dynamic formulation of optimal transport (OT) for discrete flow matching. We derive its Kantorovich formulation, where the cost function depends only on the dissimilarity function and can be optimized using minibatch strategies (Tong et al., 2024; Fatras et al., 2021). This gives a categorical Benamou-Brenier-type theorem for conditional flows with convex interpolants, the latter being the categorical analogue of shortest-path conditional continuous flows. When the dissimilarity function in the dynamic formulation is the discrete metric, the cost function in the Kantorovich formulation becomes the Hamming distance. When the dissimilarity function is the squared Euclidean distance between token embeddings, the induced cost is the squared Euclidean distance between concatenated sequence embeddings, mirroring the continuous case.

An additional implication of stochastic crossing paths in

[1]Metadialog Research. Correspondence to: Etrit Haxholli <etrit.ks@gmail.com>.

*Proceedings of the $43^{rd}$ International Conference on Machine Learning*, Seoul, South Korea. PMLR 306, 2026. Copyright 2026 by the author(s).

DFM is that we cannot use an equivalent of the instantaneous change of variable formula (Chen et al., 2018) for probability estimation. Thus, other approaches are needed for estimating the perplexity. Inspired by the discrete diffusion bounds in Lou et al. (2024); Haxholli et al. (2025), we derive two upper bounds on perplexity for discrete flow matching. These bounds enable theoretically grounded training, model evaluation and comparison with other methods.

Experiments show that minibatch-OT significantly reduces jumps in small-scale experiments and, in realistic settings (GPT2-sized model on OWT), reduces inference steps (thus time) up to 32-fold (from 1024 to 32) to achieve the same generative perplexity. We also introduce multimask flows (DFM-MMF or simply MMF), which outperform masked DFM in terms of generative perplexity, without sacrificing diversity, in particular when combined with minibatch-OT. Finally, we demonstrate that our derived bounds enable comparisons with autoregressive and discrete diffusion models.

In summary, the main **contributions** of this paper include:

- We formulate a weighted path-length-oriented dynamic OT objective that minimizes dissimilarity-weighted jumps between states. We derive its Kantorovich formulation for convex interpolant flows, establishing a categorical Benamou-Brenier-type theorem.

- We derive a precise theoretical perplexity formula, and extend two practical discrete diffusion bounds on perplexity to DFM, providing principled training objectives and enabling comparisons with autoregressive and discrete diffusion models.

- Finally, we introduce multimask flow (DFM-MM), which surpasses masked DFM models in generative perplexity without compromising diversity, with further gains achieved when applying OT. We show minibatch OT reduces inference steps (thus time) up to 32-fold (1024 to 32) while maintaining performance in terms of generative perplexity on GPT2-scale models.

## 2. Preliminaries and Notation

A summary of Discrete Flow Matching is provided below. While the following preliminary is self-contained, we also provide a discrete diffusion introduction in Appendix D.

### 2.1. Discrete Flow Matching

To expand the design space of discrete diffusion models, Campbell et al. (2024); Gat et al. (2024) introduce discrete flow matching. We follow the approach and notation of Gat et al. (2024). In discrete sequence modeling, a sequence (state) $x$ consists of $L$ elements $(x^1, x^2, \ldots, x^L)$. Each position $i$ contains an element $x^i$ from a vocabulary

$\mathcal{V} = [V] = \{1, \ldots, V\}$ of size $V$. Thus, the set of possible sequences is $\mathcal{D} = \mathcal{V}^L$. Two sequences are neighbors if they differ in only one position.

We denote with $p^i(x^i)$ the marginal of $p$ at position $i$, that is, $p^i(x^i) = \sum_{x^{-i}} p(x)$, where $x^{-i} = (x^1 \ldots, x^{i-1}, x^{i+1}, \ldots x^L)$. The following delta function notation will be particularly useful,

$$\delta_y(x) = \prod_{i=1}^{N} \delta_{y^i}(x^i), \text{ where } \delta_{y^i}(x^i) = \begin{cases} 1 & \text{if } x^i = y^i \\ 0 & \text{if } x^i \neq y^i \end{cases}. \tag{1}$$

#### 2.1.1. PROBABILITY FLOWS AND VELOCITIES

In discrete flow matching (Gat et al., 2024), the goal is to acquire a flow $p_t(z) : [0, 1] \times [V]^L \to [0, 1]$ constrained by $\sum_{z \in [V]^L} p_t(z) = 1$ that transforms source (reference) distributions $X_0 \sim p$ to target (data) distributions $X_1 \sim q$. The flow is completely defined by the choice of a probability velocity $u_t(x) : [0, 1] \times [V]^L \to \mathbb{R}^{L \times V}$, such that $u_t(z) = (u_t^1(z), \ldots, u_t^i(z), \ldots, u_t^L(z))$ and $u_t^i : [0, 1] \times [V]^L \to \mathbb{R}^V$, where $u_t^i(z)[x^i \neq z^i] \geq 0$ and $\sum_{x^i \in [V]} u_t^i(z)[x^i] = 0$, for each $i$. The update rule of the probability over states when going from time $t$ to $t + \epsilon$ is defined independently for each position in the sequence as follows: $p_{t+\epsilon|t}^i(x^i|x_t) = \delta_{x_t^i}(x^i) + \epsilon u_t^i(x^i, x_t)$, where we used $u_t^i(x^i, z) := u_t^i(z)[x^i]$. Therefore, we can see that the probability over the states in the next step depends solely on the current state, and that $u_t$ plays a similar role to a transition-rate matrix $Q_t$ in discrete diffusion, completely determining the flow. As such, if we approximate the probability velocity $u_t(z)$ using a neural network $u_t(z; \theta) : [0, 1] \times [V]^L \to \mathbb{R}^{L \times V}$, we can sample from $p$ and generate data from $q$, using the previous update rule. Before modeling the probability velocity $u_t(z)$ however, one must first design an appropriate flow $p_t(z)$ that has a suitable, practically learnable corresponding $u_t(z)$.

#### 2.1.2. CONDITIONAL PROBABILITY FLOWS

Since at time $t = 0$ and $t = 1$ we must have $p_0 = p$ and $p_1 = q$ respectively, we are already restricted regarding the endpoints of the flow. A trivial way to satisfy such constraints is to define

$$p_t(x) = \sum_{x_0, x_1 \in \mathcal{D}} p_t(x|x_0, x_1)\pi(x_0, x_1), \tag{2}$$

where $p_0(x|x_0, x_1) = \delta_{x_0}(x)$, $p_1(x|x_0, x_1) = \delta_{x_1}(x)$ and $\pi(X_0, X_1)$ is an arbitrary joint distribution of $X_0$, $X_1$ satisfying the marginals constraints $p(x) = p_0(x) = \sum_{y \in \mathcal{D}} \pi(x, y)$, $q(y) = p_1(x) = \sum_{x \in \mathcal{D}} \pi(x, y)$. Since the probability velocities update the probability independently for each position, it is natural to define $p_{t|0,1}(x|x_0, x_1)$ in-

dependently for each dimension as in Gat et al. (2024):

$$p_t(x|x_0, x_1) = \prod_{i=1}^{N} p_t^i(x^i|x_0, x_1), \qquad (3)$$

where

$$p_t^i(x^i|x_0, x_1) = (1 - k_t)\delta_{x_0^i}(x^i) + k_t\delta_{x_1^i}(x^i) \qquad (4)$$

with $k_0 = 0, k_1 = 1$ and increasing $k_t$. It is clear that this definition of $p_t(x|x_0, x_1)$ satisfies the conditions $p_0(x|x_0, x_1) = \delta_{x_0}(x)$ and $p_1(x|x_0, x_1) = \delta_{x_1}(x)$. In addition, Gat et al. (2024) show that component $i$ of the conditional probability velocity $u_t(x, z|x_0, x_1)$ corresponding to the flow defined in Equations (3) and (4) is

$$u_t^i(x^i, z|x_0, x_1) = \frac{\dot{k}_t}{1 - k_t}\left[\delta_{x_1^i}(x^i) - \delta_{z^i}(x^i)\right]. \qquad (5)$$

Furthermore, they show that the probability velocity corresponding to the unconditional flow $p_t(z)$ can be written as a weighted sum of conditional probability velocities,

$$u_t^i(x^i, z) = \sum_{x_0, x_1 \in \mathcal{D}} u_t^i(x^i, z|x_0, x_1)p(x_0, x_1|z), \qquad (6)$$

which in the case of Equations (4) and (5) implies, $u_t^i(x^i, z) = \frac{\dot{k}_t}{1-k_t}\left[p_{1|t}^i(x^i|z) - \delta_z^i(x^i)\right]$. One then approximates $u_t^i(x^i, z)$ by simply modeling $p_{1|t}^i(x^i|z)$ with a neural network $p_{1|t}^i(x^i|z; \theta)$ using the cross entropy loss $L$,

$$\mathbb{E}_{t \sim U(0,1)}\mathbb{E}_{x_0, x_1 \sim \pi(x_0, x_1)}\mathbb{E}_{x_t \sim p_{t|0,1}(\cdot|x_0, x_1)} \sum_{i=1}^{L} l_t^{i,\theta}(x_1, x_t) \qquad (7)$$

where $l_t^{i,\theta}(x_1, x_t) = -\log p_{1|t}^i(x_1^i|x_t; \theta)$. It should be mentioned that in Gat et al. (2024), the definition of $p_t^i(x^i|x_0, x_1)$ is given in a more general form, but here we focus on this specific case for the sake of simplicity and since this formulation corresponds to shortest path conditional flows in the continuous framework, that is, $X_t = (1 - t)X_0 + tX_1$.

### 2.1.3. SOURCE AND TARGET DISTRIBUTIONS

As mentioned, points $X_0$ and $X_1$ are sampled from a joint distribution $\pi(x, y)$, i.e., $(X_0, X_1) \sim \pi(X_0, X_1)$, satisfying the marginals constraints $p(x) = \sum_{y \in \mathcal{D}} \pi(x, y)$, $q(y) = \sum_{x \in \mathcal{D}} \pi(x, y)$. As a special case, the training pairs $X_0$ and $X_1$ can be sampled independently, $(X_0, X_1) \sim p(X_0)q(X_1)$. Common instantiations of source distribution $p$ are:
(i) adding a special token value often referred to as a *mask* token, denoted here by $m$, and setting the source distribution to contain only the fully masked sequence, i.e., $(X_0, X_1) = ((m, \ldots, m), X_1)$.
(ii) using uniform distribution over $\mathcal{D}$, which is equivalent to drawing each $x_0^i$ independently to be some value in $[V]$ with equal probability, denoted $p_u(x_0^i)$.

### 2.2. Optimal Transport Background

Optimal transport (OT) formulates distribution matching as an optimization problem over couplings. Given distributions $p_0$ and $p_1$ and a non-negative cost function $c(\cdot, \cdot)$, the Kantorovich formulation is

$$\inf_{\pi \in \Pi(p_0, p_1)} \int c(x_0, x_1)\, \pi(x_0, x_1)dx_0 dx_1, \qquad (8)$$

where $\Pi(p_0, p_1)$ is the set of couplings with marginals $p_0$ and $p_1$.

OT also has a dynamic formulation in continuous spaces. Instead of optimizing directly over endpoint couplings, one considers a time-dependent density $p_t$ and velocity field $v_t$ that transports $p_0$ to $p_1$. The dynamic formulation minimizes the kinetic energy

$$\inf_{(p_t, v_t)} \int_0^1 \int \frac{1}{2}p_t(x_t)\|v_t(x_t)\|^2\, dx_t\, dt, \qquad (9)$$

subject to the continuity equation and endpoint constraints $p_{t=0} = p_0$ and $p_{t=1} = p_1$. For the quadratic cost $c(x_0, x_1) = \|x_0 - x_1\|^2$, the Benamou–Brenier theorem states that this dynamic problem is equivalent to the static Kantorovich problem in Equation (8) (Benamou & Brenier, 2000).

This static-dynamic equivalence is useful for generative modeling. The dynamic view describes the complexity of the paths followed by samples, while the static view provides a practical way to choose source-target pairings. Recent work in continuous flow matching uses minibatch OT couplings to obtain simpler trajectories between source and target samples (Tong et al., 2024).

## 3. Transition Reduction Objectives in Discrete Flow Matching

We first introduce our dynamic optimal transport formulation for discrete flows with per-position convex interpolants and derive its equivalent static (Kantorovich) formulation. Then, since masked flows (Gat et al., 2024) achieve the best practical performance but cannot leverage OT, we introduce multimasked flows. The latter maintain the performance and time-independent predictive probabilities of masked flows while enabling the application of OT approaches.

### 3.1. Dynamic OT Formulation of Discrete Flows with Convex Interpolants and Its Static Equivalent

A central goal in flow-matching research is to reduce the number of function evaluations required for high-quality generation by simplifying trajectories from the source to the target distribution. Simpler paths lower computational cost, are easier for neural networks to learn, and empirically yield higher-quality models.

In the continuous setting, trajectory simplicity is achieved by minimizing kinetic energy as in Equation (9). By the Benamou–Brenier equivalence, this is equivalent to solving the static Kantorovich problem in Equation (8), which Tong et al. (2024) optimize via minibatch OT couplings.

Our goal is to develop an analogous principle and strategy for discrete flow matching. Since DFM paths are stochastic paths over categorical states rather than deterministic curves in Euclidean space, the continuous kinetic energy in Equation (9) does not apply. We therefore define a discrete dynamic objective that penalizes probability mass flowing between categorical states, and show that for the convex interpolants flows, this objective has an exact static Kantorovich form.

We observe that minimizing $\|v_t(x_t)\|^2 = v_{1,t}^2 + ... + v_{d,t}^2$ in Expression (9) corresponds to minimizing the instantaneous movement of particles from their current positions. In the discrete flow setting, there is a natural analogue: we seek to minimize the expected outflowing mass $u_t^i(x^i, x_t)$ for transitions where $x^i \neq x_t^i$. Equivalently, this amounts to maximizing $u_t^i(x_t^i, x_t)$, favoring trajectories where the mass predominantly stays in place rather than flowing between states.

Therefore, the dynamic formulation for DFM minimizes:

$$\int_0^1 \sum_{x_t} \frac{1}{2} p(x_t) \left[ \sum_{i=1}^L \left( \sum_{x^i \neq x_t^i} u_t^i(x^i, x_t) - u_t^i(x_t^i, x_t) \right) \right] dt$$

$$= \int_0^1 \sum_{x_t} p(x_t) \sum_{i=1}^L \sum_{x^i \neq x_t^i} u_t^i(x^i, x_t) dt, \qquad (10)$$

where $x^i \neq x_t^i$ denotes $x^i \in \mathcal{V} \setminus x_t^i$ and $x_t \in \mathcal{D}$. We prove this equals the Kantorovich formulation in Equation (8) when $c(x_0, x_1)$ is the Hamming distance ($d_H$) between sequences (Corollary 1).

The categorical dynamic formulation above treats all tokens equally, yet in practice tokens have varying similarities reflected in their embeddings. We should weight the outflow by token dissimilarity, penalizing transitions to dissimilar states more heavily. Moreover, for large vocabularies, sequences sampled from the source distribution $p(x_0)$ and the target data distribution $q(x_1)$ likely share few matching positions. Consequently, optimizing this expression using OT-minibatches as in Tong et al. (2024), should not offer substantial improvements in realistic DFM settings.

For these reasons, we define the categorical dynamic objective (functional) more generally as follows:

$$\int_0^1 \sum_{x_t} p(x_t) \left[ \sum_{i=1}^L \sum_{x^i} u_t^i(x^i, x_t) s(x^i, x_t^i) \right] dt$$

$$= \int_0^1 \sum_{x_t} p(x_t) \left[ \sum_{i=1}^L \sum_{x^i \neq x_t^i} u_t^i(x^i, x_t) s(x^i, x_t^i) \right] dt, \quad (11)$$

where $s(x^i, x_t^i) \geq 0$ is a dissimilarity function between two tokens $x^i$ and $x_t^i$ that is symmetric and satisfies $s(a, a) = 0$. Our previous formulation in Equation (10) used the discrete metric $s(x^i, x_t^i) = 1 - \delta_{x_t^i}(x^i)$. Another natural choice is the squared Euclidean distance between token embeddings: $s(x^i, x_t^i) = \|e_m(x^i) - e_m(x_t^i)\|^2$. For any choice of dissimilarity function, there exists a corresponding Kantorovich formulation with a cost function determined by that dissimilarity function.

**Theorem 3.1.** *Let $\pi(x_0, x_1)$ be the joint distribution of $x_0$ and $x_1$, and let $p_t$ be a flow defined as in Equations (2, 3, 4) that transforms $p = \sum_{x_1} \pi(x_0, x_1)$ into $q = \sum_{x_0} \pi(x_0, x_1)$. In this setting, the dynamic formulation given in Equation (11) equals the Kantorovich formulation:*

$$\int_0^1 \sum_{x_t} p(x_t) \sum_{i=1}^L \sum_{x^i \neq x_t^i} u_t^i(x^i, x_t) s(x^i, x_t^i) dt$$

$$= \sum_{x_0, x_1} c(x_0, x_1) \pi(x_0, x_1), \qquad (12)$$

*where the cost function is $c(x_0, x_1) = \sum_{i=1}^L s(x_0^i, x_1^i)$.*

We provide here an intuitive explanation of the result, with the rigorous proof deferred to Appendix A.1: The unconditional flow is an average over conditional flows indexed by pairs $(x_0, x_1)$, so it suffices to understand the dynamic cost of one fixed conditional flow and then average over the coupling $\pi$. For a fixed pair $(x_0, x_1)$, the flow can be analyzed position by position because the updates are independent across positions and the jump weight $s(x^i, x_t^i)$ is defined per position. Under the convex conditional flow, position $i$ only transfers mass directly from $x_0^i$ to $x_1^i$: the mass still at $x_0^i$ at time $t$ is $1 - k_t$, while the per-mass transfer rate is $\dot{k}_t/(1-k_t)$, so the amount of mass transported at time $t$ is $\dot{k}_t$ and the total transported mass is $\int_0^1 \dot{k}_t \, dt = 1$. Hence each mismatched position pays $s(x_0^i, x_1^i)$ exactly once, while matched positions pay zero since $s(a, a) = 0$. Therefore, the cost of the fixed conditional flow is $\sum_{i=1}^L s(x_0^i, x_1^i)$, and averaging this quantity over $\pi$ gives the Kantorovich objective $\sum_{x_0, x_1} \pi(x_0, x_1) \sum_{i=1}^L s(x_0^i, x_1^i)$.

The same coordinate-wise argument also shows that the theorem extends immediately to position-specific schedulers of the form $p_t^i(x^i \mid x_0, x_1) = (1 - k_t^i)\delta_{x_0^i}(x^i) + k_t^i \delta_{x_1^i}(x^i)$. Algorithm 1 describes training with minibatch OT for optimizing the Kantorovich formulation. For the categorical dynamic formulation (10), the corresponding Kantorovich cost function is the Hamming distance $d_H$:

**Corollary 3.2.** *If in Theorem 3.1 we choose $s(x^i, x_t^i) = 1 - \delta_{x_t^i}(x^i)$ then $c(x_0, x_1) = \sum_{i=1}^L s(x_0^i, x_1^i) = \sum_{i=1}^L (1 - \delta_{x_0^i}(x_1^i)) = \sum_{i=1}^L \delta_{x_0^i \neq x_1^i} = d_H(x_0, x_1)$.*

Interestingly, if $s(x^i, x_t^i) = \|e_m(x^i) - e_m(x_t^i)\|^2$, the cost function becomes the squared Euclidean distance between

concatenated sequence embeddings, mirroring continuous flow matching:

**Corollary 3.3.** *If in Theorem 3.1 we choose* $s(x^i, x_t^i) = \|e_m(x^i) - e_m(x_t^i)\|^2$ *then* $c(x_0, x_1) = \sum_{i=1}^{L} \|e_m(x_1^i) - e_m(x_0^i)\|^2$ *that is* $c(x_0, x_1) = \|e_m(x_1) - e_m(x_0)\|^2$.

*Remark* 3.4 (On novelty and contribution). Dynamic optimal transport and Benamou–Brenier-type formulations on discrete spaces have been explored in prior work (Maas, 2011; Léonard, 2016). Our result is more specialized: Theorem 3.1 specifically addresses per-position discrete flows with convex interpolants, which cover the standard discrete flows in the literature. By focusing on this specific setting and using path length instead of kinetic energy, our theorem, unlike the classical Benamou–Brenier result, admits *arbitrary* non-negative, symmetric dissimilarity functions beyond quadratic costs. In contrast, Léonard (2016) uses an intrinsic graph distance as the transport cost, while Maas (2011) defines a Wasserstein-like metric using density-dependent edge mobilities. Notably, in our setting, the dynamic and static losses match for every $\pi$ individually, before optimizing either expression, meaning that the static and dynamic functionals are one and the same within this specific DFM family. This justifies the use of minibatch optimal transport, as any reduction of the static cost implies an equal reduction of the dynamical one.

### 3.2. Multi-mask Flows

Standard masked DFM (Gat et al., 2024) uses a Dirac distribution at the fully masked sequence as its source, admitting only the trivial coupling. To enable meaningful couplings, we introduce multimask flow (DFM-MMF) with $V_s$ special mask tokens $m_1, m_2, ..., m_{V_s}$, all distinct from data tokens. Source sequences are uniformly sampled combinations of these masks. Our total vocabulary thus contains $V = V_s + V_d$ tokens: $V_d = 50,257$ data tokens $x_1, ..., x_{V_d}$ from our tokenizer and $V_s$ mask tokens, i.e., $\mathcal{V} = \{x_1, ..., x_{V_d}, m_1, m..., m_{V_s}\}$. At $t = 0$, only mask tokens appear while data tokens are assigned zero probability. This design provides two advantages: denoising probabilities remain time-independent as in masked flow (see Appendix A.6), and mask embeddings are fully unrestricted since they are untied from data-token embeddings. Effectively, we create a "fictitious grid" where each $L$-length sequence holds mass $\varepsilon = \frac{1}{V_s^L}$ for large enough $V_s$. The flow transports these small masses to the data grid, enabling OT.

## 4. Upper Bounds on the Perplexity in Discrete Flow Matching

Perplexity is a key evaluation metric for language models, making it essential to calculate or bound it in the DFM framework. While we provide a precise but computationally

prohibitive formula in Appendix A.5, the next two subsections present practical bounds. These bounds function as both principled training objectives and practical evaluation metrics for *general* discrete flows, offering the same intrinsic and objective assessment as discrete diffusion bounds.

### 4.1. An Upper Bound on the Perplexity

For probability velocities $w$ and $v$, let $I_t^i(w, v, x^i, x_t)$ (or the shortened $I_t^i(w, v)$) denote

$$w_t^i(x^i, x_t) \log \frac{w_t^i(x^i, x_t)}{v_t^i(x^i, x_t)} + v_t^i(x^i, x_t) - w_t^i(x^i, x_t).$$

To derive the first upper bound, we first provide an expression for the KL divergence between the end distributions $\bar{p}_1$ and $\bar{q}_1$ of two flows $\bar{p}_t$ and $\bar{q}_t$. To derive this expression, we extend the approaches of Opper & Sanguinetti (2007, Equation 3) and Haxholli et al. (2025) to DFM models.

**Theorem 4.1.** *For two discrete flows* $\bar{p}_t$ *and* $\bar{q}_t$ *with corresponding probability velocities* $v_t(x^i, x_t)$ *and* $w_t(x^i, x_t)$, *the following equality holds*

$$D_{KL}(\bar{q}_1 \| \bar{p}_1) = D_{KL}(\bar{q}_0 \| \bar{p}_0)$$

$$+ \int_0^1 \sum_{i=1}^{L} \sum_{x_t} \bar{q}_t(x_t) \sum_{x^i \neq x_t^i} \left( I_t^i(w, v) - I_t^i(\tilde{w}, \tilde{v}) \right) dt \quad (13)$$

*where* $\tilde{v}_t(x^i, x_t)$, $\tilde{w}_t(x^i, x_t)$ *are the respective reverse probability velocities, which generate the identical distributions of paths as the forward ones.*

A proof is provided in Appendix A.1. The key idea of the extension is to see the space as a grid, wherein the flow between non-neighbor states becomes negligible for small step sizes.

By Proposition A.1 in Appendix A, $D_{KL}(\bar{q}_1 \| \bar{p}_1)$ depends only on the forward probability velocities and the learned probability ratios between neighbor states. Unfortunately, we lack access to these probability ratios. However, the following statement provides a computable upper bound,

**Theorem 4.2.** *Under the conditions of Theorem 4.1.* $D_{KL}(\bar{q}_1 \| \bar{p}_1)$ *is bounded from above by*

$$\int_0^1 \sum_{x_t} \bar{q}_t(x_t) \sum_{i=1}^{L} \sum_{x^i \neq x_t^i} I_t^i(w, v, x^i, x_t) dt + D_{KL}(\bar{q}_0 \| \bar{p}_0).$$

A proof is provided in Appendix A.1. Motivated by the last result and Lou et al. (2024), we choose $\bar{p}_t(x)$ to be the learned approximation of flow $p_t$ in Equation (2) with the coupling $\pi(x_0, x_1)$, i.e., $\bar{p}_t(x) = p_t(x; \theta)$ and $v_t = u(\theta)$. On the other hand, we choose $\bar{q}_t(x)$ to have the dynamics of $p_t$, but with the coupling $\bar{\pi}(x, y) = p_0(x)\delta_{x_1}(y) = \int \pi(x, z)dz\delta_{x_1}(y)$. Clearly, $\bar{q}_0(x) =$

$p_0(x)$, $\bar{q}_1(x) = \delta_{x_1}(x)$ and $\bar{q}_t(x) = p_{t|1}(x|x_1)$. We notice that since $\bar{q}_0(x) = p_0(x)$ and $\bar{p}_0(x) = p_0(x)$, then $D_{KL}(\bar{q}_0 \| \bar{p}_0) = 0$. Furthermore $D_{KL}(\bar{q}_1(x) \| \bar{p}_1(x)) = D_{KL}(\delta_{x_1}(x) \| p_1(x; \theta)) = -\log p_1(x_1; \theta)$. Thus, for such choices, $-\log p_t(x_1; \theta)$ is bounded from above by

$$\int_0^1 \sum_{x_t} p_{t|1}(x_t|x_1) \sum_{i=1}^{L} \sum_{x^i \neq x_t^i} I_t^i(w_t, u_t(\theta), x^i, x_t). \quad (14)$$

This bounds the negative-log-likelihood (NLL) for *general* DFM models. Shaul et al. (2025) concurrently and independently obtained a similar result via an ELBO-based derivation. Taking expectations over $p_1(x_1)$ on both sides gives a general bound on cross entropy $H(p_1, p_1(\theta))$. For flows with convex interpolants (Equation (4)), the bound becomes

$$H(p_1, p_1(\theta)) \leq \mathcal{B}$$

$$:= \int_0^1 dt \frac{\dot{k}_t}{1 - k_t} \sum_{x_1, x_0} \pi(x_1, x_0) \sum_{x_t} p_{t|1,0}(x_t|x_1, x_0) \sum_{i=1}^{L}$$

$$\left( -\delta_{x_1^i \neq x_t^i} \log p_{1|t}^i(x_1^i|x_t; \theta) + 1 - p_{1|t}^i(x_t^i|x_t; \theta) - \delta_{x_1^i \neq x_t^i} \right). \quad (15)$$

A detailed derivation is provided in Appendix A.2. Hence $e^{\frac{\mathcal{B}}{L}}$ is a computable upper bound of the perplexity $e^{\frac{H(p_1, p_1(\theta))}{L}}$ that can be used for training and evaluation (Algorithm 2 in Appendix B). in Appendix A.5, we provide an expression for the exact perplexity, but which in practice is computationally expensive and requires approximating the learned probability ratios between neighbor states.

## 4.2. An Alternative Upper Bound on the Perplexity

Analogous to Haxholli et al. (2025)'s findings for discrete diffusion models, using the continuity equation, we show that the distribution entropy at the flow's endpoint can be expressed as follows:

**Proposition 4.3.** *Given a discrete flow $\bar{q}_t$ with a corresponding forward velocity field $w_t$, the entropy of distribution $\bar{q}_1$ can be written as*

$$H(\bar{q}_1) = H(\bar{q}_0) +$$

$$\int_0^1 \sum_{x_t} \bar{q}_t(x_t) \sum_{i=1}^{L} \sum_{x^i} w_t^i(x_t^i, x) \frac{\bar{q}_t(x)}{\bar{q}_t(x_t)} \left( \log \frac{\bar{q}_t(x)}{\bar{q}_t(x_t)} - 1 \right) dt \quad (16)$$

*where $x$ is such that $x^{-i} = x_t^{-i}$ and $x^i$ varies in the third sum.*

Combining this with Theorem 4.2 yields a direct upper bound on the cross-entropy between the terminal distributions of two flows:

**Proposition 4.4.** *Under the conditions of Theorem 4.1, the following inequality holds*

$$H(\bar{q}_1, \bar{p}_1) \leq H(\bar{q}_0) - \sum_{x_t} \bar{q}_t(x_t) \sum_{i=1}^{L} \sum_{x^i \neq x_t^i} \tilde{w}_t^i(x^i, x_t) +$$

$$\int_0^1 \sum_{x_0, x_1} \pi(x_0, x_1) \sum_{x_t} \bar{q}_t(x_t|x_0, x_1) \sum_{i=1}^{L} \sum_{x^i \neq x_t^i} \left( v_t^i(x^i, x_t) \right.$$

$$\left. + \frac{\bar{q}_t^i(x|x_0, x_1)}{\bar{q}_t^i(x_t|x_0, x_1)} \tilde{w}_t^i(x_t^i, x) \log \frac{\tilde{w}_t^i(x_t^i, x)}{v_t^i(x^i, x_t)} \right) dt + D_{KL}(\bar{q}_0 \| \bar{p}_0), \quad (17)$$

*where $x$ is defined as in Proposition 4.3.*

This provides another upper bound on perplexity. Indeed, by setting $\bar{q}_t$ as $p_t$ from Equation (2) with coupling $\pi(x_0, x_1)$, and $\bar{p}_t(x) = p_t(\theta)$, so that $H(\bar{q}_1, \bar{p}_1) = H(p_1, p_1(\theta))$, we obtain the DFM extension of the discrete diffusion bound of Haxholli et al. (2025). See Appendix A.3 for details. As shown in Gat et al. (2024), the backward probability velocity $\tilde{w}_t$ can be computed explicitly in important cases: when the coupling is independent $\pi(x_0, x_1) = p_0(x_0)q_1(x_1)$, and when the source is either masked or has i.i.d. dimensions $p_0(x_0) = \prod_{i=1}^{N} p_0(x_0^i)$. In these cases,

$$\tilde{w}_t(x^i, x_t) = -\frac{\check{k}_t}{k_t} \left[ \delta_{x_t^i}(x^i) - p_0^i(x^i) \right]. \quad (18)$$

Since we can compute all terms on the RHS of Inequality (17), it provides an alternative practical upper bound of the perplexity, as described in Algorithm 3, Appendix B. For the special masked dynamics, $p_0^i(x^i) = \delta_m(x^i)$, the two bounds coincide. A derivation can be found in Appendix A.3.1. These bounds provide principled training objectives and serve as effective evaluation metrics.

## 5. Experiments

This section empirically validates our results. As a proof of concept, we first show on small-vocabulary datasets that applying Theorem 3.1 effectively reduces jumps. Therein, we use the dissimilarity function $s(x^i, x_t^i) = 1 - \delta_{x_t^i}(x^i)$. We then confirm our bounds empirically, and in simple settings, estimate their tightness. In Section 5.3, we use the dissimilarity function $s(x^i, x_t^i) = \|e_m(x^i) - e_m(x_t^i)\|^2$, with learnable embeddings, where $e_m(k)$ denotes the $k$-th column of a learnable embedding matrix $E_{d \times V}$. There, we demonstrate that multimask flows outperform masked flows and that in realistic scenarios, applying Theorem 3.1 through minibatch-OT enables models to reach the same generative perplexity as their base counterparts with up to 8-times fewer steps. Further, we conduct ablation studies on three key factors: OT batch size, OT solver, and OT-EMA. Using an exponential moving average (EMA) to decouple

model training from OT optimization allows us to further reduce the number of steps to 32 for the Bow experiments, yielding a 32× total reduction in steps. Finally, we calculate and analyze the Sinkhorn OT overhead, showing that minibatch OT is practical and scales favorably. We use the POT implementation of Flamary et al. (2021) to compute optimal minibatch couplings throughout these experiments.

### 5.1. Proof of Concept Experiments

We trained a time-conditioned GPT-2 transformer with full attention on the Morse-code converted Shakespeare dataset, where non-convertible characters were left unchanged. The source sequence distribution used was Bag-of-Words (BoW). A sample sequence from the BoW is constructed by sampling independently per position from the token frequencies in the training set. Training consisted of 100k iterations, character-level tokenization, sequence length 128, and batch size 64. We compared standard training with minibatch-OT. Minibatch OT increased training time by $0.3\%$ without affecting inference. We used Hamming distance and the Sinkhorn algorithm with entropy regularization parameter $\epsilon$. During inference with 1024 Euclidean steps, we counted token changes at each position across 3,000 generated sequences. Results appear in Table 1. Since unstructured

*Table 1.* Jump reduction on the Shakespeare Morse-code dataset using a GPT-2 model (L=128). Minibatch OT reduces the number of jumps by $\sim 14\%$. RJ denotes Relative Jumps (normalized to the best model). Increasing the Sinkhorn entropy regularization ($\epsilon$) causes performance to regress toward the non-OT baseline.

| MODEL (L=128) | JUMPS | RJ | PERPLEXITY |
|---|---|---|---|
| NORMAL | $85.47 \pm 0.1$ | 1.14 | 2.35 |
| WITH OT $\epsilon = 0.1$ | $82.86 \pm 0.1$ | 1.1 | 2.14 |
| WITH OT $\epsilon = 0.01$ | $\mathbf{74.87 \pm 0.1}$ | **1** | **2.12** |

source sequences require modifying most tokens to generate structured data, there is a natural lower bound on required modifications. In Shakespeare Morse, the vocabulary contains three main tokens, each with probability $\sim 1/3$. Thus, any given token has probability 2/3 of needing change, yielding an expected minimum of $128 \times \frac{2}{3} = 85.33$ jumps for sequence length 128. The standard method's 85.47 jumps nearly matches this theoretical minimum, suggesting near-optimal performance. That OT reduces this to 74.84 is significant, demonstrating that OT-trained models generate samples closer to the source sequence while maintaining unbiased sampling when marginalizing across source sequences. Beyond jump counts, we report perplexity using the bound in Equation 15. For OT-trained models, we apply exact minibatch OT during testing to prevent sample repetition. Additionally, we report relative jumps (RJ), calculated as the ratio of each model's jump count to that of the best performing model. We also performed the same experi-

ments on Shakespeare using a character-level tokenizer (see Appendix C.1). As discussed in Section 3, the increased vocabulary size makes Hamming distance less effective.

### 5.2. Utilizing the Bounds and Estimating Their Tightness

We test in practice the utility of our bounds as optimization targets and evaluation metrics. In Section 4.2, we mentioned that for masked DFM, both bounds coincide and simplify to $\int_0^1 \frac{1}{1-t} \sum_{x_1,x_0} \pi(x_1, x_0) \sum_{x_t} p_{t|1,0}(x_t|x_1, x_0) \sum_{i=1}^{L} \big( - \delta_m(x_t^i) \log p_{1|t}^i(x_1^i|x_t; \theta) dt \big)$. This particular case matches the MD4 bound of Shi et al. (2024) for masked discrete diffusion (Appendix A.4). We denote models trained with this loss as DFM-S, those trained with the loss multiplied by $(1-t)$ as DFM-N, and those trained with cross-entropy as DFM-O. Using the architecture from Section 5.1 with GPT2 tokenization, we trained on OpenWebText (OWT) (Gokaslan & Cohen, 2019) for 400K steps (batch size 512, sequence length 128). Testing on datasets from Lou et al. (2024), DFM-N performed best (Appendix C.2), so we compared DFM-N against SEDD and GPT-2 for longer sequences (L=1024). The performance gap between DFM-N and DFM-S mirrors that of CEDD and CEDDT in diffusion (Haxholli et al., 2025), further validating our bounds. Table 2 demonstrates that our bounds enable direct comparison with autoregressive models, as well as that DFM-N and SEDD perform similarly as expected. A natural question is how tight our bounds are, and whether bound looseness contributes to the GPT-2/DFM-N performance gap. In Appendix C.5, we study this in controlled finite-state settings where data entropy, estimated terminal cross-entropy, and the bound can be evaluated directly. We verify theoretically that the bound is tight when the learned dynamics perfectly match the data distribution. To examine how the gap behaves under modeling error, we introduce imperfect transition models, including a deliberately bottlenecked masked-flow network scaled across vocabulary size $V$ and sequence length $L$. The results show the bound tracks the cross-entropy closely in small settings, with the gap gradually widening as the state space and modeling error increase.

### 5.3. Minibatch-OT on OpenWebText

To test minibatch-OT in practice, we trained a time-conditioned GPT-2-sized model with full attention for 400k iterations on OWT, using batch size 512 and sequence length 128. We compared: (1) a baseline (DFM-B) without OT, and (2) DFM-B-Sinkhorn, that is, DFM-B trained using minibatch-OT (Sinkhorn solver) with the squared Euclidean embedding cost (Corollary 3.3). Both used the GPT-2 tokenizer ($V_d = 50,257$). The source distribution was BoW as in Section 5.1, but with OWT as the training set. The cross-entropy loss was used in both cases. After training,

*Table 2.* Perplexity bound results evaluating DFM-N against SEDD and GPT-2. DFM-N achieves competitive performance with SEDD.

| MODEL (L=1024) | LAMBADA | WIKITEXT2 | PTB | WIKITEXT103 | LM1B |
|---|---|---|---|---|---|
| SEDD | 52.18 | 42.02 | 117.00 | 41.83 | 80.79 |
| DFM-N | 53.19 | 42.00 | **111.58** | 41.64 | 77.87 |
| GPT2 | **49.02** | **37.68** | 134.13 | **37.55** | **58.92** |

*Table 3.* Generative Perplexity Comparisons: BoW flows (DFM-B), Multimask Flows (DFM-MMF) with and without minibatch OT/EMA and masked flows (DFM-S, DFM-N, DFM-O). Evaluated using GPT-2 Large. Asterisks (*) denote the best results across all categories.

| GENERATION STEPS: | 8 | 16 | 32 | 64 | 128 | 1024 |
|---|---|---|---|---|---|---|
| DFM-B | 345.94 | 241.16 | 211.99 | 197.48 | 192.75 | 185.12 |
| DFM-B-SINKHORN | 331.88 | 233.24 | 203.08 | 191.17 | 185.06 | 178.53 |
| DFM-B-EXACT | 335.14 | 235.15 | 206.11 | 194.23 | 188.85 | 180.91 |
| DFM-B-EXACT-EMA | **302.76*** | **208.42*** | **182.40*** | **169.89** | **164.42** | **159.87** |
| DFM-S (MD4 LOSS) | 587.80 | 316.25 | 222.46 | 188.62 | 169.81 | 156.81 |
| DFM-N | **556.73** | **296.25** | 210.11 | 176.34 | 160.17 | 147.07 |
| DFM-O | 560.67 | 300.06 | **208.06** | **175.59** | **159.03** | **146.54** |
| DFM-MMF | 536.50 | 288.38 | 204.77 | 170.61 | 155.45 | 143.48 |
| DFM-MMF-SINKHORN | 525.83 | 283.10 | 199.55 | 167.86 | 153.51 | 141.92 |
| DFM-MMF-EXACT | 518.86 | 281.39 | 199.68 | 168.12 | 153.51 | 141.46 |
| DFM-MMF-EXACT-EMA | **480.61** | **259.44** | **187.68** | **156.62*** | **143.08*** | **132.55*** |

we generated 10,240 samples and evaluated quality using GPT2-large and Llama3.1 8B. OT significantly improved generative perplexity, reducing by 8-fold the generation steps (1024 to 128) needed to match the non-OT model's score (Table 3). Additionally, we measured the total transport cost in both dynamic and Kantorovich formulations. Both formulations yielded similar, lower costs than models trained without minibatch-OT (Appendix C.6).

Further, using the same experimental setting, we test multimasked flows with and without OT. We choose the number of masks $V_s$ to match the number of data tokens $V_d = 50,257$. Table 3 presents all generative perplexity results using GPT2-large for evaluation, and the perplexity bound results are provided in Appendix C.3. In Appendix C.4, we provide the standard deviations, Llama 3.1B evaluation results, and demonstrate through entropy scores that OT preserves the diversity. The Pearson correlation of perplexity bound values and generative perplexity results is 0.933 (Appendix C.7), showing very strong agreement.

### 5.4. Ablations: Isolating Incremental Contributions

To systematically isolate the contributions of our model variants and training mechanisms, we conduct an ablation study tracking the progression from baseline models to our best-performing configurations (Table 3).

First, moving from masked to multimasked flows (DFM-MMF) improves generative perplexity without hurting diversity. Second, introducing minibatch OT via Sinkhorn improves performance further, while switching to an exact OT solver yields only modest additional gains, mostly in the low-step regime. The most substantial improvement comes from addressing a fundamental instability in OT training. Flow matching with minibatch OT entails two coupled optimization processes: selecting the optimal coupling and training the flow model. Because parameter updates alter embedding-based similarities, the optimal coupling changes from one iteration to the next, which can destabilize the flow dynamics. To mitigate this feedback loop, we introduce an Exponential Moving Average (EMA) mechanism. By computing minibatch couplings using moving-average embeddings, we decouple the embeddings used for OT from those being actively updated by the network. Applying this EMA strategy with exact OT (DFM-MMF-Exact-EMA) drives the bulk of the overall improvement.

Baseline BoW flows (DFM-B) follow the exact same incremental pattern: Sinkhorn helps, exact OT is comparable, and exact OT with EMA delivers a dramatic uplift. Ultimately, by using DFM-B-Exact-EMA, the number of inference steps can be reduced 32-fold (from 1024 to 32) to reach the same generative perplexity as the non-OT baseline, entirely without losses in entropy or diversity (Table 12; Appendix C.4). Notably, the DFM-B family outperforms the DFM-MMF family in the few-step regime but is overtaken as the number of steps increases.

Regarding OT batch size, increasing the size to 4096 (while maintaining the flow batch size) slightly improves results at 32 and 64 steps, with more pronounced improvements observed in bound values (Appendix C.8). Finally, we verify that reducing the number of mask tokens to $V_s = 256$ diminishes the performance gains of OT (Appendix C.8).

*Table 4.* Timing (seconds) for 1000 training batches of size 512 with sequence length 128. The symbol 'E' indicates estimated values due to memory constraints (Nvidia GH200 reaches its maximum capacity).

| BATCH SIZE: | 32 | 64 | 128 | 256 | 512 | 1024 | 2048 | 4096 |
|---|---|---|---|---|---|---|---|---|
| POT-SINKHORN (CPU) | 1.94 | 2.23 | 2.99 | 4.91 | 12.57 | 78.90 | 275.91 | 834.77 |
| POT-SINKHORN (GPU) | 8.93 | 43.26 | 93.11 | 156.64 | 149.30 | 150.31 | 179.13 | 265.34 |
| PURE FLOW | 54.7 | 63.6 | 104.9 | 173.0 | 367.8 | 634.4 | $1176^E$ | $2305^E$ |

### 5.5. Scaling Properties of Minibatch-OT

We examine the computational overhead introduced by mini-batch OT training. While OT computation is vocabulary-size independent, its requirements increase with batch size. Table 4 compares the time for 1000 Sinkhorn couplings (CPU and GPU) against 1000 flow updates without OT, at fixed sequence length $L = 128$, where we use the same batch size for OT as for flow training. OT adds only 3.4% overhead in our experiments. Larger batch sizes (1000-4096), as used by realistic LLMs such as GPT-3, PaLM 540B, LLaMA 3 405B, maintain the overhead around 12%, and require GPU acceleration for sizes of 2048-4096.

Sequence length does not have a negative impact on computational scaling. Normally, the primary overhead stems from the solver (e.g. Sinkhorn) operation, which processes pre-computed pairwise sequence distances. Consequently, sequence length does not affect solver's computational cost. We tested the overall role of the length empirically by increasing the sequence length 8 times (from $L = 128$ to $L = 1024$), which yielded only a 4.6x increase in OT computation time (from 12,57 to 57,99 seconds per thousand minibatches). This scaling behavior has important implications for training efficiency: At $L = 128$, OT adds 3.4% to the total training time. At $L = 1024$, this overhead drops below 1.9% because flow-only training time scales at best roughly linearly (up to quadratically if attention dominates) with sequence length, whereas OT in our experiments scaled more slowly. Consequently, the relative cost of mini-batch OT is not expected to increase with sequence length.

In addition, minibatch OT scales favorably with embedding dimension $D$: OT computation scales linearly in $D$, while transformer computation scales quadratically. Consequently, as embedding dimensions grow, the relative overhead of our approach decreases compared to the flow optimization cost. Furthermore, minibatch-OT is agnostic to the number of transformer layers, meaning the relative overhead decreases even further as models scale.

## 6. Related Work

Diffusion-based models have proven highly effective in capturing the structure of continuous data distributions, leading to significant advancements in generative modeling (Song et al., 2021a;c; Kingma et al., 2021; Nichol & Dhariwal, 2021; Saharia et al., 2022; Ramesh et al., 2022). Given their success in image, video and audio synthesis, researchers have explored their applicability to language modeling (Chen et al., 2023; Gulrajani & Hashimoto, 2024; Li et al., 2022; Dieleman et al., 2022; Strudel et al., 2022; Gong et al., 2023; Karimi Mahabadi et al., 2024).

An alternative paradigm for discrete data, particularly in NLP, is discrete diffusion. Introduced by Hoogeboom et al. (2021); Austin et al. (2021) and extended to continuous-time settings (Campbell et al., 2022; Lou et al., 2024), these models offer a structured approach to learning categorical distributions. Training typically uses the variational lower bound or cross-entropy loss, similar to continuous diffusion (Dieleman et al., 2022). To expand the design space of discrete diffusion, Campbell et al. (2024); Gat et al. (2024) introduce discrete flow matching, notably avoiding conditional score ratio calculations during training and thus bypassing matrix exponential computation. Instead of focusing on a path-length-oriented objective, Shaul et al. (2025) define a kinetic energy OT objective, derive the optimum for specific DFM classes, and independently obtain a bound similar to ours from an ELBO perspective. While in this work we focus on pure DFM models, Arriola et al. (2025) introduce block diffusion language models that interpolate between discrete denoising diffusion and autoregressive models. Regarding scaling, Nie et al. (2025) train masked diffusion models up to 1.1B parameters to systematically evaluate against comparable or larger ARMs. Their 1.1B MDM outperforms the 1.1B TinyLlama trained on the same data across four of eight zero-shot benchmarks.

## 7. Conclusion

We developed a weighted path-length dynamic OT objective for DFM that minimizes dissimilarity-weighted jumps between states, derived its Kantorovich formulation establishing a categorical Benamou-Brenier-type theorem. We derived the true perplexity formula, and extended two practical discrete diffusion bounds to DFM, enabling comparisons with autoregressive and discrete diffusion models. Experiments show minibatch OT reduces inference steps up to 32-fold (1024 to 32) while maintaining generative perplexity on GPT2-scale models. Our multimask flow surpasses masked DFM in generative perplexity without sacrificing diversity, with further gains under OT.

## Impact Statement

This paper presents work whose goal is to advance the field of Machine Learning. There are many potential societal consequences of our work, none which we feel must be specifically highlighted here.

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

# A. Theoretical Results

## A.1. Proofs

**Proof of Theorem 3.1:**

We begin with

$$\int_0^1 \sum_{x_t} p(x_t) \left( \sum_{i=1}^L \sum_{x^i \neq x_t^i} u_t^i(x^i, x_t) s(x^i, x_t^i) \right) dt \tag{19}$$

which due to Equation (6) can be rewritten as

$$= \int_0^1 \sum_{x_t} p(x_t) \left( \sum_{i=1}^L \sum_{x^i \neq x_t^i} \sum_{x_0, x_1} u_t^i(x^i, x_t|x_0, x_1) p(x_0, x_1|x_t) s(x^i, x_t^i) \right) dt \tag{20}$$

$$= \int_0^1 \sum_{x_t} \sum_{i=1}^L \sum_{x^i \neq x_t^i} \sum_{x_0, x_1} u_t^i(x^i, x_t|x_0, x_1) s(x^i, x_t^i) p(x_0, x_1, x_t) dt \tag{21}$$

$$= \int_0^1 \sum_{x_0, x_1} \sum_{i=1}^L \sum_{x_t} \sum_{x^i \neq x_t^i} u_t^i(x^i, x_t|x_0, x_1) s(x^i, x_t^i) p(x_t|x_0, x_1) p(x_0, x_1) dt. \tag{22}$$

For $p(x_t|x_0, x_1)$ as in Equation (4), by Equation (5) we have that

$$u_t^i(x^i, x_t|x_0, x_1) = \frac{\dot{k}_t}{1 - k_t} \left( \delta_{x_1^i}(x^i) - \delta_{x_t^i}(x^i) \right). \tag{23}$$

Continuing from Equation (22)

$$\int_0^1 \sum_{x_0, x_1} \sum_{i=1}^L \sum_{x_t} \sum_{x^i \neq x_t^i} \frac{\dot{k}_t}{1 - k_t} \left( \delta_{x_1^i}(x^i) - \delta_{x_t^i}(x^i) \right) s(x^i, x_t^i) p(x_t|x_0, x_1) p(x_0, x_1) dt \tag{24}$$

$$= \int_0^1 \sum_{x_0, x_1} \sum_{i=1}^L \sum_{x_t^i} \sum_{x^i \neq x_t^i} \frac{\dot{k}_t}{1 - k_t} \left( \delta_{x_1^i}(x^i) - \delta_{x_t^i}(x^i) \right) s(x^i, x_t^i) p^i(x_t^i|x_0, x_1) p(x_0, x_1) dt \tag{25}$$

$$= \int_0^1 \sum_{x_0, x_1} \sum_{i=1}^L \sum_{\substack{x_t^i, x^i \\ x_t^i \neq x^i}} \frac{\dot{k}_t}{1 - k_t} \left( \delta_{x_1^i}(x^i) - \delta_{x_t^i}(x^i) \right) s(x^i, x_t^i) p^i(x_t^i|x_0, x_1) p(x_0, x_1) dt \tag{26}$$

$$= \int_0^1 \sum_{x_0, x_1} \sum_{i=1}^L \sum_{\substack{x_t^i, x^i \\ x_t^i \neq x^i}} \frac{\dot{k}_t}{1 - k_t} \delta_{x_1^i}(x^i) s(x^i, x_t^i) p^i(x_t^i|x_0, x_1) p(x_0, x_1) dt \tag{27}$$

$$= \int_0^1 \sum_{x_0, x_1} \sum_{i=1}^L \sum_{\substack{x_t^i, x^i \\ x_t^i \neq x^i \\ x^i = x_1^i}} \frac{\dot{k}_t}{1 - k_t} s(x^i, x_t^i) p^i(x_t^i|x_0, x_1) p(x_0, x_1) dt, \tag{28}$$

where again due to the choice of Equation (4)

$$\int_0^1 \sum_{x_0, x_1} \sum_{i=1}^L \sum_{\substack{x_t^i, x^i \\ x_t^i \neq x^i \\ x^i = x_1^i}} \frac{\dot{k}_t}{1 - k_t} s(x^i, x_t^i) \left( (1 - k_t) \delta_{x_0^i}(x_t^i) + k_t \delta_{x_1^i}(x_t^i) \right) p(x_0, x_1) dt \tag{29}$$

$$= \int_0^1 \sum_{x_0,x_1} \sum_{i=1}^L \left( \sum_{\substack{x_t^i,x^i \\ x_t^i \neq x^i \\ x^i = x_1^i}} \frac{\dot{k}_t}{1-k_t}(1-k_t)\delta_{x_0^i}(x_t^i) + \sum_{\substack{x_t^i,x^i \\ x_t^i \neq x^i \\ x^i = x_1^i}} \frac{\dot{k}_t}{1-k_t} k_t \delta_{x_1^i}(x_t^i) \right) s(x^i, x_t^i)p(x_0,x_1)dt \tag{30}$$

$$= \int_0^1 \sum_{x_0,x_1} \sum_{i=1}^L \left( \sum_{\substack{x_t^i,x^i \\ x_t^i \neq x^i \\ x^i = x_1^i}} \dot{k}_t \delta_{x_0^i}(x_t^i) + 0 \right) s(x^i, x_t^i)p(x_0,x_1)dt, \tag{31}$$

where the second expression is zero since in the sum one must have $x_t^i \neq x^i$ and $x^i = x_1^i$, therefore $x_t^i \neq x_1^i$ which sets $\delta_{x_1^i}(x_t^i)$ to 0. Hence we only have

$$\int_0^1 \sum_{x_0,x_1} \sum_{i=1}^L \sum_{\substack{x_t^i,x^i \\ x_t^i \neq x^i \\ x^i = x_1^i}} \dot{k}_t \delta_{x_0^i}(x_t^i) s(x^i, x_t^i)p(x_0,x_1)dt = \int_0^1 \sum_{x_0,x_1} \sum_{i=1}^L \sum_{\substack{x_t^i,x^i \\ x_t^i \neq x^i \\ x^i = x_1^i \\ x_t^i = x_0^i}} s(x^i, x_t^i)\dot{k}_t p(x_0,x_1)dt. \tag{32}$$

Expression

$$\sum_{\substack{x_t^i,x^i \\ x_t^i \neq x^i \\ x^i = x_1^i \\ x_t^i = x_0^i}} s(x^i, x_t^i) \tag{33}$$

is clearly $s(x_1^i, x_0^i)$ when $x_0^i \neq x_1^i$ and zero otherwise (therefore still $s(x_1^i, x_0^i)$ as $s(a,a)=0$). Thus, we have

$$\int_0^1 \sum_{x_t} p(x_t) \left( \sum_{i=1}^L \sum_{x^i \neq x_t^i} s(x^i, x_t^i)u_t^i(x^i, x_t) \right) dt = \int_0^1 \sum_{x_0,x_1} \sum_{i=1}^L s(x_1^i, x_0^i)\dot{k}_t p(x_0,x_1)dt \tag{34}$$

$$= \sum_{x_0,x_1} \sum_{i=1}^L s(x_1^i, x_0^i)(k_1 - k_0)p(x_0,x_1) = \sum_{x_0,x_1} \sum_{i=1}^L s(x_1^i, x_0^i)(1 - 0)p(x_0,x_1) \tag{35}$$

We conclude that

$$\int_0^1 \sum_{x_t} p(x_t) \left( \sum_{i=1}^L \sum_{x^i \neq x_t^i} u_t^i(x^i, x_t)s(x^i, x_t^i) \right) dt = \sum_{x_0,x_1} c(x_0,x_1)p(x_0,x_1), \tag{36}$$

where $c(x_0,x_1) = \sum_{i=1}^L s(x_1^i, x_0^i)$. $\qquad \square$

**Proof of Theorem 4.1:**

We begin by defining two discrete time Markov chains $\hat{p}_t$ and $\hat{q}_t$ whose timestep sizes are $\epsilon$ and the total number of steps is $K = \lfloor \frac{1}{\epsilon} \rfloor$, such that when $\epsilon \to 0$, their marginal distributions converge to those of the flows $\bar{p}_t$ and $\bar{q}_t$. The KL divergence between the paths of such Markov chains can be written as below:

$$D_{KL}(\hat{q}, \hat{p}) = \sum_{x_{0:K\epsilon}} \hat{q}(x_{0:K\epsilon}) \log \frac{\hat{q}(x_{0:K\epsilon})}{\hat{p}(x_{0:K\epsilon})} = \sum_{x_{0:K\epsilon}} \hat{q}(x_{0:K\epsilon}) \log \left( \frac{\hat{q}(x_0)}{\hat{p}(x_0)} \prod_{k=1}^K \frac{\hat{q}(x_{k\epsilon}|x_{(k-1)\epsilon}, ..., x_0)}{\hat{p}(x_{k\epsilon}|x_{(k-1)\epsilon}, ..., x_0)} \right) \tag{37}$$

$$= \sum_{x_{0:K\epsilon}} \hat{q}(x_{0:K\epsilon}) \left( \sum_{k=1}^K \log \frac{\hat{q}(x_{k\epsilon}|x_{(k-1)\epsilon})}{\hat{p}(x_{k\epsilon}|x_{(k-1)\epsilon})} + \log \frac{\hat{q}(x_0)}{\hat{p}(x_0)} \right) \tag{38}$$

$$= \sum_{k=1}^{K} \sum_{\substack{x_{k\epsilon} \\ x_{(k-1)\epsilon}}} \hat{q}(x_{k\epsilon}, x_{(k-1)\epsilon}) \log \frac{\hat{q}(x_{k\epsilon}|x_{(k-1)\epsilon})}{\hat{p}(x_{k\epsilon}|x_{(k-1)\epsilon})} + \sum_{x_0} \hat{q}(x_0) \log \frac{\hat{q}(x_0)}{\hat{p}(x_0)} \tag{39}$$

$$= \sum_{k=1}^{K} \sum_{x_{(k-1)\epsilon}} \hat{q}(x_{(k-1)\epsilon}) \sum_{x_{k\epsilon}} \hat{q}(x_{k\epsilon}|x_{(k-1)\epsilon}) \log \frac{\hat{q}(x_{k\epsilon}|x_{(k-1)\epsilon})}{\hat{p}(x_{k\epsilon}|x_{(k-1)\epsilon})} + \sum_{x_0} \hat{q}(x_0) \log \frac{\hat{q}(x_0)}{\hat{p}(x_0)} \tag{40}$$

$$= I + D_{KL}(\hat{q}(x_0)\|\hat{p}(x_0)), \tag{41}$$

where

$$I = \sum_{k=1}^{K} \sum_{x_{(k-1)\epsilon}} \hat{q}(x_{(k-1)\epsilon}) \sum_{x_{k\epsilon}} \hat{q}(x_{k\epsilon}|x_{(k-1)\epsilon}) \log \frac{\hat{q}(x_{k\epsilon}|x_{(k-1)\epsilon})}{\hat{p}(x_{k\epsilon}|x_{(k-1)\epsilon})} \tag{42}$$

is a weighted sum of KL divergences with non-negative weights, that is

$$I = \sum_{k=1}^{K} \sum_{x_{(k-1)\epsilon}} \hat{q}(x_{(k-1)\epsilon}) D_{KL}(\hat{q}(x_{k\epsilon}|x_{(k-1)\epsilon})\|\hat{p}(x_{k\epsilon}|x_{(k-1)\epsilon})). \tag{43}$$

First we will simplify notation and write $t_k = (k-1)\epsilon$, as well as $\hat{q}(x_{k\epsilon} = x|x_{(k-1)\epsilon} = z) = \hat{q}_{t_k+\epsilon|t_k}(x|z)$, where $z$ and $x$ are states. Therefore Expression (42) becomes

$$I = \sum_{k=1}^{K} \sum_{z} \hat{q}_{t_k}(z) \sum_{x} \hat{q}_{t_k+\epsilon|t_k}(x|z) \log \frac{\hat{q}_{t_k+\epsilon|t_k}(x|z)}{\hat{p}_{t_k+\epsilon|t_k}(x|z)}. \tag{44}$$

Now, we focus on computing expression $D_{KL}(\hat{q}_{t_k+\epsilon|t_k}(x|z)\|\hat{p}_{t_k+\epsilon|t_k}(x|z))$. The sum

$$\sum_{x} \hat{q}_{t_k+\epsilon|t_k}(x|z) \log \frac{\hat{q}_{t_k+\epsilon|t_k}(x|z)}{\hat{p}_{t_k+\epsilon|t_k}(x|z)}. \tag{45}$$

can be separated into three sums:

$$\sum_{\substack{x \\ d_H(x,z)=0}} \hat{q}_{t_k+\epsilon|t_k}(x|z) \log \frac{\hat{q}_{t_k+\epsilon|t_k}(x|z)}{\hat{p}_{t_k+\epsilon|t_k}(x|z)} + \sum_{\substack{x \\ d_H(x,z)=1}} \hat{q}_{t_k+\epsilon|t_k}(x|z) \log \frac{\hat{q}_{t_k+\epsilon|t_k}(x|z)}{\hat{p}_{t_k+\epsilon|t_k}(x|z)}$$

$$+ \sum_{\substack{x \\ d_H(x,z)>1}} \hat{q}_{t_k+\epsilon|t_k}(x|z) \log \frac{\hat{q}_{t_k+\epsilon|t_k}(x|z)}{\hat{p}_{t_k+\epsilon|t_k}(x|z)} \tag{46}$$

We first analyze the second sum. Since $x$ and $z$ differ at exactly one neighbor (say position $j$), from the flow matching update rule $p_{t_k+\epsilon|t_k}^i(y^i|z) = \delta_{z^i}(y^i) + \epsilon u_{t_k}^i(y^i, z)$ applied independently to each position, we can infer that

$$p_{t_k+\epsilon|t_k}(x|z) = \epsilon u_{t_k}^j(x^j, z) \prod_{\substack{i=1 \\ i \neq j}}^{L} \left(1 + u_{t_k}^i(x^i, z)\epsilon\right) = \epsilon u_{t_k}^j(x^j, z) + O(\epsilon^2) \tag{47}$$

therefore

$$\sum_{\substack{x \\ d_H(x,z)=1}} \hat{q}_{t_k+\epsilon|t_k}(x|z) \log \frac{\hat{q}_{t_k+\epsilon|t_k}(x|z)}{\hat{p}_{t_k+\epsilon|t_k}(x|z)} = \sum_{j=1}^{L} \sum_{x^j \neq z^j} \epsilon w_{t_k}^j(x^j, z) \log \frac{w_{t_k}^j(x^j, z) + O(\epsilon)}{v_{t_k}^j(x^j, z) + O(\epsilon)} + O(\epsilon^2). \tag{48}$$

For the third sum, since

$$p_{t_k+\epsilon|t_k}(x|z) = \epsilon^2 u_{t_k}^j(x^j, z) u_{t_k}^l(x^j, z) \prod_{\substack{i=1 \\ i \neq j,l}}^{L} \left(1 + u_{t_k}^i(x^i, z)\epsilon\right) = O(\epsilon^2) + O(\epsilon^3) = O(\epsilon^2) \tag{49}$$

we conclude that

$$\sum_{\substack{x \\ d_H(x,z)>1}} \hat{q}_{t_k+\epsilon|t_k}(x|z) \log \frac{\hat{q}_{t_k+\epsilon|t_k}(x|z)}{\hat{p}_{t_k+\epsilon|t_k}(x|z)} = O(\epsilon^2). \tag{50}$$

Therefore the only sum left is the first one

$$\sum_{\substack{x \\ d_H(x,z)=0}} \hat{q}_{t_k+\epsilon|t_k}(x|z) \log \frac{\hat{q}_{t_k+\epsilon|t_k}(x|z)}{\hat{p}_{t_k+\epsilon|t_k}(x|z)} = \hat{q}_{t_k+\epsilon|t_k}(z|z) \log \frac{\hat{q}_{t_k+\epsilon|t_k}(z|z)}{\hat{p}_{t_k+\epsilon|t_k}(z|z)}. \tag{51}$$

In this special case ($x = z$),

$$p_{t_k+\epsilon|t_k}(z|z) = \prod_{i=1}^{L} \left(1 + u_{t_k}^i(z^i, z)\epsilon\right) = 1 + \epsilon \sum_{i=1}^{L} u_{t_k}^i(z^i, z) + O(\epsilon^2), \tag{52}$$

thus

$$\hat{q}_{t_k+\epsilon|t_k}(z, z) = 1 + \epsilon \sum_{i=1}^{L} w_{t_k}^i(z^i, z) + O(\epsilon^2), \tag{53}$$

$$\log \hat{q}_{t_k+\epsilon|t_k}(z, z) = \epsilon \sum_{i=1}^{L} w_{t_k}^i(z^i, z) + O(\epsilon^2), \tag{54}$$

$$\log \hat{p}_{t_k+\epsilon|t_k}(z, z) = \epsilon \sum_{i=1}^{L} v_{t_k}^i(z^i, z) + O(\epsilon^2), \tag{55}$$

implying

$$\hat{q}_{t_k+\epsilon|t_k}(z|z) \log \frac{\hat{q}_{t_k+\epsilon|t_k}(z|z)}{\hat{p}_{t_k+\epsilon|t_k}(z|z)} = \hat{q}_{t_k+\epsilon|t_k}(z|z) \left(\log \hat{q}_{t_k+\epsilon|t_k}(z|z) - \log \hat{p}_{t_k+\epsilon|t_k}(z|z)\right) \tag{56}$$

$$= \epsilon \sum_{i=1}^{L} w_{t_k}^i(z^i, z) - \epsilon \sum_{i=1}^{L} v_{t_k}^i(z^i, z) + O(\epsilon^2). \tag{57}$$

When accounting for the fact that $u_{t_k}^i(z^i, z) = -\sum_{x^i \neq z^i} u_{t_k}^i(x^i, z)$, we finally have

$$\hat{q}_{t_k+\epsilon|t_k}(z|z) \log \frac{\hat{q}_{t_k+\epsilon|t_k}(z|z)}{\hat{p}_{t_k+\epsilon|t_k}(z|z)} = \epsilon \sum_{i=1}^{L} \sum_{x^i \neq z^i} \left(v_{t_k}^i(x^i, z) - w_{t_k}^i(x^i, z)\right) + O(\epsilon^2). \tag{58}$$

We get the expression for $D_{KL}(\hat{q}_{t_k+\epsilon|t_k}(x|z)\|\hat{p}_{t_k+\epsilon|t_k}(x|z)))$ by adding all three sums,

$$\epsilon \sum_{i=1}^{L} \sum_{x^i \neq z^i} \left( w_{t_k}^i(x^i, z) \log \frac{w_{t_k}^i(x^i, z) + O(\epsilon)}{v_{t_k}^i(x^i, z) + O(\epsilon)} + v_{t_k}^i(x^i, z) - w_{t_k}^i(x^i, z) \right) + O(\epsilon^2). \tag{59}$$

Plugging this last expression in $I$, one gets

$$I = \sum_{k=0}^{K-1} \epsilon \sum_{z} \hat{q}_{t_k}(z) \sum_{i=1}^{L} \sum_{x^i \neq z^i} \left( w_{t_k}^i(x^i, z) \log \frac{w_{t_k}^i(x^i, z) + O(\epsilon)}{v_{t_k}^i(x^i, z) + O(\epsilon)} + v_{t_k}^i(x^i, z) - w_{t_k}^i(x^i, z) \right) + O(\epsilon). \tag{60}$$

Finally taking the limit $\epsilon \to 0$

$$\int_0^1 \sum_{z} \bar{q}_t(z) \sum_{i=1}^{L} \sum_{x^i \neq z^i} \left( w_t^i(x^i, z) \log \frac{w_t^i(x^i, z)}{v_t^i(x^i, z)} + v_t^i(x^i, z) - w_t^i(x^i, z) \right) dt. \tag{61}$$

Based on the last formula, and by replacing $z$ with $x_t$, we can write,

$$D_{KL}(\bar{q}, \bar{p}) = D_{KL}(\bar{q}_0 \| \bar{p}_0) + \int_0^1 \sum_{x_t} \bar{q}_t(x_t) \sum_{i=1}^L \sum_{x^i \neq x_t^i} \left( w_t^i(x^i, x_t) \log \frac{w_t^i(x^i, x_t)}{v_t^i(x^i, x_t)} + v_t^i(x^i, x_t) - w_t^i(x^i, x_t) \right) dt. \quad (62)$$

Applying this result when considering flow paths $\bar{p}$ and $\bar{q}$ to have been generated in the opposite direction by the reverse probability velocities $\tilde{v}_t^i(x^i, x_t)$ and $\tilde{w}_t^i(x^i, x_t)$,

$$D_{KL}(\bar{q}, \bar{p}) = D_{KL}(\bar{q}_1 \| \bar{p}_1) + \int_0^1 \sum_{x_t} \bar{q}_t(x_t) \sum_{i=1}^L \sum_{x^i \neq x_t^i} \left( \tilde{w}_t^i(x^i, x_t) \log \frac{\tilde{w}_t^i(x^i, x_t)}{\tilde{v}_t^i(x^i, x_t)} + \tilde{v}_t^i(x^i, x_t) - \tilde{w}_t^i(x^i, x_t) \right) dt \quad (63)$$

Finally, by combining Equations (62) and (63),

$$\int_0^1 \sum_{x_t} \bar{q}_t(x_t) \sum_{i=1}^L \sum_{x^i \neq x_t^i} \left( \tilde{w}_t^i(x^i, x_t) \log \frac{\tilde{w}_t^i(x^i, x_t)}{\tilde{v}_t^i(x^i, x_t)} + \tilde{v}_t^i(x^i, x_t) - \tilde{w}_t^i(x^i, x_t) \right) dt + D_{KL}(\bar{q}_1 \| \bar{p}_1)$$

$$= \int_0^1 \sum_{x_t} \bar{q}_t(x_t) \sum_{i=1}^L \sum_{x^i \neq x_t^i} \left( w_t^i(x^i, x_t) \log \frac{w_t^i(x^i, x_t)}{v_t^i(x^i, x_t)} + v_t^i(x^i, x_t) - w_t^i(x^i, x_t) \right) dt + D_{KL}(\bar{q}_0 \| \bar{p}_0) \quad (64)$$

therefore,

$$D_{KL}(\bar{q}_1 \| \bar{p}_1) = \int_0^1 \sum_{x_t} \bar{q}_t(x_t) \sum_{i=1}^L \sum_{x^i \neq x_t^i} \left( w_t^i(x^i, x_t) \log \frac{w_t^i(x^i, x_t)}{v_t^i(x^i, x_t)} + v_t^i(x^i, x_t) - w_t^i(x^i, x_t) \right) dt$$

$$- \int_0^1 \sum_{x_t} \bar{q}_t(x_t) \sum_{i=1}^L \sum_{x^i \neq x_t^i} \left( \tilde{w}_t^i(x^i, x_t) \log \frac{\tilde{w}_t^i(x^i, x_t)}{\tilde{v}_t^i(x^i, x_t)} + \tilde{v}_t^i(x^i, x_t) - \tilde{w}_t^i(x^i, x_t) \right) dt + D_{KL}(\bar{q}_0 \| \bar{p}_0). \quad (65)$$

$\square$

**Proposition A.1.** *For two discrete flows $\bar{p}_t$ and $\bar{q}_t$ with corresponding probability velocities $v_t(x^i, x_t)$ and $w_t(x^i, x_t)$, the following equality holds*

$$D_{KL}(\bar{q}_1 \| \bar{p}_1) = D_{KL}(\bar{q}_0 \| \bar{p}_0) + \int_0^1 \sum_{x_t} \bar{q}_t(x_t) \sum_{i=1}^L \sum_{x^i \neq x_t^i} \left( w_t^i(x^i, x_t) \log \frac{w_t^i(x^i, x_t)}{v_t^i(x^i, x_t)} + v_t^i(x^i, x_t) - w_t^i(x^i, x_t) \right) dt$$

$$- \int_0^1 \sum_{x_t} \bar{q}_t(x_t) \sum_{i=1}^L \sum_{x^i \neq x_t^i} \left( r_{\bar{q}_t} w_t^i(x_t^i, x) \log \frac{r_{\bar{q}_t} w_t^i(x_t^i, x)}{r_{\bar{p}_t} v_t^i(x_t^i, x)} + r_{\bar{p}_t} v_t^i(x_t^i, x) - r_{\bar{q}_t} w_t^i(x_t^i, x) \right) dt, \quad (66)$$

*where $r_{\bar{p}_t} = r_{\bar{p}_t}(x, x_t) = \frac{\bar{p}_t(x)}{\bar{p}_t(x_t)}$, $r_{\bar{q}_t} = r_{\bar{q}_t}(x, x_t) = \frac{\bar{q}_t(x)}{\bar{q}_t(x_t)}$, and where $x$ is a state identical to the current position $x_t$, except for position (dimension) $i$.*

**Proof of Proposition A.1:** Similarly to the proof of Theorem 4.1 above, one can write

$$D_{KL}(\tilde{q}, \tilde{p}) = J + D_{KL}(\tilde{q}_1 \| \tilde{p}_1), \quad (67)$$

where

$$J = \sum_{k=1}^K \sum_z \tilde{q}_{\tau_k}(z) \sum_x \tilde{q}_{\tau_{k+1}|\tau_k}(x|z) \log \frac{\tilde{q}_{\tau_{k+1}|\tau_k}(x|z)}{\tilde{p}_{\tau_{k+1}|\tau_k}(x|z)}, \quad (68)$$

for $\tau_k = 1 - (k-1)\epsilon = 1 - t_k$. As before, we can break this expression into three sums and then focus on the ones that concern states $x, z$ that do not differ on more than one dimension. In case that $x$ and $z$ differ in exactly one dimension $(j)$ then as previously we have

$$p_{\tau_{k+1}|\tau_k}(x|z) = \frac{p_{\tau_{k+1}}(x)}{p_{\tau_k}(z)} p_{\tau_k|\tau_{k+1}}(z|x) = \frac{p_{1-t_k-\epsilon}(x)}{p_{1-t_k}(z)} p_{1-t_k|1-t_k-\epsilon}(z|x)$$

$$= \epsilon \frac{p_{1-t_k-\epsilon}(x)}{p_{1-t_k}(z)} u_{1-t_k-\epsilon}^j(z^j, x) \prod_{\substack{i=1 \\ i \neq j}}^{L} \left(1 + u_{1-t_k-\epsilon}^i(z^i, x)\epsilon\right) = \epsilon \frac{p_{1-t_k-\epsilon}(x)}{p_{1-t_k}(z)} u_{1-t_k-\epsilon}^j(z^j, x) + O(\epsilon^2). \tag{69}$$

Similarly as before, we can develop the expression for the case when the Hamming distance between $x$ and $z$ is 0. By combining these cases as in the previous theorem and taking $\epsilon \to 0$, we derive an expression for $J$:

$$D_{KL}(\bar{q}, \bar{p}) = D_{KL}(\bar{q}_1, \bar{p}_1)$$

$$+ \int_0^1 \sum_z \bar{q}_{1-t}(z) \sum_{i=1}^{L} \sum_{x^i \neq z^i} \left( r_{\bar{q}_{1-t}} w_{1-t}^i(z^i, x) \log \frac{r_{\bar{q}_{1-t}} w_{1-t}^i(z^i, x)}{r_{\bar{p}_{1-t}} v_{1-t}^i(z^i, x)} + r_{\bar{p}_{1-t}} v_{1-t}^i(z^i, x) - r_{\bar{q}_{1-t}} w_{1-t}^i(z^i, x) \right) dt.$$

We conclude the proof by setting $\tau = 1 - t$,

$$D_{KL}(\bar{q}, \bar{p}) = D_{KL}(\bar{q}_1, \bar{p}_1) + \int_0^1 \sum_z \bar{q}_\tau(z) \sum_{i=1}^{L} \sum_{x^i \neq z^i} \left( r_{\bar{q}_\tau} w_\tau^i(z^i, x) \log \frac{r_{\bar{q}_\tau} w_\tau^i(z^i, x)}{r_{\bar{p}_\tau} v_\tau^i(z^i, x)} + r_{\bar{p}_\tau} v_\tau^i(z^i, x) - r_{\bar{q}_\tau} w_\tau^i(z^i, x) \right) d\tau$$

$$\tag{70}$$

followed by $z = x_\tau$, where $r_{\bar{p}_\tau} = r_{\bar{p}_\tau}(x, z) = \frac{\bar{p}_\tau(x)}{\bar{p}_\tau(z)}$ and $r_{\bar{q}_\tau} = r_{\bar{q}_\tau}(x, z) = \frac{\bar{q}_\tau(x)}{\bar{q}_\tau(z)}$. $\square$

**Proof of Theorem 4.2:**

We now set to prove that $D_{KL}(\bar{q}(x_0)\|\bar{p}(x_0)) \leq D_{KL}(\bar{q}, \bar{p})$ and $D_{KL}(\bar{q}(x_1)\|\bar{p}(x_1)) \leq D_{KL}(\bar{q}, \bar{p})$. Since in Equation (65), the term

$$\int_0^1 \sum_{x_t} \bar{q}_t(x_t) \sum_{i=1}^{L} \sum_{x^i \neq x_t^i} \left( \tilde{w}_t^i(x^i, x_t) \log \frac{\tilde{w}_t^i(x^i, x_t)}{\tilde{v}_t^i(x^i, x_t)} + \tilde{v}_t^i(x^i, x_t) - \tilde{w}_t^i(x^i, x_t) \right) dt \tag{71}$$

is a positively weighted sum of KL divergences this immediately implies that

$$D_{KL}(\bar{q}_1\|\bar{p}_1) \leq D_{KL}(\bar{q}_0\|\bar{p}_0) + \int_0^1 \sum_{x_t} \bar{q}_t(x_t) \sum_{i=1}^{L} \sum_{x^i \neq x_t^i} \left( w_t^i(x^i, x_t) \log \frac{w_t^i(x^i, x_t)}{v_t^i(x^i, x_t)} + v_t^i(x^i, x_t) - w_t^i(x^i, x_t) \right) dt. \tag{72}$$

We can also show this result to be an immediate consequence of the Jensen inequality. Indeed,

$$-D_{KL}(\hat{q}\|\hat{p}) = \sum_{x_{0:K\epsilon}} \hat{q}(x_{0:K\epsilon}) \log \frac{\hat{p}(x_{0:K\epsilon})}{\hat{q}(x_{0:K\epsilon})} = \sum_{x_{0:K\epsilon}} \hat{q}(x_0)\hat{q}(x_{\epsilon:K\epsilon}|x_0) \log \frac{\hat{p}(x_{0:K\epsilon})}{\hat{q}(x_{0:K\epsilon})} \tag{73}$$

$$= \sum_{x_0} \hat{q}(x_0) \sum_{x_{\epsilon:K\epsilon}} \hat{q}(x_{\epsilon:K\epsilon}|x_0) \log \frac{\hat{p}(x_{0:K\epsilon})}{\hat{q}(x_{0:K\epsilon})} \leq \sum_{x_0} \hat{q}(x_0) \log \sum_{x_{\epsilon:K\epsilon}} \hat{q}(x_{\epsilon:K\epsilon}|x_0) \frac{\hat{p}(x_{0:K\epsilon})}{\hat{q}(x_{0:K\epsilon})} \tag{74}$$

$$= \sum_{x_0} \hat{q}(x_0) \log \sum_{x_{\epsilon:K\epsilon}} \frac{\hat{p}(x_{0:K\epsilon})}{\hat{q}(x_0)} = -D_{KL}(\hat{q}(x_0)\|\hat{p}(x_0)) \tag{75}$$

Therefore, $D_{KL}(\hat{q}_0\|\hat{p}_0) \leq D_{KL}(\hat{q}\|\hat{p})$, and taking the limit $\epsilon \to 0$ we get $D_{KL}(\bar{q}_0\|\bar{p}_0) \leq D_{KL}(\bar{q}\|\bar{p})$. Applying this result to the reverse processes that generate the marginals $\bar{p}$ and $\bar{q}$ gives $D_{KL}(\bar{q}_1\|\bar{p}_1) \leq D_{KL}(\bar{q}, \bar{p})$.

In total we have proved that

$$D_{KL}(\bar{q}_1\|\bar{p}_1) \leq D_{KL}(\bar{q}_0\|\bar{p}_0) + \int_0^1 \sum_{x_t} \bar{q}_t(x_t) \sum_{i=1}^{L} \sum_{x^i \neq x_t^i} \left( w_t^i(x^i, x_t) \log \frac{w_t^i(x^i, x_t)}{v_t^i(x^i, x_t)} + v_t^i(x^i, x_t) - w_t^i(x^i, x_t) \right) dt \tag{76}$$

**Proof of Proposition 4.3:**

*Proof.* First we define

$$\delta_x(y^{-i}) = \prod_{j \in \{1,2,\dots,i-1,i+1,\dots L\}} \delta_{x^j}(y^j). \tag{77}$$

From the definition of entropy,

$$\frac{\partial H(\bar{q}_t)}{\partial t} = -\frac{\partial}{\partial t} \sum_{x_t} \bar{q}_t(x_t) \log \bar{q}_t(x_t) = -\sum_{x_t} \frac{\partial \bar{q}_t(x_t)}{\partial t} \left(\log \bar{q}_t(x_t) + 1\right) = \sum_{x_t} \frac{\partial \bar{q}_t(x_t)}{\partial t} \left(\log \frac{1}{\bar{q}_t(x_t)} - 1\right). \tag{78}$$

From the Continuity Equation (Gat et al., 2024),

$$\frac{\partial \bar{q}_t(x_t)}{\partial t} = \sum_{x} \bar{q}_t(x) \sum_{i=1}^{L} \delta_x(x_t^{-i}) w_t^i(x_t^i, x), \tag{79}$$

we get that

$$\frac{\partial H(\bar{q}_t)}{\partial t} = \sum_{x_t} \frac{\partial \bar{q}_t(x_t)}{\partial t} \left(\log \frac{1}{\bar{q}_t(x_t)} - 1\right) = \sum_{x_t} \sum_{x} \bar{q}_t(x) \sum_{i=1}^{L} \delta_x(x_t^{-i}) w_t^i(x_t^i, x) \left(\log \frac{1}{\bar{q}_t(x_t)} - 1\right). \tag{80}$$

We note that

$$\frac{\partial H(\bar{q}_t)}{\partial t} = \sum_{x_t} \sum_{x} \bar{q}_t(x) \sum_{i=1}^{L} \delta_x(x_t^{-i}) w_t^i(x_t^i, x) \left(\log \frac{1}{\bar{q}_t(x_t)} - 1\right) \tag{81}$$

$$= \sum_{x_t} \sum_{x} \bar{q}_t(x) \sum_{i=1}^{L} \delta_x(x_t^{-i}) w_t^i(x_t^i, x) \left(\log \frac{\bar{q}_t(x)}{\bar{q}_t(x_t)} - 1\right) - \sum_{x_t} \sum_{x} \bar{q}_t(x) \sum_{i=1}^{L} \delta_x(x_t^{-i}) w_t^i(x_t^i, x) \log \bar{q}_t(x). \tag{82}$$

The last term is 0. Indeed,

$$\sum_{x} \bar{q}_t(x) \sum_{i=1}^{L} \sum_{x_t} \left(\delta_x(x_t^{-i}) w_t^i(x_t^i, x)\right) \log \bar{q}_t(x) = \sum_{x} \bar{q}_t(x) \sum_{i=1}^{L} 0 \log \bar{q}_t(x) = 0, \tag{83}$$

thus

$$\frac{\partial H(\bar{q}_t)}{\partial t} = \sum_{x_t} \bar{q}_t(x_t) \sum_{i=1}^{L} \sum_{x^i} w_t^i(x_t^i, x) \frac{\bar{q}_t(x)}{\bar{q}_t(x_t)} \left(\log \frac{\bar{q}_t(x)}{\bar{q}_t(x_t)} - 1\right). \tag{84}$$

Integrating from time 0 to 1 on both sides, we get

$$H(\bar{q}_1) = H(\bar{q}_0) + \int_0^1 \sum_{x_t} \bar{q}_t(x_t) \sum_{i=1}^{L} \sum_{x^i} w_t^i(x_t^i, x) \frac{\bar{q}_t(x)}{\bar{q}_t(x_t)} \left(\log \frac{\bar{q}_t(x)}{\bar{q}_t(x_t)} - 1\right) dt. \tag{85}$$

$\square$

**Proof of Proposition 4.4:**

Using the same strategy as in Proposition A.1, we can rewrite the inequality in Theorem 4.2, as

$$D_{KL}(\bar{q}_1 \| \bar{p}_1) \le D_{KL}(\bar{q}_0 \| \bar{p}_0) + \int_0^1 \sum_{x_t} \bar{q}_t(x_t) \sum_{i=1}^{L} \sum_{x^i \neq x_t^i} \left(r_{\bar{q}_t} \tilde{w}_t^i(x_t^i, x) \log \frac{r_{\bar{q}_t} \tilde{w}_t^i(x_t^i, x)}{v_t^i(x^i, x_t)} + v_t^i(x^i, x_t) - r_{\bar{q}_t} \tilde{w}_t^i(x_t^i, x)\right) dt. \tag{86}$$

where $r_{\bar{q}_t} = r_{\bar{q}_t}(x, x_t) = \frac{\bar{q}_t(x)}{\bar{q}_t(x_t)}$. Expression

$$\int_0^1 \sum_{x_t} \bar{q}_t(x_t) \sum_{i=1}^{L} \sum_{x^i \neq x_t^i} \left(r_{\bar{q}_t} \tilde{w}_t^i(x_t^i, x) \log \frac{r_{\bar{q}_t} \tilde{w}_t^i(x_t^i, x)}{v_t^i(x^i, x_t)} + v_t^i(x^i, x_t) - r_{\bar{q}_t} \tilde{w}_t^i(x_t^i, x)\right) dt \tag{87}$$

can be rewritten as

$$\int_0^1 \sum_{x_t} \bar{q}_t(x_t) \sum_{i=1}^{L} \sum_{x^i \neq x_t^i} \left(r_{\bar{q}_t} \tilde{w}_t^i(x_t^i, x) \log r_{\bar{q}_t} - r_{\bar{q}_t} \tilde{w}_t^i(x_t^i, x)\right) dt \tag{88}$$

$$+\int_0^1 \sum_{x_t} \bar{q}_t(x_t) \sum_{i=1}^L \sum_{x^i \neq x_t^i} \left( r_{\bar{q}_t} \tilde{w}_t^i(x_t^i, x) \log \frac{\tilde{w}_t^i(x_t^i, x)}{v_t^i(x^i, x_t)} + v_t^i(x^i, x_t) \right) dt \tag{89}$$

and therefore as

$$\int_0^1 \sum_{x_t} \bar{q}_t(x_t) \sum_{i=1}^L \sum_{x^i} \left( r_{\bar{q}_t} \tilde{w}_t^i(x_t^i, x) \log r_{\bar{q}_t} - r_{\bar{q}_t} \tilde{w}_t^i(x_t^i, x) \right) dt + \int_0^1 \sum_{x_t} \sum_{i=1}^L \bar{q}_t(x_t) \tilde{w}_t^i(x_t^i, x_t) dt \tag{90}$$

$$+\int_0^1 \sum_{x_t} \bar{q}_t(x_t) \sum_{i=1}^L \sum_{x^i \neq x_t^i} \left( r_{\bar{q}_t} \tilde{w}_t^i(x_t^i, x) \log \frac{\tilde{w}_t^i(x_t^i, x)}{v_t^i(x^i, x_t)} + v_t^i(x^i, x_t) \right) dt. \tag{91}$$

Therefore, the initial Inequality (86), can be rewritten as

$$H(\bar{q}_1, \bar{p}_1) - H(\bar{q}_1) \leq \int_0^1 \sum_{x_t} \bar{q}_t(x_t) \sum_{i=1}^L \sum_{x^i} \left( r_{\bar{q}_t} \tilde{w}_t^i(x_t^i, x) \log r_{\bar{q}_t} - r_{\bar{q}_t} \tilde{w}_t^i(x_t^i, x) \right) dt + \int_0^1 \sum_{x_t} \sum_{i=1}^L \bar{q}_t(x_t) \tilde{w}_t^i(x_t^i, x_t) dt \tag{92}$$

$$+\int_0^1 \sum_{x_t} \bar{q}_t(x_t) \sum_{i=1}^L \sum_{x^i \neq x_t^i} \left( r_{\bar{q}_t} \tilde{w}_t^i(x_t^i, x) \log \frac{\tilde{w}_t^i(x_t^i, x)}{v_t^i(x^i, x_t)} + v_t^i(x^i, x_t) \right) dt + D_{KL}(\bar{q}_0 \| \bar{p}_0). \tag{93}$$

and since $\tilde{w}_t^i(x_t^i, x)$ denotes the reverse probability velocity, then

$$H(\bar{q}_0) = H(\bar{q}_1) + \int_0^1 \sum_{x_t} \bar{q}_t(x_t) \sum_{i=1}^L \sum_{x^i} \left( r_{\bar{q}_t} \tilde{w}_t^i(x_t^i, x) \log r_{\bar{q}_t} - r_{\bar{q}_t} \tilde{w}_t^i(x_t^i, x) \right) dt, \tag{94}$$

and therefore we can calculate the cross entropy as follows

$$H(\bar{q}_1, \bar{p}_1) \leq H(\bar{q}_0) - \int_0^1 \sum_{x_t} \bar{q}_t(x_t) \sum_{i=1}^L \sum_{x^i \neq x_t^i} \tilde{w}_t^i(x^i, x_t) dt + D_{KL}(\bar{q}_0 \| \bar{p}_0) \tag{95}$$

$$+\int_0^1 \sum_{x_0, x_1} \pi(x_0, x_1) \sum_{x_t} \bar{q}_t(x_t | x_0, x_1) \sum_{i=1}^L \sum_{x^i \neq x_t^i} \left( \frac{\bar{q}_t(x | x_0, x_1)}{\bar{q}_t(x_t | x_0, x_1)} \tilde{w}_t^i(x_t^i, x) \log \frac{\tilde{w}_t^i(x_t^i, x)}{v_t^i(x^i, x_t)} + v_t^i(x^i, x_t) \right) dt. \tag{96}$$

## A.2. First Upper Bound Derivation Details

From

$$-\log p_t(x_1; \theta) \leq \int_0^1 \sum_{x_t} p_{t|1}(x_t | x_1) \sum_{i=1}^L \sum_{x^i \neq x_t^i} \left( w_t^i(x^i, x_t) \log \frac{w_t^i(x^i, x_t)}{u_t^i(x^i, x_t; \theta)} + u_t^i(x^i, x_t; \theta) - w_t^i(x^i, x_t) \right) dt, \tag{97}$$

in the case of the special discrete flow matching dynamics from Equation (4), the probability velocity for $\bar{p}_t(x) = p_t(x; \theta)$ is given in Equation (5), with the learned velocity being $u_t^i(x^i, x_t; \theta) = \frac{\dot{k}_t}{1-k_t}\left[ p_{1|t}(x^i | x_t; \theta) - \delta_{x_t}(x^i) \right]$. The probability velocity for $\bar{q}_t(x) = p_{t|1}(x | x_1)$ can be calculated by first calculating $p_{t|1}^i(x | x_1)$ using $p_{t|1,0}^i(x | x_1, x_0)$ in Equation (4), and finding its probability velocity. However this is not necessary because we notice that for $p_{t|1,0}^i(x^i | x_1, x_0) = (1 - k_t)\delta_{x_0^i}(x^i) + k_t \delta_{x_1^i}(x^i)$ the corresponding probability velocity is $u_t^i(x^i, x_t, | x_0, x_1) = \frac{\dot{k}_t}{1-k_t}[\delta_{x_1^i}(x^i) - \delta_{x_t^i}(x^i)]$ which does not depend on $x_0$, thus $w_t^i(x^i, x_t) = u_t^i(x^i, x_t, |x_1) = u_t^i(x^i, x_t, |x_0, x_1) = \frac{\dot{k}_t}{1-k_t}[\delta_{x_1^i}(x^i) - \delta_{x_t^i}(x^i)]$. Plugging everything into Expression (97), we get that

$$-\log p_t(x_1; \theta) \leq \int_0^1 \frac{\dot{k}_t}{1-k_t} \sum_{x_t} p_{t|1}(x_t | x_1) \sum_{i=1}^L \left( -\delta_{x_1^i \neq x_t^i} \log p_{1|t}^i(x_1^i | x_t; \theta) + 1 - p_{1|t}^i(x_t^i | x_t; \theta) - \delta_{x_1^i \neq x_t^i} \right) dt. \tag{98}$$

Therefore, taking the expectation with respect to $p_1(x_1)$, we find that

$$H(p_1, p_1(\theta)) \leq \int_0^1 \frac{\dot{k}_t}{1-k_t} \sum_{x_t, x_1} p_{t,1}(x_t, x_1) \sum_{i=1}^{L} \left( -\delta_{x_1^i \neq x_t^i} \log p_{1|t}^i(x_1^i | x_t; \theta) + 1 - p_{1|t}^i(x_t^i | x_t; \theta) - \delta_{x_1^i \neq x_t^i} \right) dt. \quad (99)$$

Finally, since the part inside the large brackets is not dependent on $x_0$, we can write

$$H(p_1, p_1(\theta)) \leq \mathcal{B} \quad (100)$$

where

$$\mathcal{B} := \int_0^1 \frac{\dot{k}_t}{1-k_t} \sum_{x_1, x_0} \pi(x_1, x_0) \sum_{x_t} p_{t|1,0}(x_t | x_1, x_0) \sum_{i=1}^{L} \left( -\delta_{x_1^i \neq x_t^i} \log p_{1|t}^i(x_1^i | x_t; \theta) + 1 - p_{1|t}^i(x_t^i | x_t; \theta) - \delta_{x_1^i \neq x_t^i} \right) dt, \quad (101)$$

hence $e^{\frac{\mathcal{B}}{L}}$ is a computable upper bound of the perplexity in practice, as described in Algorithm 2.

## A.3. Alternative Upper Bound Derivation Details

From

$$H(\bar{q}_1, \bar{p}_1) \leq H(\bar{q}_0) - \sum_{x_t} \bar{q}_t(x_t) \sum_{i=1}^{L} \sum_{x^i \neq x_t^i} \tilde{w}_t^i(x^i, x_t) + D_{KL}(\bar{q}_0 \| \bar{p}_0)$$

$$+ \int_0^1 \sum_{x_0, x_1} \pi(x_0, x_1) \sum_{x_t} \bar{q}_t(x_t | x_0, x_1) \sum_{i=1}^{L} \sum_{x^i \neq x_t^i} \left( \frac{\bar{q}_t^i(x^i | x_0, x_1)}{\bar{q}_t^i(x_t^i | x_0, x_1)} \tilde{w}_t^i(x_t^i, x) \log \frac{\tilde{w}_t^i(x_t^i, x)}{v_t^i(x^i, x_t)} + v_t^i(x^i, x_t) \right) dt \quad (102)$$

by choosing $\bar{q}_t(x)$ to be the flow $p_t$ defined in Equation (2) with the coupling distribution $\pi(x_0, x_1)$, and defining $\bar{p}_t(x)$ to be the learned approximation of this flow $\bar{p}_t(\theta)$ we have

$$H(p_1, p_1(\theta)) \leq H(p_0) - \int_0^1 \sum_{x_t} p_t(x_t) \sum_{i=1}^{L} \sum_{x^i \neq x_t^i} \tilde{w}_t^i(x^i, x_t) dt$$

$$+ \int_0^1 \sum_{x_0, x_1} \pi(x_0, x_1) \sum_{x_t} p_t(x_t | x_0, x_1) \sum_{i=1}^{L} \sum_{x^i \neq x_t^i} \left( \frac{p_t^i(x^i | x_0, x_1)}{p_t^i(x_t^i | x_0, x_1)} \tilde{w}_t^i(x_t^i, x) \log \frac{\tilde{w}_t^i(x_t^i, x)}{u_t^i(x^i, x_t; \theta)} + u_t^i(x^i, x_t; \theta) \right) dt. \quad (103)$$

which can be interpreted as the discrete flow counterpart of the bound established for discrete diffusion models in Haxholli et al. (2025).

### A.3.1. MASKED SOURCE SPECIAL CASE

As shown in Gat et al. (2024), the backward probability velocity $\tilde{w}_t$, can be explicitly computed in some important special cases. For example, if coupling distribution is independent $\pi(x_0, x_1) = p_0(x_0)q_1(x_1)$, and if the source distribution is either the masked distribution, or its dimensions are i.i.d. $p_0(x_0) = \prod_{i=1}^{N} p_0(x_0^i)$. In these cases,

$$\tilde{w}_t(x^i, x_t) = -\frac{\dot{k}_t}{k_t} \left[ \delta_{x_t^i}(x^i) - p_0^i(x^i) \right]. \quad (104)$$

For the special masked dynamics corresponding to the backward probability velocity $\tilde{w}_t$ in Equation 18, we have the following inequality:

$$H(p_1, p_1(\theta)) \leq \mathcal{B} := \int_0^1 \sum_{x_0, x_1} \pi(x_0, x_1) \sum_{x_t} p_t(x_t | x_0, x_1) \sum_{i=1}^{L} \left( \frac{\dot{k}_t}{1-k_t} (1 - p_{1|t}^i(x_t^i, x_t; \theta)) \right.$$

$$\left. -\frac{\dot{k}_t}{k_t} (1 - \delta_m(x_t^i)) - \delta_m(x_t^i) \frac{\dot{k}_t}{1-k_t} \log \left( \frac{k_t}{1-k_t} (1 - p_{1|t}^i(x_1^i, x_t; \theta)) \right) \right) dt. \quad (105)$$

Indeed, the entropy of the source distribution $H(p_0)$ is 0, as all the mass is concentrated in the masked state. The term

$$\sum_{x^i \neq x_t^i} \tilde{w}_t^i(x^i, x_t) \tag{106}$$

on the other hand can be written as

$$\sum_{x^i \neq x_t^i} \frac{\dot{k}_t}{k_t} \left[ p_0^i(x^i) - \delta_{x_t^i}(x^i) \right] \tag{107}$$

and since $p_0(x^i) = \delta_m(x^i)$ we can discern two cases:

1) $x_t^i \neq m$ implying

$$\sum_{x^i \neq x_t^i} \frac{\dot{k}_t}{k_t} \left[ \delta_m(x^i) - \delta_{x_t^i}(x^i) \right] = \frac{\dot{k}_t}{k_t} \tag{108}$$

2) $x_t^i = m$ implying $x^i \neq m$ and thus

$$\sum_{x^i \neq x_t^i} \frac{\dot{k}_t}{k_t} \left[ \delta_m(x^i) - \delta_{x_t^i}(x^i) \right] = 0. \tag{109}$$

Therefore

$$-\int_0^1 \sum_{x_t} p_t(x_t) \sum_{i=1}^L \sum_{x^i \neq x_t^i} \tilde{w}_t^i(x^i, x_t) dt = -\int_0^1 \sum_{x_t} p_t(x_t) \sum_{i=1}^L \frac{\dot{k}_t}{k_t} (1 - \delta_m(x_t^i)) dt. \tag{110}$$

The following two terms remain:

$$\int_0^1 \sum_{x_0, x_1} \pi(x_0, x_1) \sum_{x_t} p_t(x_t | x_0, x_1) \sum_{i=1}^L \sum_{x^i \neq x_t^i} \left( \frac{p_t^i(x^i | x_0, x_1)}{p_t^i(x_t^i | x_0, x_1)} \tilde{w}_t^i(x_t^i, x) \log \frac{\tilde{w}_t^i(x_t^i, x)}{u_t^i(x^i, x_t; \theta)} \right) dt \tag{111}$$

and

$$\int_0^1 \sum_{x_0, x_1} \pi(x_0, x_1) \sum_{x_t} p_t(x_t | x_0, x_1) \sum_{i=1}^L \sum_{x^i \neq x_t^i} u_t^i(x^i, x_t; \theta) dt. \tag{112}$$

The last part of the first one

$$\sum_{x^i \neq x_t^i} \left( \frac{p_t^i(x^i | x_0, x_1)}{p_t^i(x_t^i | x_0, x_1)} \tilde{w}_t^i(x_t^i, x) \log \frac{\tilde{w}_t^i(x_t^i, x)}{u_t^i(x^i, x_t; \theta)} \right) dt \tag{113}$$

can be rewritten as follows:

$$\sum_{x^i \neq x_t^i} \frac{(1 - k_t) \delta_{x_0^i}(x^i) + k_t \delta_{x_1^i}(x^i)}{(1 - k_t) \delta_{x_0^i}(x_t^i) + k_t \delta_{x_1^i}(x_t^i)} \frac{\dot{k}_t}{k_t} \left[ \delta_m(x_t^i) - \delta_{x^i}(x_t^i) \right] \log \frac{\frac{\dot{k}_t}{k_t} \left[ \delta_m(x_t^i) - \delta_{x^i}(x_t^i) \right]}{\frac{k_t}{1 - k_t} \left[ p_{1|t}^i(x^i, x_t; \theta) - \delta_{x_t^i}(x^i) \right]}. \tag{114}$$

As before we can distinguish two cases:

1) $x_t^i \neq m$, which combined with $x_0^i = m$ and $x_1^i \neq m$ gives

$$\sum_{x^i \neq x_t^i} \frac{(1 - k_t) \delta_{x_0^i}(x^i) + k_t \delta_{x_1^i}(x^i)}{(1 - k_t) \delta_{x_0^i}(x_t^i) + k_t \delta_{x_1^i}(x_t^i)} \frac{\dot{k}_t}{k_t} \left[ \delta_m(x_t^i) - \delta_{x^i}(x_t^i) \right] \log \frac{\frac{\dot{k}_t}{k_t} \left[ \delta_m(x_t^i) - \delta_{x^i}(x_t^i) \right]}{\frac{k_t}{1 - k_t} \left[ p_{1|t}^i(x^i, x_t; \theta) - \delta_{x_t^i}(x^i) \right]}$$

$$= \sum_{x^i \neq x_t^i} \left( \frac{(1 - k_t) \delta_{x_0^i}(x_t^i)}{k_t \delta_{x_1^i}(x_t^i)} + 1 \right) \frac{\dot{k}_t}{k_t} [0 - 0] \log \frac{\frac{\dot{k}_t}{k_t} \left[ \delta_m(x_t^i) - \delta_{x^i}(x_t^i) \right]}{\frac{k_t}{1 - k_t} \left[ p_{1|t}^i(x^i, x_t; \theta) - \delta_{x_t^i}(x^i) \right]} = 0. \tag{115}$$

2) $x_t^i = m$ implying $x^i \neq m$, which combined with $x_0^i = m$ and $x_1^i \neq m$ gives

$$\sum_{x^i \neq x_t^i} \frac{(1-k_t)\delta_{x_0^i}(x^i) + k_t\delta_{x_1^i}(x^i)}{(1-k_t)\delta_{x_0^i}(x_t^i) + k_t\delta_{x_1^i}(x_t^i)} \frac{\dot{k}_t}{k_t} \left[\delta_m(x_t^i) - \delta_{x^i}(x_t^i)\right] \log \frac{\frac{\dot{k}_t}{k_t}\left[\delta_m(x_t^i) - \delta_{x^i}(x_t^i)\right]}{\frac{\dot{k}_t}{1-k_t}\left[p_{1|t}^i(x^i, x_t; \theta) - \delta_{x_t^i}(x^i)\right]}$$

$$= \sum_{x^i \neq x_t^i} \frac{k_t\delta_{x_1^i}(x^i)}{1-k_t} \frac{\dot{k}_t}{k_t} \log \frac{\frac{\dot{k}_t}{k_t}}{\frac{\dot{k}_t}{1-k_t}p_{1|t}(x^i|x_t)} = \sum_{x^i \neq x_t^i} \frac{\dot{k}_t}{1-k_t}\delta_{x_1^i}(x^i) \log \frac{1-k_t}{k_t p_{1|t}(x^i, x_t; \theta)}$$

$$= -\frac{\dot{k}_t}{1-k_t} \log \frac{k_t}{1-k_t} p_{1|t}^i(x_1^i, x_t; \theta). \tag{116}$$

Combining these two cases, one concludes that

$$\int_0^1 \sum_{x_0, x_1} \pi(x_0, x_1) \sum_{x_t} p_t(x_t|x_0, x_1) \sum_{i=1}^L \sum_{x^i \neq x_t^i} \left(\frac{p_t^i(x^i|x_0, x_1)}{p_t^i(x_t^i|x_0, x_1)}\tilde{w}_t^i(x_t^i, x) \log \frac{\tilde{w}_t^i(x_t^i, x)}{u_t^i(x^i, x_t; \theta)}\right) dt$$

$$= -\int_0^1 \sum_{x_0, x_1} \pi(x_0, x_1) \sum_{x_t} p_t(x_t|x_0, x_1) \sum_{i=1}^L \delta_m(x_t^i)\frac{\dot{k}_t}{1-k_t} \log \frac{k_t}{1-k_t} p_{1|t}^i(x_1^i, x_t; \theta). \tag{117}$$

Finally, we derive the last term

$$\int_0^1 \sum_{x_0, x_1} \pi(x_0, x_1) \sum_{x_t} p_t(x_t|x_0, x_1) \sum_{i=1}^L \sum_{x^i \neq x_t^i} u_t^i(x^i, x_t; \theta). \tag{118}$$

As before

$$\sum_{x^i \neq x_t^i} u_t^i(x^i, x_t; \theta) = \frac{\dot{k}_t}{1-k_t} \sum_{x^i \neq x_t^i} \left[p_{1|t}^i(x^i, x_t; \theta) - \delta_{x_t^i}(x^i)\right]$$

where

$$\frac{\dot{k}_t}{1-k_t} \sum_{x^i \neq x_t^i} p_{1|t}^i(x^i, x_t; \theta) = \frac{\dot{k}_t}{1-k_t} \left(1 - p_{1|t}^i(x_t^i, x_t; \theta)\right).$$

Putting everything together we conclude that

$$H(p_1, p_1(\theta)) \leq \mathcal{B} = \int_0^1 \sum_{x_0, x_1} \pi(x_0, x_1) \sum_{x_t} p_t(x_t|x_0, x_1) \sum_{i=1}^L \left(-\frac{\dot{k}_t}{k_t}(1 - \delta_m(x_t^i)) - \delta_m(x_t^i)\frac{\dot{k}_t}{1-k_t} \log \frac{k_t}{1-k_t}\right.$$

$$\left. + \frac{\dot{k}_t}{1-k_t}(1 - p_{1|t}^i(x_t^i, x_t; \theta)) - \delta_m(x_t^i)\frac{\dot{k}_t}{1-k_t} \log p_{1|t}^i(x_1^i, x_t; \theta)dt\right). \tag{119}$$

We can go a step further and calculate the expression

$$\int_0^1 \sum_{x_0, x_1} \pi(x_0, x_1) \sum_{x_t} p_t(x_t|x_0, x_1) \sum_{i=1}^L \left(-\frac{\dot{k}_t}{k_t}(1 - \delta_m(x_t^i)) - \delta_m(x_t^i)\frac{\dot{k}_t}{1-k_t} \log \frac{k_t}{1-k_t}\right) dt$$

$$= \int_0^1 \sum_{x_0, x_1} \pi(x_0, x_1) \sum_{i=1}^L \sum_{x_t} p_t(x_t|x_0, x_1) \left(-\frac{\dot{k}_t}{k_t}(1 - \delta_m(x_t^i)) - \delta_m(x_t^i)\frac{\dot{k}_t}{1-k_t} \log \frac{k_t}{1-k_t}\right) dt$$

$$= L\int_0^1 \left(-\dot{k}_t - \dot{k}_t \log \frac{k_t}{1-k_t}\right) dt = -L\int_0^1 \dot{k}_t dt$$

$$= -\int_0^1 \sum_{x_0, x_1} \pi(x_0, x_1) \sum_{x_t} p_t(x_t|x_0, x_1) \sum_{i=1}^L \frac{\dot{k}_t}{1-k_t}\delta_m(x_t^i).$$

Plugging this into $\mathcal{B}$, we get:

$$\mathcal{B} = \int_0^1 \sum_{x_0,x_1} \pi(x_0,x_1) \sum_{x_t} p_t(x_t|x_0,x_1) \sum_{i=1}^L \left( -\frac{\dot{k}_t}{1-k_t} \delta_m(x_t^i) + \right.$$

$$\left. \frac{\dot{k}_t}{1-k_t}(1 - p_{1|t}^i(x_t^i, x_t; \theta)) - \delta_m(x_t^i) \frac{\dot{k}_t}{1-k_t} \log p_{1|t}^i(x_1^i, x_t; \theta) dt \right). \tag{120}$$

In this special dynamic of the masked flow, $x_t^i = m$ is equivalent to $x_t^i \neq x^i$, therefore the bound above matches the first bound:

$$\mathcal{B} = \int_0^1 \frac{\dot{k}_t}{1-k_t} \sum_{x_0,x_1} \pi(x_0,x_1) \sum_{x_t} p_t(x_t|x_0,x_1) \sum_{i=1}^L \left( -\delta_{x_t^i \neq x^i} + (1 - p_{1|t}^i(x_t^i, x_t; \theta)) - \delta_{x_t^i \neq x^i} \log p_{1|t}^i(x_1^i, x_t; \theta) dt \right). \tag{121}$$

### A.4. MD4 Special Case

We can define our model $p_{1|t}(x^i, x_t; \theta)$ in the previous subsection to be such that if a given position has been unmasked we always predict that unmasked token. This implies that $p_{1|t}(x_t^i, x_t; \theta) = 1$ when $x_t^i \neq m$. This implies that Equation (120) becomes

$$\mathcal{B} = \int_0^1 \frac{\dot{k}_t \, dt}{1-k_t} \sum_{x_0,x_1} \pi(x_0,x_1) \sum_{x_t} p_t(x_t|x_0,x_1) \sum_{i=1}^L \left( -\delta_m(x_t^i) + \delta_m(x_t^i)(1 - p_{1|t}^i(x_t^i, x_t; \theta)) - \delta_m(x_t^i) \log p_{1|t}^i(x_1^i, x_t; \theta) \right) \tag{122}$$

that is

$$\mathcal{B} = \int_0^1 \frac{\dot{k}_t}{1-k_t} \sum_{x_0,x_1} \pi(x_0,x_1) \sum_{x_t} p_t^i(x_t|x_0,x_1) \sum_{i=1}^L \left( -\delta_m(x_t^i) p_{1|t}^i(x_t^i, x_t; \theta) - \delta_m(x_t^i) \log p_{1|t}^i(x_1^i, x_t; \theta) \right) dt. \tag{123}$$

However, we can set the probability of $p_{1|t}(m, x_t; \theta)$ to zero, as we know that there are no masked tokens in the data distribution, which implies,

$$\mathcal{B} = \int_0^1 \frac{\dot{k}_t}{1-k_t} \sum_{x_0,x_1} \pi(x_0,x_1) \sum_{x_t} p_t(x_t|x_0,x_1) \sum_{i=1}^L \left( -\delta_m(x_t^i) \log p_{1|t}^i(x_1^i, x_t; \theta) \right). \tag{124}$$

The final bound was originally derived in Shaul et al. (2025) and is simply MD4 from Shi et al. (2024).

### A.5. The Precise Perplexity

By using a similar strategy as in the proof of Proposition 4.4, we get

**Theorem A.2.** *Let $\pi(x_0, x_1)$ be the joint distribution of $x_0$ and $x_1$, and let $p_t$ be a flow defined as in Equations (2, 3) that transforms $p_0 = p = \sum_{x_1} \pi(x_0, x_1)$ into $q = \sum_{x_0} \pi(x_0, x_1)$. In this setting, the cross entropy $H(p_1, p_1(\theta))$ between the learned distribution and the real distribution can be written as*

$$H(p_1, p_1(\theta)) = H(p_0) + \int_0^1 \mathbb{E}_{p_{0,1}(x_0,x_1)} \mathbb{E}_{p_{t|0,1}(x_t|x_0,x_1)} \sum_{i=1}^L \sum_{x^i \neq x_t^i} u_t^i(x^i, x_t|x_1) dt$$

$$+ \int_0^1 \mathbb{E}_{p_{0,1}(x_0,x_1)} \mathbb{E}_{p_{t|0,1}(x_t|x_0,x_1)} \sum_{i=1}^L \sum_{x^i \neq x_t^i} \left( u_t^i(x^i, x_t|x_1) \log \frac{u_t^i(x^i, x_t|x_1)}{u_t^i(x^i, x_t; \theta)} + u_t^i(x^i, x_t; \theta) - u_t^i(x^i, x_t|x_1) \right) dt -$$

$$\int_0^1 \mathbb{E}_{p_{0,1}(x_0,x_1)} \mathbb{E}_{p_{t|0,1}(x_t|x_0,x_1)} \sum_{i=1}^L \sum_{x^i \neq x_t^i} \left( u_t^i(x^i, x_t|x_1) \log \frac{u_t^i(x^i, x_t|x_1)}{\frac{p_t^\theta(x_t)}{p_t^\theta(x)} u_t^i(x^i, x_t; \theta)} + \frac{p_{t|0,1}^i(x^i|x_0, x_1)}{p_{t|0,1}^i(x_t^i|x_0, x_1)} \frac{p_t^\theta(x_t)}{p_t^\theta(x)} u_t^i(x^i, x_t; \theta) \right) dt. \tag{125}$$

A proof is provided in Appendix A.8. The only terms above that we do not have an explicit form of are the learned-probability ratios between neighbor states. These are the terms missing if we tried to directly calculate the log-probability at a point using the continuity equation,

$$\frac{\partial \log p_t^\theta(x_1)}{\partial t} = \frac{1}{p_t^\theta(x_1)}\frac{\partial p_t^\theta(x_1)}{\partial t} = \sum_x \frac{p_t^\theta(x)}{p_t^\theta(x_1)} \sum_{i=1}^L \delta_x(x_1^{-i})u_t^i(x_1^i, x; \theta). \tag{126}$$

It can be shown as in Haxholli et al. (2025); Lou et al. (2024) that

$$\frac{p_t(x)}{p_t(z)} = \sum_{x_1} \frac{p_{t|1}(x|x_1)}{p_{t|1}(z|x_1)} p_{1|t}(x_1|z) \tag{127}$$

and under the conditions above $p_{t|1}(x|x_1) = \prod_{i=1}^L p_{t|1}^i(x^i|x_1^i)$, thus

$$\frac{p_t(x)}{p_t(z)} = \sum_{x_1^j} \frac{p_{t|1}^j(x^j|x_1^j)}{p_{t|1}^j(z^j|x_1^j)} p_{1|t}^j(x_1^j|z). \tag{128}$$

Since, $p_{t|1}^j(z^j|x_1^j) = \sum_{x_0^j} p_{t|1}^j(z^j|x_1^j, x_0^j)p_0(x_0^j) = (1-k_t)p_0(z^j) + k_t\delta_{x_1^j}(z^j)$ one can use the approximation:

$$\frac{p_t^\theta(x)}{p_t^\theta(z)} \approx \left(\frac{p_t(x)}{p_t(z)}\right)_\theta := \sum_{x_1^j \in \mathcal{V}} \frac{(1-k_t)p_0(x^j) + k_t\delta_{x_1^j}(x^j)}{(1-k_t)p_0(z^j) + k_t\delta_{x_1^j}(z^j)} p_{1|t}^j(x_1^j|z; \theta), \tag{129}$$

that is

$$\frac{p_t^\theta(x)}{p_t^\theta(x_t)} \approx \left(\frac{p_t(x)}{p_t(x_t)}\right)_\theta := \sum_{x_1^j \in \mathcal{V}} \frac{(1-k_t)p_0(x^i) + k_t\delta_{x_1^i}(x^i)}{(1-k_t)p_0(x_t^i) + k_t\delta_{x_1^i}(x_t^i)} p_{1|t}^j(x_1^i|x_t; \theta). \tag{130}$$

We can simplify the latter as follows:

$$\left(\frac{p_t(x)}{p_t(x_t)}\right)_\theta = \frac{p_0(x^i)}{p_0(x_t^i)} \sum_{v \in \mathcal{V}\setminus\{x^i, x_t^i\}} p_{1|t}^j(v|x_t; \theta) + \frac{(1-k_t)p_0(x^i) + k_t}{(1-k_t)p_0(x_t^i)} p_{1|t}^j(x^i|x_t; \theta) + \frac{(1-k_t)p_0(x^i)}{(1-k_t)p_0(x_t^i) + k_t} p_{1|t}^j(x_t^i|x_t; \theta). \tag{131}$$

$$\left(\frac{p_t(x)}{p_t(x_t)}\right)_\theta = \frac{p_0(x^i)}{p_0(x_t^i)} + \frac{k_t}{(1-k_t)p_0(x_t^i)} p_{1|t}^j(x^i|x_t; \theta) - \frac{\alpha_t p_0(x^i)}{((p_0(x_t^i) + \alpha_t)p_0(x_t^i)} p_{1|t}^j(x_t^i|x_t; \theta). \tag{132}$$

where $\alpha_t = \frac{k_t}{1-k_t}$.

Using this approximation in Equation 125, in the masked flow case, gives the following:

$$H(p_1, p_1(\theta)) \approx \int_0^1 \frac{\dot{k}_t}{1-k_t} \sum_{x_0, x_1} \pi(x_0, x_1) \sum_{x_t} p_t(x_t|x_0, x_1) \sum_{i=1}^L \left(-\delta_m(x_t^i)\log p_{1|t}^i(x_1^i, x_t; \theta)\right) = \mathcal{B}. \tag{133}$$

Thus the upper bound in the masked case is simply the true perplexity formula when we use the approximation above for the ratios of the learned probabilities between neighbor states.

## A.6. Time-independence of Predictive Probabilities in Multimasked Flows

The following proposition is a generalization of that given in Gat et al. (2024) for masked flows.

**Proposition A.3.** *In the case of multimasked flows, predictive probabilities $p_{1|t}(x^i \mid x_t)$ are time independent.*

*Proof.* From

$$p_{t|0,1}^i(z^i|x_0, x_1) = (1-k_t)\delta_{x_0^i}(z^i) + k_t\delta_{x_1^i}(z^i) = \begin{cases} (1-k_t)\delta_{x_0^i}(z^i), & z^i > V_d, \\ k_t\delta_{x_1^i}(z^i), & z^i \leq V_d. \end{cases} \tag{134}$$

we have that

$$p_{t|0,1}(z \mid x_0, x_1) = \left[\prod_{i:\, z^i > V_d} (1 - k_t)\right]\left[\prod_{i:\, z^i \le V_d} k_t\right]\left[\prod_{i:\, z^i > V_d} \delta_{x_0^i}(z^i)\right]\left[\prod_{i:\, z^i \le V_d} \delta_{x_1^i}(z^i)\right] \tag{135}$$

$$p_{0,1|t}(x_0, x_1 \mid z) = \frac{p_{t|0,1}(z \mid x_0, x_1)p(x_0, x_1)}{p_t(z)} = \frac{p_{t|0,1}(z \mid x_0, x_1)p(x_0, x_1)}{\sum_{\tilde{x}_0, \tilde{x}_1} p_{t|0,1}(z \mid \tilde{x}_0, \tilde{x}_1)p(\tilde{x}_0, \tilde{x}_1)} \tag{136}$$

$$= \frac{\left(\left[\prod_{i:\, z^i > V_d}(1 - k_t)\right]\left[\prod_{i:\, z^i \le V_d} k_t\right]\right)\left[\prod_{i:\, z^i > V_d} \delta_{x_0^i}(z^i)\right]\left[\prod_{i:\, z^i \le V_d} \delta_{x_1^i}(z^i)\right]p(x_0, x_1)}{\sum_{\tilde{x}_0, \tilde{x}_1}\left(\left[\prod_{i:\, z^i > V_d}(1 - k_t)\right]\left[\prod_{i:\, z^i \le V_d} k_t\right]\right)\left[\prod_{i:\, z^i > V_d} \delta_{\tilde{x}_0^i}(z^i)\right]\left[\prod_{i:\, z^i \le V_d} \delta_{\tilde{x}_1^i}(z^i)\right]p(\tilde{x}_0, \tilde{x}_1)} \tag{137}$$

$$= \frac{\left[\prod_{i:\, z^i > V_d} \delta_{x_0^i}(z^i)\right]\left[\prod_{i:\, z^i \le V_d} \delta_{x_1^i}(z^i)\right]p(x_0, x_1)}{\sum_{\tilde{x}_0, \tilde{x}_1}\left[\prod_{i:\, z^i > V_d} \delta_{\tilde{x}_0^i}(z^i)\right]\left[\prod_{i:\, z^i \le V_d} \delta_{\tilde{x}_1^i}(z^i)\right]p(\tilde{x}_0, \tilde{x}_1)} = p_{0,1}(x_0, x_1 \mid z) \tag{138}$$

which does not depend on $t$. Thus

$$p_{1|t}(x^i \mid z) = \sum_{x_0, x_1} \delta_{x_1}(x^i)p_{0,1|t}(x_0, x_1 \mid z) = \sum_{x_0, x_1} \delta_{x_1}(x^i)p_{0,1}(x_0, x_1 \mid z) = p_1(x^i \mid z) \tag{139}$$

does not depend on time either. $\qquad\square$

## A.7. Invariability of Unmasked Positions in Multimasked Flows

**Proposition A.4.** *For an multimasked-flow $p_t$, if position $i$ of a state $x_t$ has been unmasked, i.e., $x_t^i = v$ for some data token $v$ in the vocabulary, then $p_{1|t}^i(v|x_t) = 1$ given that $p(x_t) > 0$.*

*Proof.*

$$p(x_t|x_0, x_1) = \prod_{j=1}^{L} p^j(x_t^j|x_0, x_1) = ((1 - k_t)\delta_{x_0^i}(x_t^i) + k_t\delta_{x_1^i}(x_t^i))\prod_{j\neq i}^{L} p^j(x_t|x_0, x_1) \tag{140}$$

$$= ((1 - k_t)\delta_{x_0^i}(v) + k_t\delta_{x_1^i}(v))\prod_{j\neq i}^{L} p^j(x_t|x_0, x_1) \tag{141}$$

Thus if $x_1^i \neq v$,

$$p(x_t|x_0, x_1) = ((1 - k_t)0 + k_t 0)\prod_{j\neq i}^{L} p^j(x_t|x_0, x_1) = 0. \tag{142}$$

This implies that if $x_1^i \neq v$, then

$$p(x_0, x_1|x_t) = \frac{p(x_t|x_0, x_1)p(x_0, x_1)}{p(x_t)} = 0. \tag{143}$$

If $p(x_t) = 0$ we can define predictive probabilities arbitrarily without modifying the generated distribution.

Therefore, integrating with respect to $(x_0, x_1^{-i})$, we have that if $x_1^i \neq v$, then $p^i(x_1^i|x_t) = 0$. Since $\sum_{x^i \in \mathcal{V}_{datatokens}} p^i(x_1^i|x_t) = 1$, this implies $p^i(v|x_t) = 1$. $\qquad\square$

### A.8. Proof of Theorem A.2

First, we prove that

$$H(p_1, p_1(\theta)) = H(p_0) + \int_0^1 \mathbb{E}_{p_{0,1}(x_0,x_1)} \mathbb{E}_{p_{t|0,1}(x_t|x_0,x_1)} \sum_{i=1}^L \sum_{x^i \neq x_t^i} u_t^i(x^i, x_t|x_1) dt$$

$$+ \int_0^1 \mathbb{E}_{p_{0,1}(x_0,x_1)} \mathbb{E}_{p_{t|0,1}(x_t|x_0,x_1)} \sum_{i=1}^L \sum_{x^i \neq x_t^i} \left( u_t^i(x^i, x_t|x_1) \log \frac{u_t^i(x^i, x_t|x_1)}{u_t^i(x^i, x_t; \theta)} + u_t^i(x^i, x_t; \theta) - u_t^i(x^i, x_t|x_1) \right) dt -$$

$$\int_0^1 \mathbb{E}_{p_{0,1}(x_0,x_1)} \mathbb{E}_{p_{t|0,1}(x_t|x_0,x_1)} \sum_{i=1}^L \sum_{x^i \neq x_t^i} \left( u_t^i(x^i, x_t|x_1) \log \frac{u_t^i(x^i, x_t|x_1)}{\frac{p_t^\theta(x_t)}{p_t^\theta(x)} u_t^i(x^i, x_t; \theta)} + \frac{p_{t|0,1}^i(x^i|x_0, x_1)}{p_{t|0,1}^i(x_t^i|x_0, x_1)} \frac{p_t^\theta(x_t)}{p_t^\theta(x)} u_t^i(x^i, x_t; \theta) \right) dt.$$

$$(144)$$

We start with the Equation 66 from Proposition A.1,

$$D_{KL}(\bar{q}_1 \| \bar{p}_1) = D_{KL}(\bar{q}_0 \| \bar{p}_0) + \int_0^1 \sum_{x_t} \bar{q}_t(x_t) \sum_{i=1}^L \sum_{x^i \neq x_t^i} \left( w_t^i(x^i, x_t) \log \frac{w_t^i(x^i, x_t)}{v_t^i(x^i, x_t)} + v_t^i(x^i, x_t) - w_t^i(x^i, x_t) \right) dt$$

$$- \int_0^1 \sum_{x_t} \bar{q}_t(x_t) \sum_{i=1}^L \sum_{x^i \neq x_t^i} \left( r_{\bar{q}_t} w_t^i(x_t^i, x) \log \frac{r_{\bar{q}_t} w_t^i(x_t^i, x)}{r_{\bar{p}_t} v_t^i(x_t^i, x)} + r_{\bar{p}_t} v_t^i(x_t^i, x) - r_{\bar{q}_t} w_t^i(x_t^i, x) \right) dt. \qquad (145)$$

On the right-hand side, the first term two terms are the upper bound, while the third term is what was discarded to get the upper bound, so here we try to compute it:

$$\int_0^1 \sum_{x_t} \bar{q}_t(x_t) \sum_{i=1}^L \sum_{x^i \neq x_t^i} \left( \frac{\bar{q}_t(x)}{\bar{q}_t(x_t)} w_t^i(x_t^i, x) \log \frac{\frac{\bar{q}_t(x)}{\bar{q}_t(x_t)} w_t^i(x_t^i, x)}{\frac{\bar{p}_t(x)}{\bar{p}_t(x_t)} v_t^i(x_t^i, x)} + \frac{\bar{p}_t(x)}{\bar{p}_t(x_t)} v_t^i(x_t^i, x) - \frac{\bar{q}_t(x)}{\bar{q}_t(x_t)} w_t^i(x_t^i, x) \right) dt \qquad (146)$$

$$= \int_0^1 \sum_{x_t} \bar{q}_t(x_t) \sum_{i=1}^L \sum_{x^i \neq x_t^i} \left( \frac{\bar{q}_t(x)}{\bar{q}_t(x_t)} w_t^i(x_t^i, x) \log \frac{\bar{q}_t(x)}{\bar{q}_t(x_t)} - \frac{\bar{q}_t(x)}{\bar{q}_t(x_t)} w_t^i(x_t^i, x) \right) dt \qquad (147)$$

$$+ \int_0^1 \sum_{x_t} \bar{q}_t(x_t) \sum_{i=1}^L \sum_{x^i \neq x_t^i} \left( \frac{\bar{q}_t(x)}{\bar{q}_t(x_t)} w_t^i(x_t^i, x) \log \frac{w_t^i(x_t^i, x)}{\frac{\bar{p}_t(x)}{\bar{p}_t(x_t)} v_t^i(x_t^i, x)} + \frac{\bar{p}_t(x)}{\bar{p}_t(x_t)} v_t^i(x_t^i, x) \right) dt \qquad (148)$$

$$= \int_0^1 \sum_{x_t} \bar{q}_t(x_t) \sum_{i=1}^L \sum_{x^i} \left( \frac{\bar{q}_t(x)}{\bar{q}_t(x_t)} w_t^i(x_t^i, x) \log \frac{\bar{q}_t(x)}{\bar{q}_t(x_t)} - \frac{\bar{q}_t(x)}{\bar{q}_t(x_t)} w_t^i(x_t^i, x) \right) dt \qquad (149)$$

$$- \int_0^1 \sum_{x_t} \bar{q}_t(x_t) \sum_{i=1}^L \left( \frac{\bar{q}_t(x_t)}{\bar{q}_t(x_t)} w_t^i(x_t^i, x_t) \log \frac{\bar{q}_t(x_t)}{\bar{q}_t(x_t)} - \frac{\bar{q}_t(x_t)}{\bar{q}_t(x_t)} w_t^i(x_t^i, x_t) \right) dt \qquad (150)$$

$$+ \int_0^1 \sum_{x_t} \bar{q}_t(x_t) \sum_{i=1}^L \sum_{x^i \neq x_t^i} \left( \frac{\bar{q}_t(x)}{\bar{q}_t(x_t)} w_t^i(x_t^i, x) \log \frac{w_t^i(x_t^i, x)}{\frac{\bar{p}_t(x)}{\bar{p}_t(x_t)} v_t^i(x_t^i, x)} + \frac{\bar{p}_t(x)}{\bar{p}_t(x_t)} v_t^i(x_t^i, x) \right) dt \qquad (151)$$

From Proposition 4.4 we have that:

$$H(\bar{q}_1) - H(\bar{q}_0) = \int_0^1 \sum_{x_t} \bar{q}_t(x_t) \sum_{i=1}^L \sum_{x^i} w_t^i(x_t^i, x) \frac{\bar{q}_t(x)}{\bar{q}_t(x_t)} \left( \log \frac{\bar{q}_t(x)}{\bar{q}_t(x_t)} - 1 \right) dt, \qquad (152)$$

therefore

$$\int_0^1 \sum_{x_t} \bar{q}_t(x_t) \sum_{i=1}^L \sum_{x^i \neq x_t^i} \left( \frac{\bar{q}_t(x)}{\bar{q}_t(x_t)} w_t^i(x_t^i, x) \log \frac{\frac{\bar{q}_t(x)}{\bar{q}_t(x_t)} w_t^i(x_t^i, x)}{\frac{\bar{p}_t(x)}{\bar{p}_t(x_t)} v_t^i(x_t^i, x)} + \frac{\bar{p}_t(x)}{\bar{p}_t(x_t)} v_t^i(x_t^i, x) - \frac{\bar{q}_t(x)}{\bar{q}_t(x_t)} w_t^i(x_t^i, x) \right) dt \tag{153}$$

$$= H(\bar{q}_1) - H(\bar{q}_0) - \int_0^1 \sum_{x_t} \bar{q}_t(x_t) \sum_{i=1}^L \left( - w_t^i(x_t^i, x_t) \right) dt \tag{154}$$

$$+ \int_0^1 \sum_{x_t} \bar{q}_t(x_t) \sum_{i=1}^L \sum_{x^i \neq x_t^i} \left( \frac{\bar{q}_t(x)}{\bar{q}_t(x_t)} w_t^i(x_t^i, x) \log \frac{w_t^i(x_t^i, x)}{\frac{\bar{p}_t(x)}{\bar{p}_t(x_t)} v_t^i(x_t^i, x)} + \frac{\bar{p}_t(x)}{\bar{p}_t(x_t)} v_t^i(x_t^i, x) \right) dt \tag{155}$$

$$= H(\bar{q}_1) - H(\bar{q}_0) - \int_0^1 \sum_{x_t} \bar{q}_t(x_t) \sum_{i=1}^L \sum_{x^i \neq x_t^i} w_t^i(x^i, x_t) dt \tag{156}$$

$$+ \int_0^1 \sum_{x_t} \bar{q}_t(x_t) \sum_{i=1}^L \sum_{x^i \neq x_t^i} \left( \frac{\bar{q}_t(x)}{\bar{q}_t(x_t)} w_t^i(x_t^i, x) \log \frac{w_t^i(x_t^i, x)}{\frac{\bar{p}_t(x)}{\bar{p}_t(x_t)} v_t^i(x_t^i, x)} + \frac{\bar{p}_t(x)}{\bar{p}_t(x_t)} v_t^i(x_t^i, x) \right) dt. \tag{157}$$

We choose $\bar{q}_t(x)$ to have the dynamics of the flow $p_t$, but with the coupling distribution $\bar{\pi}(x, y) = p_0(x)\delta_{x_1}(y) = \int \pi(x, z) dz \delta_{x_1}(y)$. Clearly, we have $\bar{q}_0(x) = p_0(x)$, $\bar{q}_1(x) = \delta_{x_1}(x)$, $\bar{q}_t(x) = p_{t|1}(x|x_1)$ and $w_t(x^i, x_t) = u_t(x^i, x_t|x_1) = \frac{\dot{k}_t}{1-k_t} \left( \delta_{x_1^i}(x^i) - \delta_{x_t^i}(x^i) \right)$.

On the other hand, we choose $\bar{p}_t$ to be the learned flow $p_t(\cdot; \theta)$, and therefore $\bar{p}_0(x) = p_0(x)$ and $v_t^i(x^i, x_t) = u_t(x^i, x_t; \theta) = \frac{\dot{k}_t}{1-k_t} \left( p_{1|t}(x^i, x_t; \theta) - \delta_{x_t^i}(x^i) \right)$

We notice that since $\bar{q}_0(x) = p_0(x)$ and $\bar{p}_0(x) = p_0(x)$, then $D_{KL}(\bar{q}_0 \| \bar{p}_0) = 0$. Furthermore $D_{KL}(\bar{q}_1(x) \| \bar{p}_1(x)) = D_{KL}(\delta_{x_1}(x) \| p_1(x; \theta)) = -\log p_1(x_1; \theta)$. Thus for such particular choices Equation 145 becomes

$$-\log p_1(x_1; \theta) = \int_0^1 \sum_{x_t} p_{t|1}(x_t|x_1) \sum_{i=1}^L \sum_{x^i \neq x_t^i} \left( u_t^i(x^i, x_t|x_1) \log \frac{u_t^i(x^i, x_t|x_1)}{u_t^i(x^i, x_t; \theta)} + u_t^i(x^i, x_t; \theta) - u_t^i(x^i, x_t|x_1) \right) dt -$$

$$\left[ H(\delta_{x_1}(x)) - H(p_0) - \int_0^1 \sum_{x_t} p_{t|1}(x_t|x_1) \sum_{i=1}^L \sum_{x^i \neq x_t^i} u_t^i(x^i, x_t|x_1) dt \right.$$

$$\left. + \int_0^1 \sum_{x_t} p_{t|1}(x_t|x_1) \sum_{i=1}^L \sum_{x^i \neq x_t^i} \left( \frac{p_{t|1}(x|x_1)}{p_{t|1}(x_t|x_1)} u_t^i(x_t^i, x|x_1) \log \frac{u_t^i(x_t^i, x|x_1)}{\frac{p_t^\theta(x)}{p_t^\theta(x_t)} u_t^i(x_t^i, x; \theta)} + \frac{p_t^\theta(x)}{p_t^\theta(x_t)} u_t^i(x_t^i, x; \theta) \right) dt \right], \tag{158}$$

thus

$$-\log p_1(x_1; \theta) = \int_0^1 \sum_{x_t} p_{t|1}(x_t|x_1) \sum_{i=1}^L \sum_{x^i \neq x_t^i} \left( u_t^i(x^i, x_t|x_1) \log \frac{u_t^i(x^i, x_t|x_1)}{u_t^i(x^i, x_t; \theta)} + u_t^i(x^i, x_t; \theta) - u_t^i(x^i, x_t|x_1) \right) dt$$

$$+ H(p_0) + \int_0^1 \sum_{x_t} p_{t|1}(x_t|x_1) \sum_{i=1}^L \sum_{x^i \neq x_t^i} u_t^i(x^i, x_t|x_1) dt$$

$$- \int_0^1 \sum_{x_t} p_{t|1}(x_t|x_1) \sum_{i=1}^L \sum_{x^i \neq x_t^i} \left( \frac{p_{t|1}(x|x_1)}{p_{t|1}(x_t|x_1)} u_t^i(x_t^i, x|x_1) \log \frac{u_t^i(x_t^i, x|x_1)}{\frac{p_t^\theta(x)}{p_t^\theta(x_t)} u_t^i(x_t^i, x; \theta)} + \frac{p_t^\theta(x)}{p_t^\theta(x_t)} u_t^i(x_t^i, x; \theta) \right) dt, \tag{159}$$

Now, we can take the expectation with respect to the data distribution on both sides of Equation 159:

$$H(p_1, p_1(\theta)) = \int_0^1 \sum_{x_1} \sum_{x_t} p_{t,1}(x_t, x_1) \sum_{i=1}^L \sum_{x^i \neq x_t^i} \left( u_t^i(x^i, x_t|x_1) \log \frac{u_t^i(x^i, x_t|x_1)}{u_t^i(x^i, x_t; \theta)} + u_t^i(x^i, x_t; \theta) - u_t^i(x^i, x_t|x_1) \right) dt$$

$$+ H(p_0) + \int_0^1 \sum_{x_1} \sum_{x_t} p_{t,1}(x_t, x_1) \sum_{i=1}^L \sum_{x^i \neq x_t^i} u_t^i(x^i, x_t|x_1) dt$$

$$- \int_0^1 \sum_{x_1} \sum_{x_t} p_{t,1}(x_t, x_1) \sum_{i=1}^L \sum_{x^i \neq x_t^i} \left( \frac{p_{t,1}(x, x_1)}{p_{t,1}(x_t, x_1)} u_t^i(x_t^i, x|x_1) \log \frac{u_t^i(x_t^i, x|x_1)}{\frac{p_t^\theta(x)}{p_t^\theta(x_t)} u_t^i(x_t^i, x; \theta)} + \frac{p_t^\theta(x)}{p_t^\theta(x_t)} u_t^i(x_t^i, x; \theta) \right) dt \quad (160)$$

$$= \int_0^1 \sum_{x_1} \sum_{x_t} p_{t,1}(x_t, x_1) \sum_{i=1}^L \sum_{x^i \neq x_t^i} \left( u_t^i(x^i, x_t|x_1) \log \frac{u_t^i(x^i, x_t|x_1)}{u_t^i(x^i, x_t; \theta)} + u_t^i(x^i, x_t; \theta) - u_t^i(x^i, x_t|x_1) \right) dt$$

$$+ H(p_0) + \int_0^1 \sum_{x_1} \sum_{x_t} p_{t,1}(x_t, x_1) \sum_{i=1}^L \sum_{x^i \neq x_t^i} u_t^i(x^i, x_t|x_1) dt$$

$$- \int_0^1 \sum_{x_1} \sum_{x_t} \sum_{i=1}^L \sum_{x^i \neq x_t^i} \left( \frac{p_{t,1}(x, x_1)}{1} u_t^i(x_t^i, x|x_1) \log \frac{u_t^i(x_t^i, x|x_1)}{\frac{p_t^\theta(x)}{p_t^\theta(x_t)} u_t^i(x_t^i, x; \theta)} + p_{t,1}(x_t, x_1) \frac{p_t^\theta(x)}{p_t^\theta(x_t)} u_t^i(x_t^i, x; \theta) \right) dt \quad (161)$$

$$= \int_0^1 \sum_{x_0} \sum_{x_1} \sum_{x_t} p_{t,1,0}(x_t, x_1, x_0) \sum_{i=1}^L \sum_{x^i \neq x_t^i} \left( u_t^i(x^i, x_t|x_1) \log \frac{u_t^i(x^i, x_t|x_1)}{u_t^i(x^i, x_t; \theta)} + u_t^i(x^i, x_t; \theta) - u_t^i(x^i, x_t|x_1) \right) dt$$

$$+ H(p_0) + \int_0^1 \sum_{x_0} \sum_{x_1} \sum_{x_t} p_{t,1,0}(x_t, x_1, x_0) \sum_{i=1}^L \sum_{x^i \neq x_t^i} u_t^i(x^i, x_t|x_1) dt -$$

$$\int_0^1 \sum_{x_0} \sum_{x_1} \sum_{x_t} \sum_{i=1}^L \sum_{x^i \neq x_t^i} \left( \frac{p_{t,1,0}(x, x_1, x_0)}{1} u_t^i(x_t^i, x|x_1) \log \frac{u_t^i(x_t^i, x|x_1)}{\frac{p_t^\theta(x)}{p_t^\theta(x_t)} u_t^i(x_t^i, x; \theta)} + p_{t,1,0}(x_t, x_1, x_0) \frac{p_t^\theta(x)}{p_t^\theta(x_t)} u_t^i(x_t^i, x; \theta) \right) dt,$$

$$(162)$$

$$= \int_0^1 \mathbb{E}_{p_{0,1}(x_0, x_1)} \mathbb{E}_{p_{t|0,1}(x_t|x_0, x_1)} \sum_{i=1}^L \sum_{x^i \neq x_t^i} \left( u_t^i(x^i, x_t|x_1) \log \frac{u_t^i(x^i, x_t|x_1)}{u_t^i(x^i, x_t; \theta)} + u_t^i(x^i, x_t; \theta) - u_t^i(x^i, x_t|x_1) \right) dt -$$

$$+ H(p_0) + \int_0^1 \mathbb{E}_{p_{0,1}(x_0, x_1)} \mathbb{E}_{p_{t|0,1}(x_t|x_0, x_1)} \sum_{i=1}^L \sum_{x^i \neq x_t^i} u_t^i(x^i, x_t|x_1) dt$$

$$\int_0^1 \sum_{x_0} \sum_{x_1} p_{t,1,0}(x_t, x_1, x_0) \sum_{x_t} \sum_{i=1}^L \sum_{x^i \neq x_t^i} \left( \frac{p_{t,1,0}(x, x_1, x_0)}{p_{t,1,0}(x_t, x_1, x_0)} u_t^i(x_t^i, x|x_1) \log \frac{u_t^i(x_t^i, x|x_1)}{\frac{p_t^\theta(x)}{p_t^\theta(x_t)} u_t^i(x_t^i, x; \theta)} + \frac{p_t^\theta(x)}{p_t^\theta(x_t)} u_t^i(x_t^i, x; \theta) \right) dt$$

$$(163)$$

therefore,

$$H(p_1, p_1(\theta)) = H(p_0) + \int_0^1 \mathbb{E}_{p_{0,1}(x_0, x_1)} \mathbb{E}_{p_{t|0,1}(x_t|x_0, x_1)} \sum_{i=1}^L \sum_{x^i \neq x_t^i} u_t^i(x^i, x_t|x_1) dt$$

$$+ \int_0^1 \mathbb{E}_{p_{0,1}(x_0, x_1)} \mathbb{E}_{p_{t|0,1}(x_t|x_0, x_1)} \sum_{i=1}^L \sum_{x^i \neq x_t^i} \left( u_t^i(x^i, x_t|x_1) \log \frac{u_t^i(x^i, x_t|x_1)}{u_t^i(x^i, x_t; \theta)} + u_t^i(x^i, x_t; \theta) - u_t^i(x^i, x_t|x_1) \right) dt$$

$$- \int_0^1 \mathbb{E}_{p_{0,1}(x_0, x_1)} \mathbb{E}_{p_{t|0,1}(x_t|x_0, x_1)} \sum_{i=1}^L \sum_{x^i \neq x_t^i} \left( \frac{p_{t|1,0}^i(x^i|x_1, x_0)}{p_{t|1,0}^i(x_t^i|x_1, x_0)} u_t^i(x_t^i, x|x_1) \log \frac{u_t^i(x_t^i, x|x_1)}{\frac{p_t^\theta(x)}{p_t^\theta(x_t)} u_t^i(x_t^i, x; \theta)} + \frac{p_t^\theta(x)}{p_t^\theta(x_t)} u_t^i(x_t^i, x; \theta) \right) dt$$

$$(164)$$

The only terms above for which we do not have an immediate explicit form are the learned-probability ratios between neighbor states.

Now, we can rewrite

$$H(p_1, p_1(\theta)) = H(p_0) + \int_0^1 \mathbb{E}_{p_{0,1}(x_0,x_1)} \mathbb{E}_{p_{t|0,1}(x_t|x_0,x_1)} \sum_{i=1}^L \sum_{x^i \neq x_t^i} u_t^i(x^i, x_t|x_1) dt$$

$$+ \int_0^1 \mathbb{E}_{p_{0,1}(x_0,x_1)} \mathbb{E}_{p_{t|0,1}(x_t|x_0,x_1)} \sum_{i=1}^L \sum_{x^i \neq x_t^i} \left( u_t^i(x^i, x_t|x_1) \log \frac{u_t^i(x^i, x_t|x_1)}{u_t^i(x^i, x_t; \theta)} + u_t^i(x^i, x_t; \theta) - u_t^i(x^i, x_t|x_1) \right) dt$$

$$- \int_0^1 \mathbb{E}_{p_{0,1}(x_0,x_1)} \mathbb{E}_{p_{t|0,1}(x_t|x_0,x_1)} \sum_{i=1}^L \sum_{x^i \neq x_t^i} \left( \frac{p_{t|1,0}(x|x_1,x_0)}{p_{t|1,0}(x_t|x_1,x_0)} u_t^i(x_t^i, x|x_1) \log \frac{u_t^i(x_t^i, x|x_1)}{\frac{p_t^\theta(x)}{p_t^\theta(x_t)} u_t^i(x_t^i, x; \theta)} + \frac{p_t^\theta(x)}{p_t^\theta(x_t)} u_t^i(x_t^i, x; \theta) \right) dt \tag{165}$$

to be computationally cheaper. Indeed, the last term

$$\int_0^1 \mathbb{E}_{p_{0,1}(x_0,x_1)} \mathbb{E}_{p_{t|0,1}(x_t|x_0,x_1)} \sum_{i=1}^L \sum_{x^i \neq x_t^i} \left( \frac{p_{t|1,0}(x|x_1,x_0)}{p_{t|1,0}(x_t|x_1,x_0)} u_t^i(x_t^i, x|x_1) \log \frac{u_t^i(x_t^i, x|x_1)}{\frac{p_t^\theta(x)}{p_t^\theta(x_t)} u_t^i(x_t^i, x; \theta)} + \frac{p_t^\theta(x)}{p_t^\theta(x_t)} u_t^i(x_t^i, x; \theta) \right) dt \tag{166}$$

can be rewritten as

$$\int_0^1 \mathbb{E}_{p_{0,1}(x_0,x_1)} \sum_{i=1}^L \mathbb{E}_{p_{t|0,1}(x_t|x_0,x_1)} \sum_{x^i \neq x_t^i} \left( \frac{p_{t|1,0}(x|x_1,x_0)}{p_{t|1,0}(x_t|x_1,x_0)} u_t^i(x_t^i, x|x_1) \log \frac{u_t^i(x_t^i, x|x_1)}{\frac{p_t^\theta(x)}{p_t^\theta(x_t)} u_t^i(x_t^i, x; \theta)} + \frac{p_t^\theta(x)}{p_t^\theta(x_t)} u_t^i(x_t^i, x; \theta) \right) dt, \tag{167}$$

where by changing the order of the sums,

$$\int_0^1 \mathbb{E}_{p_{0,1}(x_0,x_1)} \sum_{i=1}^L \sum_{x_t} \sum_{x} \delta_{x^{-i}}(x_t^{-i}) \delta_{x^i \neq x_t^i} \left( p_{t|1,0}(x|x_1,x_0) u_t^i(x_t^i, x|x_1) \log \frac{u_t^i(x_t^i, x|x_1)}{\frac{p_t^\theta(x)}{p_t^\theta(x_t)} u_t^i(x_t^i, x; \theta)} \right. \tag{168}$$

$$\left. + p_{t|0,1}(x_t|x_0,x_1) \frac{p_t^\theta(x)}{p_t^\theta(x_t)} u_t^i(x_t^i, x; \theta) \right) dt \tag{169}$$

to

$$\int_0^1 \mathbb{E}_{p_{0,1}(x_0,x_1)} \sum_{i=1}^L \sum_{x} \sum_{x_t^i \neq x^i} \left( p_{t|1,0}(x|x_1,x_0) u_t^i(x_t^i, x|x_1) \log \frac{u_t^i(x_t^i, x|x_1)}{\frac{p_t^\theta(x)}{p_t^\theta(x_t)} u_t^i(x_t^i, x; \theta)} \right. \tag{170}$$

$$\left. + p_{t|0,1}(x_t|x_0,x_1) \frac{p_t^\theta(x)}{p_t^\theta(x_t)} u_t^i(x_t^i, x; \theta) \right) dt \tag{171}$$

and by switching the notation between $x$ and $x_t$:

$$\int_0^1 \mathbb{E}_{p_{0,1}(x_0,x_1)} \sum_{i=1}^L \sum_{x_t} \sum_{x^i \neq x_t^i} \left( p_{t|1,0}(x_t|x_1,x_0) u_t^i(x^i, x_t|x_1) \log \frac{u_t^i(x^i, x_t|x_1)}{\frac{p_t^\theta(x_t)}{p_t^\theta(x)} u_t^i(x^i, x_t; \theta)} \right. \tag{172}$$

$$\left. + p_{t|0,1}(x|x_0,x_1) \frac{p_t^\theta(x_t)}{p_t^\theta(x)} u_t^i(x^i, x_t; \theta) \right) dt \tag{173}$$

therefore,

$$\int_0^1 \mathbb{E}_{p_{0,1}(x_0,x_1)} \mathbb{E}_{p_{t|0,1}(x_t|x_0,x_1)} \sum_{i=1}^L \sum_{x^i \neq x_t^i} \left( u_t^i(x^i, x_t|x_1) \log \frac{u_t^i(x^i, x_t|x_1)}{\frac{p_t^\theta(x_t)}{p_t^\theta(x)} u_t^i(x^i, x_t; \theta)} + \frac{p_{t|0,1}(x|x_0,x_1)}{p_{t|0,1}(x_t|x_0,x_1)} \frac{p_t^\theta(x_t)}{p_t^\theta(x)} u_t^i(x^i, x_t; \theta) \right) dt \tag{174}$$

thus finally,

$$\int_0^1 \mathbb{E}_{p_{0,1}(x_0,x_1)} \mathbb{E}_{p_{t|0,1}(x_t|x_0,x_1)} \sum_{i=1}^L \sum_{x^i \neq x_t^i} \left( u_t^i(x^i, x_t|x_1) \log \frac{u_t^i(x^i, x_t|x_1)}{\frac{p_t^\theta(x_t)}{p_t^\theta(x)} u_t^i(x^i, x_t; \theta)} + \frac{p_{t|0,1}^i(x^i|x_0,x_1)}{p_{t|0,1}^i(x_t^i|x_0,x_1)} \frac{p_t^\theta(x_t)}{p_t^\theta(x)} u_t^i(x^i, x_t; \theta) \right) dt. \tag{175}$$

# B. Algorithms

---

**Algorithm 1** Discrete Flow Matching with OT Minibatches

---

**Input:** Set of samples $\mathcal{D}$ from $\pi(x_0, x_1)$, model $p^i_{1|t}(x^i|x_t; \theta)$
**repeat**
    1) Sample minibatch $\mathcal{D}_j$ from $\mathcal{D}$.
    2) $\bar{\pi}(x, y) \leftarrow \text{OT}(\mathcal{D}_j)$, s.t. $p(x) = \sum_{y \in \mathcal{D}_j} \bar{\pi}(x, y) = \frac{1}{|\mathcal{D}_j|}$, $q(y) = \sum_{x \in \mathcal{D}_j} \bar{\pi}(x, y) = \frac{1}{|\mathcal{D}_j|}$.
    3) Sample $t$ form $U(0, 1)$.
    4) Sample $x_0, x_1$ from $\bar{\pi}(x_0, x_1)$.
    5) Sample $x_t$ using Equation (4).
    6) Calculate the gradient of the loss $\mathcal{L}$ (e.g. Expression (7))
    7) Update parameters $\theta$
**until** Convergence or stopping criterion

---

**Algorithm 2** Computing the perplexity upper bound

---

**Input:** samples from $\pi(x_0, x_1)$, model $p^i_{1|t}(x^i|x_t; \theta)$
Initialize an empty array: $\mathcal{A} = []$
**repeat**
    1) Sample $t$ form $U(0, 1)$.
    2) Sample $x_0, x_1$ from $\pi(x_0, x_1)$.
    3) Sample $x_t$ using Equation (4).
    4) Append $\frac{\dot{k}_t}{1-k_t} \sum_{i=1}^{L} \left( -\delta_{x_1^i \neq x_t^i} \log p^i_{1|t}(x_1^i|x_t; \theta) + 1 - p^i_{1|t}(x_t^i|x_t; \theta) - \delta_{x_1^i \neq x_t^i} \right)$ to array $\mathcal{A}$.
**until** Test set is exhausted
Return $\exp\left(\frac{\text{average}(\mathcal{A})}{L}\right)$

---

**Algorithm 3** Computing the alternative perplexity bound

---

**Input:** samples from $\pi(x_0, x_1)$, modeled $u^i_t(x^i, x_t; \theta)$, backward probability velocity $\tilde{w}_t$
Initialize an empty array: $\mathcal{A} = []$
**repeat**
    1) Sample $t$ form $U(0, 1)$.
    2) Sample $x_0, x_1$ from $\pi(x_0, x_1)$.
    3) Sample $x_t$ using Equation (4).
    4) Append $\sum_{i=1}^{L} \sum_{x^i \neq x_t^i} \left( u^i_t(x^i, x_t; \theta) - \tilde{w}^i_t(x^i, x_t) + \frac{p^i_t(x^i|x_0^i, x_1^i)}{p^i_t(x_t^i|x_0^i, x_1^i)} \tilde{w}^i_t(x_t^i, x) \log \frac{\tilde{w}^i_t(x_t^i, x)}{u^i_t(x^i, x_t; \theta)} \right)$ to array $\mathcal{A}$.
**until** Test set is exhausted
Return $\exp\left(\frac{H(p_0) + \text{average}(\mathcal{A})}{L}\right)$

---

# C. Additional Experimental Results

The foundational architecture of the model we use is that of Lou et al. (2024); Haxholli et al. (2025) which is based on the diffusion transformer paradigm outlined by Peebles & Xie (2023), which adapts the classic encoder-only transformer structure, such as that introduced in Vaswani et al. (2017); Devlin et al. (2019), by incorporating time-based conditioning. This approach introduces slight architectural modifications, notably the use of rotary positional embeddings as described in Su et al. (2024). Due to the addition of time conditioning, the model's parameter count is approximately 5% higher than that of a typical transformer (e.g., GPT-2). Tokenization and dataset splits are kept consistent with previous work to maintain comparability and minimize confounding variables.

The architecture comprises 12 transformer layers, each equipped with 12 attention heads and a hidden dimensionality of 768, matching the configuration commonly referred to as GPT-2. A dedicated conditioning dimension of 128 is used to capture temporal features essential to the flow process. It utilizes conventional scaled dot-product attention and applies a dropout rate of 0.1 to counter overfitting.

Regarding the training setup for OWT experiments, each model was trained with sequence lengths of 128 using a single GH200 GPU. The vocabulary includes 50,257 tokens, and the training batch size is fixed at 512. The training schedule encompasses 400,000 steps, and takes 44 hours in the standard case, which increases to 45 when using OT.

The OpenWebText dataset serves as the primary training corpus with local data storage employed to reduce latency. Optimization is handled via the AdamW algorithm, set with a learning rate of 3e-4, beta values of (0.9, 0.999), and an epsilon of 1e-8. No weight decay is used, favoring pure learning rate dynamics. A warm-up phase of 2,500 steps is included to enhance training stability, and gradient clipping is applied at a value of 1.

In the OT-EMA experiments we employ an Exponential Moving Average (EMA) of the model parameters with a high target decay rate of $\beta = 0.99999$. To prevent initialization bias, the decay is dynamically ramped up at step $t$ according to $\min(\beta, \frac{1+t}{10+t})$. Such EMA parameters are only used when calculation minibatch OT between embeddings.

## C.1. Character Level Shakespeare Experiment

Table 5 presents the results of the experiment described in Section 5.1, with the sole modification that the training set consists of the original Shakespeare text, without conversion to Morse code.

*Table 5.* Using minibatch OT reduces the number of jumps by $\sim 5\%$.

| MODEL (L=128) | JUMPS | RELATIVE JUMPS | PERPLEXITY |
|---|---|---|---|
| NORMAL | $113.23 \pm 0.002$ | 1.05 | 5.31 |
| WITH OT | $\mathbf{107.41 \pm 0.002}$ | **1** | **4.89** |

## C.2. Training Bound Comparisons

We train flows wherein the source distribution is chosen to be the Dirac delta at the sequence of all masked tokens. We choose $k_t = t$ in all cases. We tried 3 settings:

a) DFM-O uses cross entropy as the optimization objective as in Gat et al. (2024): $\int_0^1 \sum_{x_1,x_0} \pi(x_1,x_0) \sum_{x_t} p_{t|1,0}(x_t|x_1,x_0) \sum_{i=1}^L [-\log p_{1|t}^i(x_1^i|x_t;\theta)]dt$.

b) DFM-S is the flow matching approach which uses the bound in Equation 16 (as simplified in Appendix A.4): $\int_0^1 \frac{1}{1-t} \sum_{x_1,x_0} \pi(x_1,x_0) \sum_{x_t} p_{t|1,0}(x_t|x_1,x_0) \sum_{i=1}^L -\delta_m(x_t^i) \log p_{1|t}^i(x_1^i|x_t;\theta)dt$.

c) DFM-N is the same as DFM-S but multiplied by $(1-t)$:

$\int_0^1 \sum_{x_1,x_0} \pi(x_1,x_0) \sum_{x_t} p_{t|1,0}(x_t|x_1,x_0) \sum_{i=1}^L -\delta_m(x_t^i) \log p_{1|t}^i(x_1^i|x_t;\theta)dt$.

The model architecture in all cases is identical in design as the one in Section 5.1, but here we use the GPT2 tokenizer and to match related work, we train on OWT (Gokaslan & Cohen, 2019) for 400K steps with batch size of 512, sequence length of 128. For 'DFM-S', our bound becomes the MD4 of Shi et al. (2024) (see Appendix A.4). The bound is tested on the test sets found in Lou et al. (2024), more precisely: 1BW, LAMBADA, PTB, WikiText2 and WikiText103 (Chelba et al., 2014; Paperno et al., 2016; Marcus et al., 1993; Merity et al., 2017). In addition, we compare against SEDD of Lou et al. (2024).

The results can be below in Table 6.

*Table 6.* Perplexity bound results in the case of masked flow/diffusion variants.

| MODEL (L=128) | LAMBADA | WIKITEXT2 | PTB | WIKITEXT103 | LM1B |
|---|---|---|---|---|---|
| SEDD ABSORB | 67.06 | 69.39 | 208.67 | 69.18 | 83.86 |
| DFM-O | 71.90 | 71.23 | 221.62 | 70.80 | 82.60 |
| DFM-N | 67.50 | **67.00** | **204.80** | **66.65** | **80.29** |
| DFM-S | **66.61** | 68.48 | 208.37 | 68.04 | 81.46 |

## C.3. Section 5.3 Perplexity Bound Results

In Table 7 and 8, we provide the perplexity bound results on the five test sets for the models trained with cross entropy described in Section 5.3.

*Table 7.* DFM-B perplexity bound results comparing normal training vs OT.

| DATASET | LAMBADA | WIKITEXT2 | PTB | WIKITEXT103 | LM1B |
|---|---|---|---|---|---|
| DFM-B | 184.81 | 211.66 | 723.15 | 207.73 | 230.87 |
| DFM-B-SINKHORN | 190.21 | 204.16 | 654.88 | 204.22 | 222.42 |
| DFM-B-EXACT | 168.02 | 189.16 | 676.29 | 191.16 | 209.75 |
| DFM-B-EXACT-EMA | **167.10** | **175.09** | **618.78** | **176.53** | **204.14** |

Note that bound estimation for OT-trained models is problematic, as minibatch OT defines an implicit coupling we cannot access. Since sampling from this coupling during the calculation of the bound is impossible, we approximate it by sampling minibatches and performing OT on them. This heuristic approach makes such values only approximations.

*Table 8.* DFM-MMF perplexity bound results comparing normal training vs OT. Sinkhorn Solver at test time.

| DATASET | LAMBADA | WIKITEXT2 | PTB | WIKITEXT103 | LM1B |
|---|---|---|---|---|---|
| DFM-O | 71.90 | 71.23 | 221.62 | 70.80 | **82.60** |
| DFM-MMF | 68.65 | **68.38** | 204.17 | **68.68** | 85.09 |
| DFM-MMF-SINKHORN | **68.63** | 69.33 | **204.06** | 69.21 | 83.45 |

When we use exact OT during testing to get a better estimation of the optimal minibatch coupling and remove the potential repetition of testing samples we get the results in Table 9.

*Table 9.* DFM-MMF perplexity bound results comparing normal training vs OT. Exact OT solver at test time.

| DATASET | LAMBADA | WIKITEXT2 | PTB | WIKITEXT103 | LM1B |
|---|---|---|---|---|---|
| DFM-O | 71.90 | 71.23 | 221.62 | 70.80 | 82.60 |
| DFM-MMF | 68.65 | 68.38 | 204.17 | 68.68 | 85.09 |
| DFM-MMF-SINKHORN | 68.37 | 69.10 | 202.63 | 69.02 | 83.05 |
| DFM-MMF-EXACT | **66.27** | 67.60 | **197.05** | 67.62 | 82.98 |
| DFM-MMF-EXACT-EMA | 68.62 | **67.37** | 213.14 | **67.32** | **82.34** |

## C.4. Section 5.3 LLama-judged Generative Perplexity, Entropy and Standard Deviations

In what follows we present the full generative perplexity results of the experiments described in Section 5.3. That is, we show results when Llama is used as a judge, the entropy values and standard deviations (calculated over generated samples, not over runs). Such resulrs can be found in tables 10, 11 and 12.

*Table 10.* Results with and without minibatch OT. GPT-2 Large was used as a judge.

| GENERATION STEPS: | 8 | 16 | 32 | 64 | 128 | 1024 |
|---|---|---|---|---|---|---|
| DFM-B | 345.94 | 241.16 | 211.99 | 197.48 | 192.75 | 185.12 |
| STANDARD DEVIATION | ±1.71 | ±1.32 | ±1.12 | ±1.10 | ±1.04 | ±1.04 |
| DFM-B-SINKHORN | 331.88 | 233.24 | 203.08 | 191.17 | 185.06 | 178.53 |
| STANDARD DEVIATION | ±1.67 | ±1.26 | ±1.06 | ±1.00 | ±1.01 | ±0.96 |
| DFM-B-EXACT | 335.14 | 235.15 | 206.11 | 194.23 | 188.85 | 180.91 |
| STANDARD DEVIATION | ±2.03 | ±1.42 | ±1.31 | ±1.38 | ±1.22 | ±1.28 |
| DFM-B-EXACT-EMA | 302.76 | 208.42 | 182.40 | 169.89 | 164.42 | 159.87 |
| STANDARD DEVIATION | ±1.85 | ±1.34 | ±1.19 | ±1.03 | ±0.99 | ±1.04 |
| DFM-S | 587.80 | 316.25 | 222.46 | 188.62 | 169.81 | 156.81 |
| STANDARD DEVIATION | ±3.35 | ±1.85 | ±1.39 | ±1.23 | ±1.04 | ±0.97 |
| DFM-N | 556.73 | 296.25 | 210.11 | 176.34 | 160.17 | 147.07 |
| STANDARD DEVIATION | ±3.17 | ±1.73 | ±1.21 | ±1.08 | ±1.01 | ±0.91 |
| DFM-O | 560.67 | 300.06 | 208.06 | 175.59 | 159.03 | 146.54 |
| STANDARD DEVIATION | ±3.17 | ±1.78 | ±1.20 | ±1.08 | ±1.01 | ±0.89 |
| DFM-MMF | 536.50 | 288.38 | 204.77 | 170.61 | 155.45 | 143.48 |
| STANDARD DEVIATION | ±2.92 | ±1.65 | ±1.16 | ±1.02 | ±0.85 | ±0.95 |
| DFM-MMF-SINKHORN | 525.83 | 283.10 | 199.55 | 167.86 | 153.51 | 141.92 |
| STANDARD DEVIATION | ±2.87 | ±2.09 | ±1.31 | ±1.02 | ±0.95 | ±0.88 |
| DFM-MMF-EXACT | 518.86 | 281.39 | 199.68 | 168.12 | 153.51 | 141.46 |
| STANDARD DEVIATION | ±2.99 | ±1.57 | ±1.19 | ±0.97 | ±0.95 | ±0.98 |
| DFM-MMF-EXACT-EMA | 480.61 | 259.44 | 187.68 | 156.62 | 143.08 | 132.55 |
| STANDARD DEVIATION | ±2.66 | ±1.38 | ±1.10 | ±0.90 | ±0.79 | ±0.82 |

*Table 11.* Results with and without minibatch OT. LLama 3.1 8B was used as a judge.

| GENERATION STEPS: | 8 | 16 | 32 | 64 | 128 | 1024 |
|---|---|---|---|---|---|---|
| DFM-B | 394.67 | 283.71 | 252.04 | 235.98 | 231.61 | 223.77 |
| STANDARD DEVIATION | ±1.96 | ±1.66 | ±1.54 | ±1.50 | ±1.43 | ±1.41 |
| DFM-B-SINKHORN | 380.29 | 274.68 | 243.92 | 230.36 | 225.36 | 216.48 |
| STANDARD DEVIATION | ±1.96 | ±1.51 | ±1.51 | ±1.42 | ±1.48 | ±1.41 |
| DFM-B-EXACT | 387.05 | 276.84 | 248.27 | 232.19 | 227.61 | 218.93 |
| STANDARD DEVIATION | ±2.56 | ±1.88 | ±1.90 | ±1.70 | ±1.75 | ±1.86 |
| DFM-B-EXACT-EMA | 351.51 | 246.36 | 220.30 | 204.70 | 197.86 | 193.70 |
| STANDARD DEVIATION | ±2.24 | ±1.68 | ±1.59 | ±1.48 | ±1.44 | ±1.50 |
| DFM-S | 681.89 | 378.98 | 271.73 | 231.96 | 212.22 | 198.35 |
| STANDARD DEVIATION | ±3.97 | ±2.34 | ±1.83 | ±1.68 | ±1.60 | ±1.57 |
| DFM-N | 645.79 | 359.97 | 256.33 | 218.68 | 197.46 | 184.23 |
| STANDARD DEVIATION | ±3.71 | ±2.15 | ±1.61 | ±1.63 | ±1.37 | ±1.40 |
| DFM-O | 652.05 | 361.53 | 253.53 | 217.16 | 198.60 | 184.14 |
| STANDARD DEVIATION | ±3.76 | ±2.29 | ±1.59 | ±1.50 | ±1.54 | ±1.44 |
| DFM-MMF | 621.39 | 345.53 | 249.95 | 210.75 | 195.65 | 179.49 |
| STANDARD DEVIATION | ±3.10 | ±2.07 | ±1.55 | ±1.30 | ±1.65 | ±1.31 |
| DFM-MMF-SINKHORN | 620.84 | 348.39 | 243.31 | 210.87 | 191.42 | 178.62 |
| STANDARD DEVIATION | ±3.37 | ±1.96 | ±2.08 | ±1.33 | ±1.17 | ±1.45 |
| DFM-MMF-EXACT | 602.34 | 337.54 | 245.29 | 207.90 | 190.66 | 177.82 |
| STANDARD DEVIATION | ±3.47 | ±2.03 | ±1.64 | ±1.39 | ±1.35 | ±1.41 |
| DFM-MMF-EXACT-EMA | 555.42 | 311.88 | 230.42 | 194.47 | 178.12 | 164.78 |
| STANDARD DEVIATION | ±3.02 | ±1.81 | ±1.5 | ±1.26 | ±1.25 | ±1.23 |

Finally we show that entropy remains unchanged, unlike in the case of improper sampling of SEDD, in which the entropy was shown to drop up to 20% (Zheng et al., 2025).

*Table 12.* Entropy results with and without minibatch OT.

| GENERATION STEPS: | 8 | 16 | 32 | 64 | 128 | 1024 |
|---|---|---|---|---|---|---|
| DFM-B | 6.30 | 6.27 | 6.26 | 6.25 | 6.25 | 6.25 |
| STANDARD DEVIATION | ±0.001 | ±0.001 | ±0.001 | ±0.001 | ±0.001 | ±0.001 |
| DFM-B-SINKHORN | 6.30 | 6.27 | 6.26 | 6.26 | 6.25 | 6.25 |
| STANDARD DEVIATION | ±0.001 | ±0.001 | ±0.001 | ±0.001 | ±0.001 | ±0.001 |
| DFM-B-EXACT | 6.29 | 6.27 | 6.25 | 6.25 | 6.25 | 6.24 |
| STANDARD DEVIATION | ±0.001 | ±0.001 | ±0.001 | ±0.001 | ±0.001 | ±0.001 |
| DFM-B-EXACT-EMA | 6.29 | 6.26 | 6.25 | 6.25 | 6.24 | 6.24 |
| STANDARD DEVIATION | ±0.001 | ±0.001 | ±0.001 | ±0.001 | ±0.001 | ±0.001 |
| DFM-S | 6.36 | 6.32 | 6.29 | 6.27 | 6.26 | 6.25 |
| STANDARD DEVIATION | ±0.001 | ±0.001 | ±0.001 | ±0.001 | ±0.001 | ±0.001 |
| DFM-N | 6.35 | 6.31 | 6.28 | 6.26 | 6.25 | 6.24 |
| STANDARD DEVIATION | ±0.001 | ±0.001 | ±0.001 | ±0.001 | ±0.001 | ±0.002 |
| DFM-O | 6.35 | 6.32 | 6.29 | 6.27 | 6.25 | 6.24 |
| STANDARD DEVIATION | ±0.001 | ±0.001 | ±0.001 | ±0.001 | ±0.001 | ±0.002 |
| DFM-MMF | 6.35 | 6.31 | 6.28 | 6.26 | 6.25 | 6.24 |
| STANDARD DEVIATION | ±0.001 | ±0.001 | ±0.001 | ±0.001 | ±0.001 | ±0.002 |
| DFM-MMF-SINKHORN | 6.35 | 6.31 | 6.28 | 6.26 | 6.25 | 6.24 |
| STANDARD DEVIATION | ±0.001 | ±0.001 | ±0.001 | ±0.001 | ±0.001 | ±0.002 |
| DFM-MMF-EXACT | 6.35 | 6.31 | 6.28 | 6.26 | 6.25 | 6.24 |
| STANDARD DEVIATION | ±0.001 | ±0.001 | ±0.001 | ±0.001 | ±0.001 | ±0.002 |
| DFM-MMF-EXACT-EMA | 6.35 | 6.31 | 6.28 | 6.26 | 6.25 | 6.23 |
| STANDARD DEVIATION | ±0.001 | ±0.001 | ±0.001 | ±0.001 | ±0.002 | ±0.002 |

## C.5. Tightness of Bounds Evaluation

The expressions of the perplexity bounds are derived by initially dropping the term

$$-\int_0^1 \sum_{x_t} \bar{q}_t(x_t) \sum_{i=1}^L \sum_{x^i \neq x_t^i} \left( \tilde{w}_t^i(x^i, x_t) \log \frac{\tilde{w}_t^i(x^i, x_t)}{\tilde{v}_t^i(x^i, x_t)} + \tilde{v}_t^i(x^i, x_t) - \tilde{w}_t^i(x^i, x_t) \right) dt \qquad (176)$$

from the full expression of the KL divergence between the data and the learned distribution in Theorem 4.1. This term can be rewritten as

$$-\int_0^1 \sum_{x_t} \bar{q}_t(x_t) \sum_{i=1}^L \sum_{x^i \neq x_t^i} \left( r_{\bar{q}_t} w_t^i(x_t^i, x) \log \frac{r_{\bar{q}_t} w_t^i(x_t^i, x)}{r_{\bar{p}_t} v_t^i(x_t^i, x)} + r_{\bar{p}_t} v_t^i(x_t^i, x) - r_{\bar{q}_t} w_t^i(x_t^i, x) \right) dt, \qquad (177)$$

which shows that it depends on the ratios of induced pathwise probabilities under the model, which are intractable. Unfortunately, this makes this term difficult to estimate in practice, as computing these ratios would require summing over all possible trajectories that reach a given state at time $t$, which is infeasible due to the uncountable infinite number of such paths.

However, it should be pointed out that when the model learns the flow perfectly, this term becomes zero. Indeed, if $w_t$ matches $v_t$, then the induced probabilities, and therefore the induced ratios match so $r_{\bar{q}_t} = r_{\bar{p}_t}$ implying

$$-\int_0^1 \sum_{x_t} \bar{q}_t(x_t) \sum_{i=1}^L \sum_{x^i \neq x_t^i} \left( r_{\bar{q}_t} w_t^i(x_t^i, x) \log 1 + r_{\bar{p}_t} v_t^i(x_t^i, x) - r_{\bar{p}_t} v_t^i(x_t^i, x) \right) dt = 0. \qquad (178)$$

Therefore, we expect this term to decrease as the model improves and more closely approximates the target flow. Even though we cannot estimate the tightness of the bound in real settings, we evaluate it in simplified settings, by conducting the following three analyses.

*Our first analysis* is empirical. The vocabulary consists of three tokens: $M, A, B$ where $M$ is the masked state. The sequence length is two, and the ground truth probabilities over each states are: $P(A, A) = 0.15, P(A, B) = 0.5, P(B, A) = 0.05, P(B, B) = 0.3$.

We assume our model has learned the following imperfect flow:

$p^1_{1|t}(z^1, (M, M); \theta) = [0.9, 0.1]$, (so: $p^1_{1|t}(A, (M, M); \theta) = 0.9$ and $p^1_{1|t}(B, (M, M); \theta) = 0.1$),

$p^2_{1|t}(z^2, (M, M); \theta) = [0.1, 0.9], p^2_{1|t}(z^2, (A, M); \theta) = [0.2, 0.8]$,

$p^2_{1|t}(z^2, (B, M); \theta) = [0.3, 0.7], p^1_{1|t}(z^1, (M, A); \theta) = [0.8, 0.2]$,

$p^1_{1|t}(z^1, (M, B); \theta) =: [0.5, 0.5]$,

and as in the case of DFM-S and DFM-N, once the flow unmasks a token, it always predicts that same token in that position, with a probability of $100\%$. We run a Monte-Carlo simulation to calculate the probability assigned by this flow to each of the four states $(A, A), (A, B), (B, A)$ and $(B, B)$, which returns the following values:

$\tilde{P}(A, A) = 0.12953, \tilde{P}(A, B) = 0.58568, \tilde{P}(B, A) = 0.02529, \tilde{P}(B, B) = 0.2595$

Calculating the cross-entropy between the data and the modelled distribution using the ground truth probabilities and the probabilities above, we get $H(P, \tilde{P}) = 1.1626$. Then we use our bound in Equation (15) which in this case becomes the MD4 of Shi et al. The value of the bound is 1.2998 (that is, $H(P, \tilde{P}) \leq 1.2998$), which is about $11\%$ higher then the true value.

The difference between the precise NLL (1.90) and the NLL bound (with value 2.02) from Equation (98) is similar to the differences between the true likelihood and the bound reported in the case of continuous diffusion Song et al. (2021b, Thms. 1 and 3; Table 2).

*The second analysis* is theoretical. As before, the vocabulary consists of three tokens: $M, A, B$ where $M$ is the masked state, and we define a flow that is *independent* of the current state. The sequence length, as previously, is selected to be two. We choose a 'learned' flow such that the probabilities $p^i_{1|t}(x^i_t)$ of jumping to A are $a$ for the first position, and $b$ for the second one. Once a position is unmasked, it never changes just as in DFM-S and DFM-N.

We write the ground truth distribution over states $(A, A), (A, B), (B, A)$ and $(B, B)$ as $p(A, A), p(A, B), p(B, A)$ and $p(B, B)$. The true cross entropy is clearly: $-(p(A, A) \log ab + p(A, B) \log a(1 - b) + p(B, A) \log (1 - a)b + p(B, B) \log (1 - a)(1 - b))$.

Regarding the bound, for $x_1 = (A, A)$, we get $\int_0^1 \frac{1}{1-t} p(A, A)[(1 - t)^2(-\log a - \log b) - (1 - t)t \log a - t(1 - t) \log b]dt = -\int_0^1 p(A, A)[(1 - t)(\log ab) + t \log ab]dt = -p(A, A) \log ab$. Similarly, when calculating the rest, we get $-(p(A, A) \log ab + p(A, B) \log a(1 - b) + p(B, A) \log (1 - a)b + p(B, B) \log (1 - a)(1 - b))$ which is the true cross-entropy, i.e., the bound is tight for this setting. However, this example studies a simple case of a chain whose dynamics are independent of the current state.

*The third analysis* studies how the gap between the bound and the estimated terminal cross-entropy changes as the vocabulary size $V$ and sequence length $L$ increase. In the previous toy analyses, the imperfect transition probabilities could be specified explicitly. This becomes impractical once the state space grows, since the number of terminal states scales as $V^L$ and the number of partially masked states scales as $(V + 1)^L$. We therefore replace the hand-specified imperfect transition tables with a learned transition model.

Concretely, we construct an exact ground-truth distribution over sequences using a fixed bigram model. This gives a controlled non-factorized data distribution with adjacent-token correlations, while still allowing exact computation of the data entropy for the small values of $V$ and $L$ considered here. We then train a small MLP, with a deliberately bottlenecked hidden dimension of 3, using the masked-flow objective to approximate the conditional denoising probabilities $p^i_{1|t}(x^i_1 \mid x_t)$. The resulting network approximates the correct transitions well in the smallest setting but develops nonzero modeling error as the state space grows. After training, we evaluate the network on all partially masked states to obtain a lookup table

of learned transition probabilities. As in the first analysis, once a token is unmasked, the flow deterministically keeps it unchanged.

We report three quantities. The first is the data entropy $H(p_{\text{data}})$, computed exactly from the ground-truth sequence distribution. The second is the cross-entropy $H(p_{\text{data}}, p_\theta)$, estimated by simulating the learned continuous-time masked Markov chain over a dense temporal grid until conclusion and comparing the resulting terminal distribution to the ground truth. The third is the bound from Equation 15, which for masked flows simplifies to the MD4 objective. All other settings are kept the same as in the first analysis.

Table 13. Bound tightness for imperfect transition models as vocabulary size and sequence length increase.

| SETTING | DATA ENTROPY | CROSS-ENTROPY | BOUND |
|---|---|---|---|
| $V = 2, L = 2$ | 0.471 | 0.472 | 0.474 |
| $V = 4, L = 4$ | 2.204 | 2.287 | 2.302 |
| $V = 5, L = 5$ | 3.208 | 3.432 | 3.503 |

The results show the expected behavior. In the smallest case, the learned transition model is nearly perfect: the cross-entropy is almost equal to the data entropy, and the bound is correspondingly tight. As $V$ and $L$ increase, the MLP no longer recovers the exact conditional transitions, so the cross-entropy rises above the data entropy. The bound also rises, but remains close to the estimated cross-entropy. The relative gap between the bound and cross-entropy is approximately $0.4\%$ for $V = L = 2$, $0.7\%$ for $V = L = 4$, and $2.1\%$ for $V = L = 5$. Thus, the bound becomes looser as the learned dynamics become more imperfect, but the degradation remains gradual in these controlled settings.

This experiment should be interpreted as a small-scale sanity check rather than a direct estimate of the large-language-model regime. Already at $V = L = 5$, the terminal state space contains $5^5 = 3125$ states, which is close to the practical limit for accurately estimating the full terminal (learned) distribution by simulating the chain over many iterations. In realistic language modeling, the corresponding state space is vastly larger, for example $50257^{128}$ for GPT-2 tokenization with length 128. In that regime, exact cross-entropy estimation by enumerating or accurately sampling the terminal distribution is infeasible. The experiment nevertheless supports the qualitative claim that the proposed bound tracks cross-entropy closely when the learned transition model is accurate, and that the bound gap increases with modeling error.

### C.6. Dynamic and Kantorovich total costs

We generated 3200 samples for each (OT and non-OT), and measured the $L_2$ distance between the changed embeddings at each time steps across all positions, during generation. That is, if some positions change at time $t$ during generation, we add the $L_2$ distance between the embeddings of the changed tokens. We do this for all time points $t$ across all positions, and report the total sum of changes. Based on our first theorem, we expect OT to reduce this quantity, which it does as seen in Table 14.

Table 14. Transport costs for models trained with and without OT

| | DYNAMIC | KANTOROVICH |
|---|---|---|
| No OT | 6574.68 | 6507.24 |
| OT | 6328.71 | 6357.15 |

Similarly, for both, the model trained with OT and the one without, we calculate the coupling cost by computing the average of 1200 batches of size 512. This provides the estimated cost of the Kantorovich formulation. Results are shown in Table 14.

### C.7. Correlation Between the Perplexity Bound and Generative Perplexity Results

For each model in Table 3, we compute two metrics: (1) the average perplexity bound across 5 test sets, and (2) the generative perplexity measured by GPT2-Large with 1024 generation steps. The Pearson correlation coefficient between these metrics is 0.933, indicating very strong correlation. If we normalize the columns of the perplexity results in order to equalize the contribution of each testing set, then the Pearson correlation coefficient changes to 0.932.

## C.8. Ablation Experiments

**Ablation 1: OT Solver.** We replace the Sinkhorn solver with exact OT for multimasked flows (MMF). Results appear in Tables 15 and 16. We did not notice differences in training time.

*Table 15.* Exact OT (MMF) — 512 OT, 512 dff

| MODEL | 8 | 16 | 32 | 64 | 128 | 1024 |
|---|---|---|---|---|---|---|
| GPT | 518.86 | 281.39 | 199.68 | 168.12 | 153.51 | 141.46 |
| STANDARD DEVIATION | ±2.99 | ±1.57 | ±1.19 | ±0.97 | ±0.95 | ±0.98 |
| LLaMA | 602.34 | 337.54 | 245.29 | 207.90 | 190.66 | 177.82 |
| STANDARD DEVIATION | ±3.47 | ±2.03 | ±1.64 | ±1.39 | ±1.35 | ±1.41 |
| ENTROPY | 6.35 | 6.31 | 6.28 | 6.26 | 6.25 | 6.24 |
| STANDARD DEVIATION | ±0.001 | ±0.001 | ±0.001 | ±0.001 | ±0.001 | ±0.002 |

*Table 16.* Perplexity Results for the Experiment in Table 15

| SET | LAMBADA | WIKITEXT2 | PTB | WIKITEXT103 | LM1B |
|---|---|---|---|---|---|
| VALUE | 66.27 | 67.60 | 197.05 | 67.62 | 82.98 |

**Ablation 2: OT Batch Size.** We increase the OT batch size from 512 to 4096 while maintaining the flow batch size at 512, performing 8 parameter updates per OT batch. We continue using exact OT. Results appear in Tables 17 and 18.

*Table 17.* Generative Perplexity: Exact OT (MMF) — 4096 OT, 512 dff

| MODEL | 8 | 16 | 32 | 64 | 128 | 1024 |
|---|---|---|---|---|---|---|
| GPT | 518.52 | 281.71 | 197.45 | 164.64 | 152.22 | 140.15 |
| STANDARD DEVIATION | ±3.03 | ±1.64 | ±1.16 | ±0.96 | ±0.96 | ±0.80 |
| LLaMA | 602.80 | 339.27 | 240.70 | 204.79 | 188.87 | 177.11 |
| STANDARD DEVIATION | ±3.67 | ±2.13 | ±1.52 | ±1.35 | ±1.43 | ±1.28 |
| ENTROPY | 6.34 | 6.30 | 6.27 | 6.25 | 6.25 | 6.23 |
| STANDARD DEVIATION | ±0.001 | ±0.001 | ±0.001 | ±0.001 | ±0.001 | ±0.002 |

*Table 18.* Perplexity Results for the Experiment in Table 17

| SET | LAMBADA | WIKITEXT2 | PTB | WIKITEXT103 | LM1B |
|---|---|---|---|---|---|
| VALUE | 65.81 | 65.76 | 191.06 | 65.58 | 78.92 |

**Ablation 3: Number of Mask Tokens.** We test the sensitivity of Multimask Flows to the number of mask tokens $V_s$ by reducing it to 256 (instead of matching the data vocabulary size $V_d = 50,257$). We compare normal training (No OT) against our best configuration (Exact-OT-EMA). The generative perplexity results are provided in Tables 19 and 20, and the bound values in Table 21.

*Table 19.* Generative perplexities as measured by GPT2-large when using 256 masks.

| GENERATION STEPS: | 8 | 16 | 32 | 64 | 128 | 1024 |
|---|---|---|---|---|---|---|
| NO OT | 539.38 | 286.57 | 199.86 | 167.27 | 152.09 | 140.24 |
| EXACT-OT-EMA | 512.62 | 276.28 | 198.20 | 164.62 | 150.10 | 139.51 |

*Table 20.* Generative perplexities as measured by Llama 3.1 8B when using 256 masks.

| GENERATION STEPS: | 8 | 16 | 32 | 64 | 128 | 1024 |
|---|---|---|---|---|---|---|
| NO OT | 625.64 | 343.13 | 243.12 | 205.47 | 189.27 | 176.92 |
| EXACT-OT-EMA | 592.07 | 332.27 | 242.38 | 204.51 | 185.18 | 175.23 |

Entropy is virtually the same between the two configurations, differing by 0.01 at maximum. Clearly, in this case, the difference between the OT and the normal model is much smaller than when using $V_s = 50,257$. This highlights the importance of splitting the grid into tiny parts with mass $\frac{1}{V_s^L}$ in order to enable proper couplings. Notably, the amount of compute is completely independent from the number of masks, so zero compute overhead is added when increasing $V_s$ to match the full vocabulary.

Below we provide the perplexity bound results for this setting:

*Table 21.* Perplexity bound results when using 256 masks.

| SET | LAMBADA | WIKITEXT2 | PTB | WIKITEXT103 | LM1B | OWT TEST |
|---|---|---|---|---|---|---|
| NO OT | 70.27 | 65.62 | 204.31 | 65.62 | 82.05 | 39.06 |
| EXACT-OT-EMA | 67.07 | 63.21 | 199.46 | 63.39 | 77.70 | 37.64 |

Note that in this experiment, we split the OpenWebText (OWT) dataset into a 98% training set and a 2% testing set to evaluate the OWT test bound. As shown in Table 21, the results on this internal test set are consistent with the performance differences observed across the external test sets.

**Ablation 4: Alternative Cost Functions.** While we primarily explore cost functions induced by the Hamming distance and $L_2$ distance on embeddings, we briefly investigate one intuitive extension: using a learned cost function defined as the probability that the network generates the target sequence in a single step,

$$c(x_0, x_1) = -\sum_{i=1}^{L} \log p_{1|0}^i(x_1^i|x_0;\theta) \tag{179}$$

The generative perplexity and entropy results are provided in Table 22, and the bound values in Table 23. We notice that $L_2$ performs better for a larger number of steps, but for 8 and 16 steps this learned cost function significantly outperforms $L_2$.

*Table 22.* Generative Perplexity: Exact OT (MMF) — $p_{1|0}^i$ metric, OT minibatch size of 512, flow minibatch size of 512.

| MODEL | 8 | 16 | 32 | 64 | 128 | 1024 |
|---|---|---|---|---|---|---|
| GPT | 458.18 | 208.91 | 205.43 | 173.36 | 161.12 | 150.27 |
| LLAMA | 543.35 | 331.41 | 256.18 | 218.51 | 204.13 | 190.56 |
| ENTROPY | 6.30 | 6.27 | 6.26 | 6.25 | 6.24 | 6.23 |

*Table 23.* Perplexity bound results when using the $p_{1|0}^i$ learned cost function.

| SET | LAMBADA | WIKITEXT2 | PTB | WIKITEXT103 | LM1B |
|---|---|---|---|---|---|
| VALUE | 72.70 | 74.09 | 214.75 | 73.53 | 84.38 |

## C.9. Comparisons with Continuous Diffusion

In this subsection, in Table 24, we compare the discrete approaches with the continuous diffusion PLAID (Gulrajani & Hashimoto, 2024). The PLAID results are taken from Lou et al. (2024). The metric used is the perplexity bound.

*Table 24.* Results comparing SEDD, DFM-N, GPT2, and PLAID.

| MODEL (L=1024) | LAMBADA | WIKITEXT2 | PTB | WIKITEXT103 | LM1B |
|---|---|---|---|---|---|
| SEDD | 52.18 | 42.02 | 117.00 | 41.83 | 80.79 |
| DFM-N | 53.19 | 42.00 | **111.58** | 41.64 | 77.87 |
| GPT2 | **49.02** | **37.68** | 134.13 | **37.55** | **58.92** |
| PLAID | 57.28 | 51.80 | 142.60 | 50.86 | 91.12 |

# D. Introduction to Discrete Diffusion Models

## D.1. Discrete-Time Markov Chains Over Finite-State Spaces

A stochastic process $X_1, X_2, \ldots, X_T$, where each state $X_t$ depends solely on the preceding $X_{t-1}$ is called a discrete-time Markov Chain (DTMC). If the states $X_t$ can take any value from the set $\{1, 2, \ldots, S\}$, where $S$ denotes the total number of possible states, and $T$ represents the number of time steps, then we say that this process is a finite-state space DTMC. The probability that at time $t$ we are at $x$ is

$$p_t(X_t = x) = \sum_{y=1}^{S} p(X_t = x, X_{t-1} = y) = \sum_{y=1}^{S} p_{t|t-1}(X_t = x|X_{t-1} = y)p_{t-1}(X_{t-1} = y). \tag{180}$$

If we place all such probabilities $p_t(X_t = x)$ in a vector $s_t$ of shape $S \times 1$, such that $s_t(x) = p_t(X_t = x)$, then from above we can deduce that

$$s_t = Ps_{t-1}, \tag{181}$$

where $P(x, y) = p_{t|t-1}(X_t = x|X_{t-1} = y)$. Given an initial probability distribution $s_0$ over states, the equation above fully determines the evolution of the probability over states with respect to time.

## D.2. Continuous-Time Markov Chains Over Finite-State Spaces (Discrete Diffusion)

It is possible to define a stochastic process with the Markov property in finite-state spaces, for $t \in [0, T]$, (Doob, 1953). As previously, we can define a discrete-time process, on time points $\{0, \epsilon, \ldots, T - \epsilon, T\}$, such that there is $\epsilon$ probability of activating the previous transition mechanism when progressing from time $t - \epsilon$ to $t$, otherwise we stay where we are with probability $(1 - \epsilon)$. Removing the random variables to simplify notation, we have

$$p_t(x) = (1 - \epsilon)p_{t-\epsilon}(x) + \epsilon \sum_{y=1}^{S} p_{t|t-\epsilon}(x|y)p_{t-\epsilon}(y). \tag{182}$$

We notice that when $\epsilon = 1$ the equation above coincides with Equation (180), and in addition as before we can write Equation (182) in matrix form

$$s_t = (1 - \epsilon)s_{t-\epsilon} + \epsilon Ps_{t-\epsilon} = (I + \epsilon(P - I)) s_{t-\epsilon} = (I + \epsilon Q) s_{t-\epsilon} \text{ , where } Q = P - I. \tag{183}$$

From Equation (183), we see that $\frac{s_t - s_{t-\epsilon}}{\epsilon} = Qs_{t-\epsilon}$, which when taking the limit $\epsilon \to 0$ becomes $\frac{ds_t}{dt} = Qs_t$. Given an initial probability distribution $s_0$ over states, the equation above fully determines the evolution (flow) of the probability $p_t$ over states with respect to time. Indeed, the distribution over states at time $t$ is $s_t = e^{tQ}s_0$. This formulation can be generalized, such that $Q$ is allowed to evolve with time,

$$\frac{ds_t}{dt} = Q_t s_t. \tag{184}$$

For the choice $Q_t = \sigma'(t)Q$, where $\sigma$ is monotonically increasing, $\sigma(0) = 0$ and $\lim_{t \to 1} \sigma(t) = T$, we have $s_t = e^{\sigma(t)Q}s_0$. Matrices $Q$ must satisfy the properties of transition-rate matrices (Suhov & Kelbert, 2008), that is, they have non-negative non-diagonal entries, and the elements in each column add to 0. The choice for $Q$ is made such that: $s_1$ is an easy reference distribution to sample from and the matrix exponential $e^{\sigma(t)Q}$ is easy to calculate (Austin et al., 2021; Campbell et al., 2022). Unfortunately, these conditions greatly restrict the design space in this framework.

# E. Generated Samples

The following are non-cherrypicked text samples generated from GPT-2–sized models trained under various experimental setups. Outputs may contain hallucinations, inaccuracies, or culturally sensitive content. They are presented solely to illustrate qualitative differences in generation behavior, such as coherence, topical relevance, fluency, and do not reflect the views or endorsements of the authors.

*Listing 1.* Generated text from DFM-O, with sequence length L=128.

```
 Take a good look at running on ice volleyball ball from the sidelines. Do a party
    crunch once and get bored from another game away. Then mess something with a
    determined and pleasing smoke summon. Might not change.

It would have happened if I were busy much less myself.

When your fictional boss feels threatened these dark mysteries are no warning to ignore
    .

Instead ignore what you're working for and watch then acteduate what you're doing on
    view, move around the screen and how your boss detected Kung Fu as around you. They
    are not throwing a police officer at your feet. They are just accepting
====================================================================
 on Blue Bird" in the Night. Considered a regular occurrence in contemporary daytime
    arts circles, as well as the soundtrack to The Breakfast Club (1979), The Heavens
    Door, Russian-inspired duo's nature, Ooboh (and Zoeppo In Peace), and even a
    German-wave song Not To Olmy (1977 album). The highlight of the album's Elephant A
     Ring is the song's Kiss Of Saint John (December 1950), in which the island
    inhabitants embrace a beautiful Viking.

gluk188b - Now the more authentic Azgothic's complex,
====================================================================
 folk culture to the masses. In 1900, Dash organized the New Draveenjoci Friends Dinah
    festival, brought together with 50 local folk groups, including canoeclub, and
    First Father Township.

I spoke with other Dile Dash guardians listening to the show. She's the eighth person
    to stand near church faces. Getting some of the staff to volunteer, there were lots
     of screaming in hopes of hearing someone who feels the right to join a church
    member or perform the go-to edition Untitled. To calm their spirits, winners
    brought up the fact that "Polynesicans respect God's
====================================================================
 sessions for a down. Even after that, it's all over the board as the V&L must-ens.
     They're also sure to add the other survivors: wildcat goal catchers. -Ben Harper,
    punter reporter (HL)

But it's still challenging to be able to gain a reaction, especially with the tack
    dropping far further down. Instead of matching up with position experts, I started
    to assess how they would perform and I started to track nightly games against '80
    first-team coaches.

The fielders had to look past Dumervil, with Gibbs being
```

*Listing 2.* Generated text from DFM-N, with sequence length L=128.

```
 the migrant galeslam in Calais.

The Telegraph reported that Lord Dacre's Wembley address included an additional
    briefing on the complaints.

The EU referendum, on which he was asked to vote, said:

'But I am profoundly disappointed with this piece of inquiry that was appointed to
    breach the rules and EU rules and it is unlikely that his Labour Government will be
     affected.

'The Government has lost sight of this blatant interference and has attempted to ignore
     it.' -Ojes

To the Daily Mail, Jimmy once commented: 'Scared to say the verid
================================================================
s commit investment by defer to the SEC.

The Governor raised serious concerns about stopping the proposed measure by arriving to
     New York City on the day of the pact's July 31 deadline - if legal - though it
    would probably do little to cut any gains for his state's most successful investors
    .

Brown administration officials have ruled out complying with short-term hedge funds
    trading rules. That still appears to be only a possibility as any OIRP deals seem
    to crash or soon come into force.<|endoftext|>There is "no chance we either profit
    from the #LossLiveup." - the MMQB

================================================================
 good.

30 Cole Springfield 2016

Springfield alone averaged a superb .667 in his junior season with a 6-inch pitcher, 6-
    foot hoop, a 14-curry well and a close connection.

As close as any player can have in a baseball academy (the last time he had a game) is
    Gavin Bentley. Less former WSU defensive lineman. But Mayau had his playing style
    over someone else.

31 James Wood, 1925-2002

As well known Oxford export, the Tigers "There Were None" for his fellow
    topronouncement of Juneau, who was the
================================================================
, and did choke off the second one to show the new media coverage. I'm just going to do
     the rest of our work and ask the city of Montreal to discontinue the multi-year
    tradition of photography. That's my last piece. Here in Montreal, we're excited to
    try to work our cities way our working-class citizens.

The The Tonge Room, Le Grand Le Collective is a celebration of various global
    libertarian and anarchist events. Read more on live music from our first event.
    Read more about our team. We spoke with the Chanesque Art Project director about
    the theme we set out for Rockavaloon
```

*Listing 3.* Generated text from DFM-S, with sequence length L=128.

```
motion of a catcher in which the mechanical properties of the cooling fluid's
    electrical discharge are sure to be overcome of depth" says Benfeldt, half year
    undergraduate in medical tics, in the 2005 semester, "where we needed to develop a
     more comprehensive model of the precipitation of motion and equality of motion
    general to animal dimensions, there has been a discussion about deluge, Form, spin
     and Motion".[3]

Field motion has played a natural role that mimics parallel movements in the laying and
    loading of a field-dependency container, and therefore change the accepted realism
    of motion. Intuitively, when one can demonstrate non
==================================================================
 the help of Indian FA Dr. Natalia Sekuni to help NYC full backs Lilian Balfour and
    Remis Elijah Mahrez.

Maryab Kardy also had three league games throughout his career with Toronto FC.

Korian scored 4.5 goals and 2 assists in 24 Bundesliga appearances last season, first
    for FC Nordsbank Leiburg and has 10.4 goals, 5 assists in 7 starts this season. He
    collected a 1-0 assist for MacLilleux in 1914-19.

Korian scored one goal against FC Seattle minutes into the game and led the Reds to a
    21-
==================================================================
 DeVos delivered various policy and campaign announcements for him.

Cuomo later claimed that he had seen "thousands" of potential voters in the state.

Sanders, who spoke in the city in November, charged whether Trump would boost the
    economy, saying, "This is the way I use something I know because everybody in
    America has a great choice and both parties today. He had said that the system
    worked for some but that there were a better solutions for the voters."

Trump may present himself as he most likely to have a marquee issue.

Still, Trump did not just hear thunder from his previous candidate Trump,

==================================================================
 in the countryside. Even though the government's actions were however far out to be
    heartening the protesters so deeply, many peasants who were intending to give the
    poor the title instead had asked why they should choose to be a representative
    citizen and therefore stand up to defend the peasant family as well as society.
    Immediately, after we saw the working class and even the middle-level
    intellectuals in one segment, they had little doubt that the class that raised
    them was all or part by them of resisting the situation, showing why working
    classes can be an irritable about the bourgeois who participated irresponsible
    actions and drove the country up to chaos.
```

*Listing 4.* Generated text from the DFM-B model with BoW source distribution and normal training, with sequence length L=128.

```
 but it certainly doesn't exclude modules for employees from sand Reels resorts," said
     Ziefen. Those boutique area stores will also draw the attention out of local
     rushers and American area brokers.

"This office doesn't see it as a part of proving that a profitable stand-up
     representative, clean, independent business."<|endoftext|>Keith Young's secret of
     the worms' DNA may come from the Cparagon Green while studying the region's
     biodiverse flora. Draggio early cartilaginian creatures, from one of their native
     heights to the earth to corn and barley leaves, stirred their
================================================================
's program was broken down into monopoly vehicle lending. Wells Fargo and The New York
     Times are the largest auto lenders seeking infinite certainty on loans. And despite
      the design principles they must comply with the law Wisconsin auto lenders are
     requiring Wells Fargo because no one denies compliance.

Last year, the federal government closed its loan to college and micro-urch, said the
     group of governors. Families across the state also have concerns that while the
     costs of the loans are "viable, federal lenders were allowed to prevent borrowers
     from using acceptable financing policies because federal loans, including seeking
     credit default, are denied."

Even if
================================================================
 and I'm good at learning a few more stuff, I bet that he's the second roaster supreme
     in authority in school that doesn't do any arithmetic at all."Christopher
     Goldstein and Marc Cruz

A friend Jonathan has done quite a little art, and I am sure you've heard of English
     Roles, Almor and Sacnegramald.

It's obvious that we have found guilty of a terrible English plot at this time and so
     are 22-year-olds. It might be instructive to get on blowing the sand and doing
     masters-levels without getting
================================================================
 I call itself a farmland: "hot habitat garden enticiest that our civic/boxing
     advantage won't have to dismantle overnight;

Mayer that life is simply being ecologically ec

Note: That sets me out to it: stupidly conscious beautiful plants versus stupidly
     conscious living beings. If we leave Human space then it will feel much less secure
     .

And grafts down over solutions down on imperfect.

Which is terrible in my research, & which is why we have a political ecologic.

A key inner dilemma in thinking of ecological phenomena is aging. Their vitality
     involves increasing drastically
```

*Listing 5.* Generated text from the DFM-B-Sinkhorn model with BoW source distribution and minibatch-OT training, with sequence length L=128.

```
fuel festivals should serve as their short-term goals.

Only one of many Charity & Human Advocates have been written in the past to promote
    free free markets. They can give invitations and help repute any who describe the
    offer and apply their own informational refinements. The resistance to or even
    being that the FairMormon group will have to come to educational causes and careers
     and thinkers from not only in Millionaires Web Groups, educational and family
    activities. These groups can also donate by mailing lists to kiosks by Comeback and
     drive by the same publisher Samples from that advocacy group by Oxfam. For many
    birthday auctions, groups
================================================================
 Queen City Building in Albine, Romania, the United House said.

Agrini's Airport-blocking congested Charles Avenue area was Baldini's first free kick
    when his 10-yard header gave the Italian side the league first of the World Cup,
    and won them two World Championship and trophies.

The Italian native, aged just 20, won his first World Cup title, West Club Athletes
    Player of the Year, and received the Interim Di'Solo from the Udinese club's run of
     P2.5 million, a deal that will be considered a move in a Napoli bid.

Speaking
================================================================
 in a way.) Excellent, by e-mails me (good) Shihuan - and its reader - for this
    question, I have already translated this piece into fantasy literary thriller.
    This book is phenomenally interesting and amusing and is not really even in
    writing. The explanation is particularly fascinating to add to the fan community
    and this book contains lots of spoilers.

I started writing earnest Vegan Essentials. That book was attached to the Rules of My
    Feast! My first reviews were seven years ago and is still a category ninth. This is
    due to ongoing vegan activism and loyalty to other affected readers.

Vegan everyone at the
================================================================
 book entry based on Midnight Symphony. It's an addictive decision, and it's going to
    become what's deciding actions it is going to take to mirror what I said to people
     to Love Your Experience-a kind of confluence.

Here's the first theatrical teaser trailer, high-speed footage from Rex Arena Theatre
    with Howard Aller and Marshall Carter at the New Sound Day Festival this summer.

But with all the scenes in actual driving mode without going in front of a production
    vehicle, I think there's a huge difference where you're essentially in cycling mode
    ; I
```

*Listing 6.* Generated text from the DFM-MMF model with multi-mask source distribution and normal training, with sequence length L=128.

```
 laws were part of the way to keep everybody related to one type of plants in their
     social roles."

Mr. Zhou shushed, saying, "My vocals won't show up for weeks, but we'll be showing
    civil expressions. We wanted to call for community involvement."

At 26, Noonan was really pushing for contentious statements.

To prove the point, Tonelli participated in a group meeting in Hannamkel, Georgia (you
    won't find room between sour'' and screwhead.k spoke toward all of these dairy
    farmers). The four always told each

====================================================================
 is through lots of reporting and reerforming ways on the realised check.

In essence, a simple move gives the developer a rescall of mutable and push limits for
    wethers which is typical in code. To start and alleviate checksums and other
    exceptions. I find this technique is especially actually interesting, when times
    are changing and a design change for push limits a degree away from mutable and
    wethers:

Implementation

The code explicitly gives the right to namespace crash if the dynamic codebase's
    changing anything, and nothing that does change triggers entryulating. On the other
     hand, providing

====================================================================
 somewhat), and police encryption systems discussed in detail don't with this approach
     have very high level security.

The most significant hole would be close between the fake Syed GED listing and the
    bogus public AR-15 in restriction that they were unaware of NSA activity in the
    past. Instead, they enlisted isardars like Shin Intel's George Singleton TP Program
     to gain access to a small subset of unknowns Syed before April 2002.

Borse and spoofing is never wise enough, however. There was a whole site beside that
    old swatch and telly when it first was instigated by the feds. While a

====================================================================
 behaviour changes, resulting in a some degree of glacial DNA diversity in the
     signaling system. The research results suggest similar variations in the system
     evolved, at least since 2004.[48]

and the initial development of military radio networks began. In January 2008, to fund
    his experiments, the US founded Jo Kutus, a micro-channel engineer and orthopedic
    surgeon near Ulisz, south-west to decodel dynamic television imagery and dropped it
     rain. The total cost would be US$135 million for the I-3 plan and 10% before TV
    broke.

Except an ideal early model, most countries
```

*Listing 7.* Generated text from the DFM-MMF-Sinkhorn model with multi-mask source distribution and minibatch-OT training, with sequence length L=128.

```
 " came out earlier this year.

Absent Films begins production in partnership with the Entertainment Agency Europe (
    MEGO), the Italian news agency Gazeta and Spain's Forza National Investigation
    Agency (ASIO. It is said to be producing about 21 films worldwide, titled "Nobody
    Loves Worries."

Absency Films' CEO, Michael Agiloh, offers an explanation of why many of his characters
     appear on screen -- "In these many shows, each of us are in the center of our
    heads, filled with energy to do that go outside our cells to stimulate, stimulate,
    and recreate love

================================================================
 Dudley on the brink of a Joakier contract, the Knicks could be more optimistic this
     summer, not on any physical trade for Thomas.

[An MLB free trade period. Here's what we have here.]

They are also no longer in the trade market for center Raymond Felton, which was traded
     to Andrea Bargnani and was busy signing out a 2021 deal. Ono, who has been a
    productive player on the roster, would get significant financial relief with a new
    deal. Thomass agent signing would mean he leaves an expiring contract after the NBA
     season in 2017.

For longer, they

================================================================
 random two Anthrax databases, when marked cases are cleaned up, and one still has
     access to records from within the center of a case.

Researchers say they have concocted an elaborate system of polygraphs that process
    documentions, original polygraphs, which can then be used to sort and review on
    file case in a bid to preserve the files.

Public Citizen, which organized the document,, learned of the recording of phone
    conversations between President George H. W. Bush of Odessa, Conn., during a
    February 2005 trip to New York City.

The FBI is using its investigative techniques to shut down a

================================================================
Repeat this after under Run as menu and under Advanced. Now we've obtained the empty
    Zone_X file so application requires synchronous and Authentication to search for
    those within Zone. After doing so, the field name for the Zone_X and the
    abbreviation the Zone_X field are activated.

The first and second column or column change Zone's current property. Bring it back
    from the new view to complete the Forms.

Above will show some settings generated in the previous view that was enabled by bot-
    originator . Here they appear in a Parameters screen :

Siren Bot Name Screen Off-screen Name Dimensions
```

