# OpenReview forum: "Minibatch Optimal Transport and Perplexity Bound Estimation in Discrete Flow Matching"
_ICML.cc/2026/Conference — ICML 2026 regular_

### Official Review · Reviewer_jU4y · 2026-03-07

**Soundness:** 3
**Presentation:** 2
**Significance:** 2
**Originality:** 2
**Overall Recommendation:** 3
**Confidence:** 4

**Summary:**

This paper introduces a path-length-oriented optimal transport (OT) objective for Discrete Flow Matching (DFM), which aims to minimize dissimilarity-weighted jumps between states. The authors show its connections with Kantorovich and Benamou-Brenier theorem. To enable the application of OT to practical models, the paper proposes multi-mask flows (MMF), which use multiple mask tokens to create a richer source distribution. In addition, the paper derives two computable upper bounds for perplexity in DFM models. These bounds provide a principled way to train and evaluate models. Through experiments on OpenWebText (OWT), the authors demonstrate that minibatch OT improves performance. It also achieves the same generative perplexity with up to 8x fewer inference steps.

**Compliance With Llm Reviewing Policy:**

Affirmed.

**Key Questions For Authors:**

1. Could the authors provide a comparison table that summarizes the objectives and sampling processes of DFM-B, DFM-S, DFM-N, DFM-O, DFM-FFM, DFM-B-OT, DFM-FFM-OT, DFM-FFM-Exact, DFM-FFM-Exact-EMA?
2. Given the fact that the proposed method is scalable (Section 5.5), could the authors provide experimental comparison between DFM-MMF-OT under the same L=1024 setup?
3. Could the authors also provide the validation perplexity on OWT?
4. The MMF introduces multiple mask tokens. Is there a guideline behind choosing the optimal number of mask tokens ($V_s$)? The paper uses $V_s =V_d$, but is performance robust to this hyperparameter?
5. The choice of similarity measure $s$ is central to the OT objective. While the paper explores Hamming distance and L2 distance on embeddings, have the authors experimented with other similarity measures?

**Limitations:**

yes

**Strengths And Weaknesses:**

**Strengths**
- The paper is well-written. The notations and equations are clear and well-presented.
- The empirical results show a substantial decrease in the number of inference steps needed to achieve a target generative perplexity.

**Weaknesses**
- The paper introduces numerous model variants (DFM-B, DFM-O, DFM-N, DFM-S, DFM-MMF, and multiple OT-enhanced versions), making it challenging to isolate the contribution of each individual component. While the best overall model is identifiable (DFM-MMF-EXACT-EMA), the incremental value of multi-mask tokens, different OT solvers, and the EMA mechanism is not presented in a systematically controlled ablation. (See Q1)
- The main proposed MMF models are excluded from the direct comparison with SEDD and GPT-2 (Table 2). This comparison is conducted only with single-mask models (DFM-N) at a longer sequence length (1024). As a result, it is unclear how the best-performing multi-mask models would perform against these established baselines, or whether the derived perplexity bounds remain effective. (See Q2)

---

> ### Author Rebuttal · Authors · 2026-03-31
>
> We sincerely thank the reviewer for their time and in-depth review. We are glad the reviewer finds the paper to be well-written and appreciates the empirical results.
>
> **W1 and Q1: The paper introduces numerous model variants..**
>
> We thank the reviewer for raising this point. In response, we will augment Table 3 with additional results and improve the paper’s presentation by elaborating on the following points:
>
> Moving from masked to multimasked flows (DFM-MMF) improves generative perplexity without hurting diversity. Adding minibatch OT via Sinkhorn improves it further, while switching to an exact solver yields only modest gains, mostly in the low-step regime. The biggest jump comes from using EMA with exact OT , which dramatically boosts results. The takeaway: multimasked flows enable minibatch OT and contribute meaningfully, but OT with EMA drives the bulk of the improvement.
>
> DFM-B flows underperform at high step counts while excelling in the low-step regime in comparison to MMFs. Overall, DFM-B flows follow the same pattern: Sinkhorn helps, exact OT is comparable, and exact OT with EMA again delivers a dramatic uplift. In fact, we found that the number of steps can be reduced 32 times from 1024 to 32 to reach the same generative perplexity without any losses in entropy (diversity):
>
> |Generation Steps|8|16|32|64|128|1024|
> |-|-|-|-|-|-|-|
> |DFM-B|345.94|241.16|211.99|197.48|192.75|185.12|
> |DFM-B Exact|335.14|235.15|206.11|194.23|188.85|180.91|
> |DFM-B-Exact-EMA|302.76|208.42|182.40|169.89|164.42|159.87|
>
> These rows should be seen as additions to Table 3 in the paper. Similarly, below we show the perplexity bound results which are to augment Table 7.
>
> |Dataset|Lambada|Wiki2|PTB|Wiki3|LM1B|
> |-|-|-|-|-|-|
> |DFM-B|184.81|211.66|723.15|207.73|230.87|
> |DFM-B-Exact|168.02|189.16|676.29|191.16|209.75|
> |DFM-B-Exact-EMA|167.10|175.09|618.78|176.53|204.14|
>
> We would be very happy to provide additional explanations if anything remains unclear.
>
> **W2 and Q2: The main proposed MMF models..**
>
> Unfortunately, due to the limited rebuttal period and compute constraints, we were unable to train an MMF model with L = 1024. Nevertheless, we expect the results to extend naturally to larger sequence lengths. The main modification that may be required is retuning the EMA hyperparameters, as changing L can alter gradient magnitudes when EMA is used.
>
> However we point out that, for $L=128$, we do compare the perplexity bounds between all the models we introduced and all masked flow variants: Tables 6, 7, 8, 9 in the Appendix. We intend to gather all these results in single table as well.
>
> **Q3: Could the authors also provide the validation perplexity on OWT?..**
>
> The models presented in the paper were trained on the full OWT dataset, as is standard in the literature. Therefore, for those experiments, it is not possible to report validation perplexity on OWT. However, we do report such results for the new experiments below (Q4). These results are consistent with the performance differences observed across external test sets.
>
> **Q4: The MMF introduces multiple mask tokens...**
>
> We believe that increasing the number of mask tokens is generally beneficial, in particular since it does not introduce any computational overhead. For smaller networks, however, doing so increases memory consumption. In larger networks, the embedding matrix constitutes a smaller fraction of the total parameter count, so the number of masks can be increased further, especially since compute remains unaffected. As suggested by `R-3(K83J)`, we report results using 256 masks in our response to their Q1. In that setting, the gap between the OT model and the baseline becomes smaller, which highlights the importance of partitioning the grid into very fine regions with mass $\frac{1}{V^L}$ in order to enable more effective couplings.
>
> **Q5: The choice of...**
>
> This is a very interesting direction. However, given the large number of experiments already included in the paper, we believe it would be better explored in a dedicated follow-up paper so that it can be studied thoroughly. Here, we briefly investigate one intuitive extension: using a learned cost function defined as the probability that the network generates the target sequence in a single step,
> $$ c(x_0, x_1) = - \sum_{i=1}^L \log p_{1|0}^i (x_1^i|x_0; \theta)$$ The results are provided below.
>
>
> *Generative Perplexity: Exact OT (MMF) — $p_{1|0}^\theta$ metric, 512 OT, 512 flow*
>
> |Model|8|16|32|64|128|1024|
> |-|-|-|-|-|-|-|
> |GPT|458.18|208.91|205.43|173.36|161.12|150.27|
> |LLaMA|543.35|331.41|256.18|218.51|204.13|190.56|
> |Entropy|6.30|6.27|6.26|6.25|6.24|6.23|
>
> *Perplexity Bound Results*
>
> |Set|Lambada|Wikitext2|PTB|Wikitext103|LM1B|
> |-|-|-|-|-|-|
> |Value|72.70|74.09|214.75|73.53|84.38|
>
> We notice that $L_2$ performs better for larger number of steps, but for 8 and 16 steps this cost function significantly outperforms $L_2$.
>
> We again sincerely thank the reviewer for their thorough and constructive review.

---

> > ### Author Rebuttal · Reviewer_jU4y · 2026-04-05
> >
> > I thank the authors for their thorough response, which have partially addressed my concerns. I will maintain my assessment of this paper.

---

> > > ### Author Response · Authors · 2026-04-07
> > >
> > > We sincerely thank the reviewer for their time, their careful evaluation of the paper, and their engagement with our rebuttal. We are glad the reviewer found our response thorough, and we appreciate the acknowledgment that it partially addressed the initial concerns.

---

### Official Review · Reviewer_K83J · 2026-03-11

**Soundness:** 3
**Presentation:** 2
**Significance:** 3
**Originality:** 3
**Overall Recommendation:** 4
**Confidence:** 2

**Summary:**

This paper works on discrete flow matching (DFM) for categorical data generation. Since discrete paths are stochastic, two tools from continuous flows don't carry over: rectification (for straightening paths) and change-of-variables (for exact likelihoods),  and the authors proposes solutions for both. For path efficiency, a dynamic optimal transport objective is formulated and shown to equal a Kantorovich cost depending only on per-token similarity which enables minibatch OT training. For evaluation, the authors propose two perplexity upper bounds and also introduces multimask flows, using multiple mask tokens to allow non-trivial OT couplings. Experiments at GPT-2 scale on OpenWebText shows up to 8× step reduction while only incurring marginal training overhead.

**Compliance With Llm Reviewing Policy:**

Affirmed.

**Key Questions For Authors:**

1.  How sensitive are multimask results to the number of mask tokens? Would 256 or 1000 work comparably to 50,257? Also how much of a computational overhead would this introduce?
2. How would the  tightness of the perplexity bounds scale with vocabulary size and sequence length? Is there a feasible way to estimate the bound gap at a higher scale, even approximately? If not, even empirical results showing the variation when scaling to a slightly higher scale would be valuable.
3. Any intuitive explanations for why decoupling embedding optimization from OT coupling computation helps so much?
4. For the L2 similarity metric, how sensitive are results to the embedding initialization?
5. The improvement in step counts is useful, but could you also provide the actual wall-clock generation (and training) time comparisons to confirm the practical speedup claim?

**Limitations:**

yes

**Strengths And Weaknesses:**

## Strengths

1. The main theoretical result (Theorem 3.1) connects the dynamic and Kantorovich formulations for arbitrary similarity functions. The equality holds per-coupling rather than requiring infima over flows, which appears to go beyond the classical Benamou-Brenier setting.

2. The perplexity bounds give DFM a way to estimate likelihoods and the high correlation with generative perplexity in experiments suggests they can be used to rank models well.

3. Minibatch OT is shown to add only a slight training overhead while achieving practically significant reduction in inference steps.

4. Multimask flows seems like a simple yet practical idea that addresses the issue of having trivial couplings in standard masked flows.

## Weaknesses

1. The presentation could be clearer. Some core concepts are used without sufficient background, which can make the paper harder to follow for readers not already familiar with the area.
   - Optimal transport (Kantorovich formulation, Benamou-Brenier theorem, minibatch OT) could use a self-contained introduction.
   - Multimask flows appear somewhat late and could be motivated earlier, and the trivial coupling issue that makes them necessary could be stated more upfront and in detail.
   - The role of convex interpolants in enabling Theorem 3.1 isn't very intuitive.
   - Tables would benefit from a more detailed captioning, and maybe also labeling which source distribution each model uses (BoW / multimask / mask).

2. Bound tightness at realistic scale is unclear. The ~11% gap between bound and true NLL is only in very small settings. Since the bounds are used to compare DFM against GPT-2 in Table 2, it would help to know whether this gap holds, shrinks, or grows at that scale.

3. OT generation quality is tested only at L=128. Overhead and perplexity bounds are validated at longer sequences, but the actual step-reduction experiments aren't. A confirmation at L=1024 would help.

4. The similarity metric space could be explored more. Theorem 3.1 holds for any s(·,·), but experiments mainly use L2 on learnable embeddings. More systematic comparison studying how the choice of similarity metric affects the behaviour could be informative.

5. The perplexity bounds contribution is more or less incremental. The first bound seems to overlap with the ELBO derived by Shaul et al. (2025) via a different route, and the authors acknowledge this.

---

> ### Author Rebuttal · Authors · 2026-03-31
>
> We sincerely thank the reviewer for their time and in-depth review. We are happy the reviewer appreciates the novelty of Theorem 3.1, the usefulness of the bounds, the multimask contribution and the practicality of OT training.
>
> **W1: The presentation...**
>
> We agree and will use the extra page to make the OT introduction more self-contained, expand the intuition behind the necessary conditions in Theorem 3.1 (see our response to `R-1(uekX)`'s W3 and Q3), and make table captions more detailed.
>
> **W2 and Q2: Bound tightness**
>
> Since generating fixed imperfect transition tables is infeasible for larger $V$ and $L$, we instead use an MLP trained via masked flow to model them, which for small $V$ and $L$ closely approximates the true transitions. The rest of settings remain as in `Appendix.-C.5`:
> |Metric|Data-Entropy|Cross-Entropy|bound|
> |-|-|-|-|
> |V=2,L=2|0.471|0.472|0.474|
> |V=4,L=4|2.204|2.287|2.302|
> |V=5,L=5|3.208|3.432|3.503|
>
> When the model is perfect (data entropy $\approx$ cross-entropy, row 1), the bound is tight, as per our theoretical claim. As modelling error grows, so does the bound gap: from 0.4\% at $V=L=4$ (256 states) to 2\% at $V=L=5$ (3125 states, which is near the Monte Carlo practical limit). We expect a larger gap in practice, where the state space is $50257^{128}$. We thank the reviewer for this suggestion, which further supports our claim that the bound tends to the true cross-entropy as the model becomes exact.
>
> We will update `Appx.-C.5` to expand this discussion and include the new experiments.
>
> **W3: OT generation..**
>
> Unfortunately, due to the limited rebuttal period and compute constraints, we were unable to train an MMF model with L = 1024. Nevertheless, we expect the results to extend naturally to larger sequence lengths. The main modification that may be required is retuning the EMA hyperparameters, as changing L can alter gradient magnitudes when EMA is used.
>
> **W4: The similarity..**
>
> We agree this generality can be exploited further and believe it warrants a dedicated paper to do it justice. As a short exploration, we use a learned *cost* function defined as the probability the model generates the target sequence in one step:$$c(x_0, x_1) = - \sum_{i=1}^L \log p_{1|0}^i (x_1^i | x_0; \theta)$$Results are provided in our response to `R-4(jU4y)`'s Q5.
>
> **W5: The perplexity bounds..**
>
> Indeed, as we stated in the paper, the work of Shaul et al, independently and concurrently derived the first bound in our paper. We emphasize, however, that the two works were developed concurrently, and as the reviewer highlights, our derivation offers a complementary perspective. We thank the reviewer for the careful reading.
>
> **Q1: How sensitive..**
>
> We thank the reviewer for the important question. We provide the results when using 256 masks below:
>
> Generative perplexities as measured by GPT2-large.
> ||8|16|32|64|128|1024|
> |-|-|-|-|-|-|-|
> |No OT|539.38|286.57|199.86|167.27|152.09|140.24|
> |Exact-OT-EMA|512.62|276.28|198.20|164.62|150.10|139.51|
>
> Generative perplexities as measured by Llama 3.1 8B.
>
> ||8|16|32|64|128|1024|
> |-|-|-|-|-|-|-|
> |No OT|625.64|343.13|243.12|205.47|189.27|176.92|
> |Exact-OT-EMA|592.07|332.27|242.38|204.51|185.18|175.23|
>
> Entropy is virtually the same between the two, differing by 0.01 at maximum.
>
> Clearly in this case the difference between the OT and the normal model is  smaller, showing the importance of splitting the grid into tiny parts with mass $\frac{1}{V^L}$ in order to enable proper couplings. Luckily, the amount of compute is completely independent from the number of masks, so zero compute overhead is added when we increase $V_s$.
>
> Below we provide the perplexity bound results,
>
> |Set|Lambada|Wikitext2|PTB|Wikitext103|LM1B|OWT test|
> |-|-|-|-|-|-|-|
> |No OT|70.27|65.62|204.31|65.62|82.05|39.06|
> |Exact-OT-EMA|67.07|63.21|199.46|63.39|77.70|37.64|
>
> **Q3: Any intuitive..**
>
> Our intuition is the following: the change in couplings between iterations disrupts the learning of a single flow, while embedding updates during flow training alter the optimal coupling. By using moving-average embeddings for OT, we decouple model embeddings from those used in minibatch coupling, so these two optimization procedures do no disrupt one-another as much.
>
> **Q4: For the L2..**
>
> From our experience, generative perplexity tends to change by roughly $\pm$ 1%.
>
> **Q5: The improvement..**
>
> For training, we increased $L$ to 1024, for 1000 iterations, and the overhead dropped to 1.9% for the Exact-EMA case. For generation, wall-clock time is reduced by $8\times$ when the number of steps is reduced by $8\times$. The time per generation-step for the non-OT case was 0.08% slower than for the OT-trained model, but this is within noise. As shown in our rebuttal to `R-2(3Dpn)` (Significance), the number of steps is reduced 32 times in the DFM-B experiments (from 1024 to 32), making generation $32\times$ faster.
>
> We again sincerely thank the reviewer for their thorough and constructive review.

---

> > ### Author Rebuttal · Reviewer_K83J · 2026-04-02
> >
> > Thank you for providing the clarifications and additional results, I believe my concerns have been adequately addressed.

---

> > > ### Author Response · Authors · 2026-04-07
> > >
> > > We sincerely thank the reviewer for their time, their careful evaluation of the paper, and their engagement with our rebuttal. We appreciate the positive assessment and are glad that the remaining concerns have been addressed.

---

### Official Review · Reviewer_3Dpn · 2026-03-12

**Soundness:** 4
**Presentation:** 3
**Significance:** 4
**Originality:** 3
**Overall Recommendation:** 5
**Confidence:** 4

**Summary:**

The paper improves the discrete flow matching algorithm for categorical sequence generation by explicitly minimizing the expected number of transitions along the generated trajectory. In order to do it, they introduce flow-style optimal transport objective and show that it is equivalent to static Kantorovich formulation between the initial random distribution and the target data distribution. They utilize it by computing optimal couplings using entropic OT between source and target minibatch data during training. Since exact perplexity is not tractable for these stochastic discrete flows, authors derive computable upper bounds and on perplexity and use them for evaluation. Additionally, they propose multi-masking for the initial random distribution to make the OT objective non-degenerate. The authors demonstrate empirically that proposed approach reduces the number of transitions and achieves comparable perplexity in substantially fewer inference steps.

**Compliance With Llm Reviewing Policy:**

Affirmed.

**Final Justification:**

The authors rebuttal addressed my concerns and reinforces my positive assessment of the paper. I continue to view the paper as technically strong, well motivated and particularly useful. The paper has careful theoretical analysis and strong empirical validation, including many ablations. Its significance is clear for discrete flow models: it shows the number of inference steps can be reduced significantly while keeping comparable perplexity. In terms of originality, I see the main contribution in the adaptation of OT ideas to discrete flow matching, rather than entirely new OT theory. Overall, the rebuttal clarified the scope of the model and its relation to prior work, but didn't change my overall view. I therefore maintain my accept recommendation.

**Key Questions For Authors:**

None

**Limitations:**

Yes

**Strengths And Weaknesses:**

Soundness. The claims of the paper are supported by the careful theoretical analysis and extensive empirical validation. The authors run many ablations to justify key design choices (e.g. Sinkhorn regularization, OT batch size), and they study the tightness of the perplexity bounds in simplified settings. The main remaining gap is the absence  of comparison to continuous-flow baselines and diffusion models in embedding space, which would clarify the benefit of discrete OT formulation further, but it doesn't undermine the correctness of proposed methods.

Presentation. The paper is generally well-written: the main text explains the contribution and the setup cleanly, the appendix is comprehensive, and the experiments are described with sufficient detail. My main concern is accessibility: this is a math-heavy paper, and some ideas (for example, the dynamic OT viewpoint) could benefit from more intuition. Several proofs and derivations in the appendix are lengthy, and some of them might be derived from the existing literature, rather than reproducing large formulas. Overall, the presentation is good but could be more reader-friendly.

Significance. The paper is significant for engineering of discrete flow models -- reducing the number of inference steps is a clear gain for discrete generators; if the result about "same perplexity with 8 times less steps" holds broadly, this is a clear win. The method is easy to adopt and appears empirically stable. The math is not fundamentally new, but justifies why OT helps to reduce number of jumps and provides computable bounds for benchmarking. The tightness of bounds is studied.

Originality. The paper is practically useful and well motivated, but some of its theory is based on standard tools from mathematical OT and Markov chain literature. In particular, dynamic OT viewpoint for discrete flows is well-established in math literature (https://arxiv.org/abs/1102.5238, https://arxiv.org/pdf/1308.0226); same for perplexity analysis -- it sounds like a direct adaptation of well-known formulas for Markov jump processes to the discrete flow matching (as also suggested by their citation of Opper & Sanguinetti). The main originality therefore lies less in KL/OT formulas, but more in applying them effectively to DFM, providing a workable minibatch entropic OT (Sinkhorn) during training, and introducing multi-mask source to make coupling non-degenerate.

---

> ### Author Rebuttal · Authors · 2026-03-31
>
> We sincerely thank the reviewer for their time and in-depth review. We are very glad to hear that the reviewer appreciates the careful theoretical analysis and extensive empirical validation, including the many ablations.
>
> **Soundness. The claims of the paper...**
>
> We thank the reviewer for recognizing the careful theoretical analysis and extensive empirical validation, including the ablations supporting key design choices such as Sinkhorn regularization and OT batch size, as well as the study of the tightness of the perplexity bounds in simplified settings. Regarding comparisons with continuous-flow baselines, to our knowledge continuous flow or diffusion models of comparable size currently underperform strong discrete models. For this reason, consistent with most prior work in this area, we focused our comparisons on discrete flow, discrete diffusion, and autoregressive baselines. That said, we agree that such a comparison could further clarify the scope of the benefits, and we intend to add an additional comparison table in the final version if the paper is accepted.
>
> **Presentation. The paper is generally well-written...**
>
> We thank the reviewer for the positive assessment of the presentation and for noting that the main text explains the contribution and setup clearly, that the appendix is comprehensive, and that the experiments are described in sufficient detail. We also appreciate the concern about accessibility. This was an important consideration for us, and we made a strong effort to provide enough background and explanation to reach a broader audience. In the revision, and especially taking advantage of the additional page allowed in the camera-ready version, we intend to include more background on optimal transport, provide more intuition for Theorem 3.1, and expand the discussion of the proof intuition for Theorem 4.1. Regarding the length of some proofs, one of our goals was to keep them as elementary and self-contained as possible so that they remain accessible to readers coming from the machine learning community rather than from the optimal transport literature.
>
> **Significance. The paper is significant...**
>
> We are glad the reviewer finds the paper to be significant in the engineering aspect as well. In fact, in the case of DFM-B experiments we found that the number of steps can be reduced 32 times from 1024 to 32 to reach the same generative perplexity without any losses in entropy (diversity):
>
> |Generation Steps|8|16|32|64|128|1024|
> |-|-|-|-|-|-|-|
> |DFM-B|345.94|241.16|211.99|197.48|192.75|185.12|
> |DFM-B-Exact|335.14|235.15|206.11|194.23|188.85|180.91|
> |DFM-B-Exact-EMA|302.76|208.42|182.40|169.89|164.42|159.87|
>
> These rows should be seen as additions to Table 3 in the paper. Similarly, below we show the perplexity bound results which are to augment Table 7.
>
> |Dataset|Lambada|Wiki2|PTB|Wiki3|LM1B|
> |-|-|-|-|-|-|
> |DFM-B|184.81|211.66|723.15|207.73|230.87|
> |DFM-B-Exact|168.02|189.16|676.29|191.16|209.75|
> |DFM-B-Exact-EMA|167.10|175.09|618.78|176.53|204.14|
>
>
> **Originality. The paper is practically useful..**
>
> We thank the reviewer for pointing out the foundational literature on dynamic OT in discrete spaces (e.g., Maas, 2011; Leonard, 2013), which we will cite and discuss more prominently in the revision. We agree that the general dynamic OT viewpoint is classical. Our claim is narrower: Theorem 3.1 establishes a dynamic--static equivalence tailored to modern DFM settings that, to our knowledge, is not covered directly by those prior frameworks. In particular, our result allows (i) arbitrary similarity functions rather than intrinsic graph metrics, which is important for embedding-based costs such as squared $L_2$, and (ii) an exact identity for any coupling $\pi(x_0, x_1)$, not only at the optimum, which is what supports the minibatch OT training procedure used in practice. Similarly, regarding the perplexity bounds, we do adapt the approach of Opper \& Sanguinetti for discrete flows, but this requires nontrivial modifications. In fact, the original paper of Gat et al. that introduced discrete flow matching did not derive such a bound and instead relied on downstream tasks and generative perplexity to measure performance.
>
> We again sincerely thank the reviewer for their thorough and constructive review.

---

> > ### Author Rebuttal · Reviewer_3Dpn · 2026-04-04
> >
> > Thank you for the response. My questions and suggestions were addressed

---

> > > ### Author Response · Authors · 2026-04-07
> > >
> > > We sincerely thank the reviewer for their time, their careful evaluation of the paper, and their engagement with our rebuttal. We deeply appreciate the positive assessment of our work.

---

### Official Review · Reviewer_uekX · 2026-03-13

**Soundness:** 3
**Presentation:** 3
**Significance:** 3
**Originality:** 3
**Overall Recommendation:** 5
**Confidence:** 3

**Summary:**

This paper studies discrete flow matching for categorical data and addresses two central challenges: path rectification and principled evaluation without exact likelihoods. It introduces a dynamic optimal transport objective that yields a categorical Benamou-Brenier-style result, derives computable perplexity upper bounds as a proxy for training and evaluation, and proposes Multimask Flows (MMF) to enable meaningful OT couplings in masked settings. Experiments on GPT-2-scale language models show that minibatch OT can reduce generation steps by up to 8x while largely preserving judge-based generative perplexity. The supplementary material further strengthens the paper with detailed proofs, additional timing and ablation results, correlation analysis between the bound and generative perplexity, and non-cherry-picked generated samples.

**Compliance With Llm Reviewing Policy:**

Affirmed.

**Final Justification:**

The authors well answered the my questions

**Key Questions For Authors:**

1.The paper and supplementary material show that OT helps within the MMF setting. Could the authors further clarify how much of the overall gain comes from the OT alignment itself versus the MMF design, perhaps through a cleaner decomposition or additional apples-to-apples ablations?
2.The supplementary material suggests that OT adds only modest end-to-end overhead in the reported GPT-2-scale setup. Do the authors have additional evidence or intuition about how this overhead would scale to substantially larger models or training regimes?
3.The main theoretical results are developed under assumptions such as convex interpolants and per-position similarity measures. Could the authors comment more explicitly on which parts of the theory are expected to generalize beyond the current formulation, and which parts are more tightly tied to this specific setup?
4.The supplementary material adds judge-based metrics, entropy, and non-cherry-picked samples. Could the authors provide further discussion of generated text quality, for example in terms of coherence, readability, or downstream behavior, to better connect the proposed method to practical text generation performance?

**Limitations:**

The paper discusses several technical limitations implicitly through its experiments and supplementary material, but the limitations section could be more explicit. In particular, it would be helpful to more clearly discuss the current scalability boundary of minibatch OT, the extent to which the theory depends on specific assumptions such as convex interpolants, and the fact that current text-quality validation remains indirect. I do not see major immediate negative societal-impact concerns beyond those common to text generation models, though a brief acknowledgement of potential misuse of generated text would improve completeness.

**Strengths And Weaknesses:**

This is a technically strong and meaningful paper on discrete flow matching for categorical data. A major strength is the theoretical contribution: the paper formulates a dynamic OT objective for discrete flows and derives a categorical Benamou-Brenier-style result, providing a nontrivial bridge between continuous OT-style rectification and discrete generative modeling. The derivation of computable perplexity upper bounds is also valuable, since it addresses an important evaluation gap in DFM and offers a principled proxy for training and comparison when exact likelihoods are unavailable. The proposed MMF design is well motivated and practically useful, since it makes OT-based couplings meaningful in masked settings. Empirically, the paper demonstrates substantial generation-step reduction while largely preserving judge-based generative perplexity, and the supplementary material significantly strengthens the work with detailed proofs, timing analysis, ablations, entropy results, correlation analysis, and non-cherry-picked samples.

The main weaknesses are about scope and completeness rather than core soundness. First, the relative contribution of MMF versus OT is still not entirely clear, because MMF itself changes the source distribution and vocabulary structure. Second, while the supplementary material suggests that OT adds only modest overhead in the reported GPT-2-scale setup, the evidence for much larger models or training regimes remains limited. Third, the theoretical results are developed under specific assumptions such as convex interpolants and per-position similarity measures, so their broader generality is not fully established. Finally, although the paper goes beyond a single perplexity metric by including judge-based evaluation and entropy, the practical relevance would be stronger with more direct validation of text quality, such as coherence-focused analysis, human evaluation, or downstream zero-shot performance. Overall, I find the paper strong enough to lean positive, but with some reservations.

---

> ### Author Rebuttal · Authors · 2026-03-31
>
> We sincerely thank the reviewer for their time and in-depth review. We appreciate that the reviewer recognizes the paper’s theoretical contributions, including the OT framework and perplexity bound, as well as the empirical evaluation, notably the timing analysis, ablations, entropy results, correlation analysis, and non-cherry-picked samples.
>
> We address the remaining questions and concerns below.
>
> **W1 and Q1: The paper and supplementary material...**
>
> Due to space constraints, we address this point in our response to `R-4(jU4y)`'s **W1 and Q1**.
>
> **W2 and Q2: The supplementary material suggests that OT adds only modest end-to-end..**
>
> We appreciate this question, as it highlights an important strength of our approach. The computational overhead of minibatch OT is independent of the underlying neural network architecture. As model size increases, the relative overhead introduced by OT decreases, since the OT cost remains effectively constant with respect to model size. Importantly, OT is also unaffected by vocabulary size. The only factors tied to the neural network that influence OT are the sequence length ($L$) and embedding dimension ($D$), as these determine the cost of computing pairwise distances between embeddings. However, OT scales linearly with both $L$ and $D$. In contrast, the transformer architecture scales quadratically with respect to both $L$ and $D$. As a result, as model size grows, the OT component becomes increasingly negligible relative to the overall training cost. Empirically, for $L = 1024$, the OT overhead drops to approximately 1.9\%.
>
> **W3 and Q3: The main theoretical results are developed under assumptions such as convex interpolants and per-position similarity measures. Could the authors comment more explicitly on which parts of the theory are expected to generalize beyond the current formulation, and which parts are more tightly tied to this specific setup?**
>
> We thank the reviewer for this insightful question. We address generality at two levels: (i) the bounds, and (ii) the OT result.
>
> *1. Generality of the bounds (Theorem 4.1)*
>
> Theorem 4.1 holds for general discrete flows; the specialization to convex interpolants is only used to derive Equation (15). This is not restrictive in practice, as widely used flows (masked, uniform, BoW, multi-mask) fall into this class.
>
> *2. Generality of the OT result (Theorem 3.1).*
>
> The strength of Theorem 3.1 comes from its specificity to convex interpolants, which allows the use of arbitrary similarity functions $s$, unlike the $L_2$-restricted classical Benamou--Brenier setting.
>
> A key difference is that, in the classical case, minimizing the Kantorovich objective does not necessarily reduce the dynamic cost unless the optimal coupling is reached. In contrast, under convex interpolants, we show that the dynamic and Kantorovich costs coincide for any coupling. Thus, improving the coupling directly improves the dynamic objective.
>
> The proof of this theorem relies critically on the convex-interpolant structure. In particular, conditions such as $\delta_{x_1^i}(x^i)$ together with $x^i \neq x_t^i$ imply $x_t^i = x_0^i$, enabling simplifications such as reducing $s(x_t^i, x^i)$ to $s(x_0^i, x_1^i)$. The per-position similarity assumption is also essential, as marginalization over $x_t^{-i}$ (step (24)$\to$(25)) would not be tractable for a general $s(x_t, x)$.
>
> *Summary*
>
> Theorem 4.1 is broadly applicable, while Theorem 3.1 intentionally leverages convex interpolants and per-position similarities. This design matches the typical flow/diffusion models and enables stronger guarantees than more general OT formulations.
>
> **W4 and Q4: The supplementary material adds judge-based metrics, entropy, and non-cherry-picked samples. Could the authors provide further discussion of generated text quality, for example in terms of coherence, readability, or downstream behavior, to better connect the proposed method to practical text generation performance?**
>
> We thank the reviewer for this question. As the reviewer notes, we report several complementary indicators of generation quality, including perplexity, generative perplexity, entropy, and non-cherry-picked samples. This allows us to fulfill our goal of establishing a foundational framework for discrete normalizing flows and to evaluate it primarily through statistical modeling quality and sampling efficiency, in line with prior work. We agree that downstream metrics such as coherence or readability would also be valuable, but at the GPT-2 scale such differences are often difficult to measure reliably. We believe this is why prior work at this scale typically focuses on sampling efficiency and core statistical modeling performance, where distinctions are easier to detect.
>
>
> We again sincerely thank the reviewer for their thorough and constructive review.

---

> > ### Author Rebuttal · Reviewer_uekX · 2026-04-07
> >
> > thanks for your answering

---

> > > ### Author Response · Authors · 2026-04-07
> > >
> > > We sincerely thank the reviewer for their time, their careful evaluation of the paper, and their engagement with our rebuttal. We greatly appreciate the increased score.

---

### Decision · Program_Chairs · 2026-04-30

**Decision:**

Accept (regular)

**Comment:**

The present paper generalizes the concept of minibatched couplings from continuous to discrete flow matching. Experimentally, this results in a significant reduction in generation steps at preserved output quality at little computational overhead. All reviewers agree that this is a technically strong paper with significant empirical consequence. I therefore recommend acceptance.

For the camera ready version, I encourage the authors to extend the paper with the L=1024 setup requested by reviewer jU4y, in addition to the changes promised in the rebuttal. In particular, the authors made a great job defending the choices in their theory in the rebuttal, and this should be reflected in the paper.